# AC-PKAN: Attention-Enhanced and Chebyshev Polynomial-Based Physics-Informed Kolmogorov–Arnold Networks

**Hangwei Zhang[1,2]**   **Zhimu Huang[3]**   **Yan Wang[1]***
**[1]Institute for AI Industry Research, Tsinghua University**
**[2]Beihang University**
**[3]Beijing Institute of Technology**
`wangyan@air.tsinghua.edu.cn, hangweizhang@buaa.edu.cn, 1120222421@bit.edu.cn`

**Reviewed on OpenReview:** `https://openreview.net/forum?id=J4SkwpIgj7`

## Abstract

Kolmogorov–Arnold Networks (KANs) have recently shown promise for solving partial differential equations (PDEs). Yet their original formulation is computationally and memory intensive, motivating the introduction of Chebyshev Type-I-based KANs (Cheby1KANs). Although Cheby1KANs have outperformed the vanilla KANs architecture, our rigorous theoretical analysis reveals that they still suffer from rank collapse, ultimately limiting their expressive capacity. To overcome these limitations, we enhance Cheby1KANs by integrating wavelet-activated MLPs with learnable parameters and an internal attention mechanism. We prove that this design preserves a full-rank Jacobian and is capable of approximating solutions to PDEs of arbitrary order. Furthermore, to alleviate the loss instability and imbalance introduced by the Chebyshev polynomial basis, we externally incorporate a Residual Gradient Attention (RGA) mechanism that dynamically re-weights individual loss terms according to their gradient norms and residual magnitudes. By jointly leveraging internal and external attention, we present AC-PKAN, a novel architecture that constitutes an enhancement to weakly supervised Physics-Informed Neural Networks (PINNs) and extends the expressive power of KANs. Experimental results from nine benchmark tasks across three domains show that AC-PKAN outperforms or matches state-of-the-art models such as PINNsFormer, establishing it as a highly effective tool for solving complex real-world engineering problems in zero-data or data-sparse regimes. The code is publicly available at `https://github.com/fogradio/ACPKAN`.

## 1 Introduction

Numerical solutions of partial differential equations (PDEs) are essential in science and engineering (Zienkiewicz & Taylor, 2005; Liu, 2009; Fornberg, 1998; Brebbia et al., 2012). Physics-informed neural networks (PINNs) (Lagaris et al., 1998; Raissi et al., 2019) have emerged as a promising approach in scientific machine learning (SciML), especially when data are unavailable or scarce. Traditional PINNs typically employ multilayer perceptrons (MLPs) (Cybenko, 1989) due to their ability to approximate nonlinear functions (Hornik et al., 1989) and their success in various PDE-solving applications (Yu et al., 2018; Han et al., 2018).

However, PINNs encounter limitations, including difficulties with multi-scale phenomena (Kharazmi et al., 2021), the curse of dimensionality in high-dimensional spaces (Jagtap & Karniadakis, 2020), and challenges with nonlinear PDEs (Yuan et al., 2022). These issues arise from both the complexity of PDEs and limitations in PINN architectures and training methods. To address these challenges, existing methods focus on improving both the internal architecture of PINNs and their external learning strategies. Internal improvements include novel architectures like Quadratic Residual Networks (Qres) (Bu & Karpatne, 2021),

First-Layer Sine (FLS) (Wong et al., 2022), and PINNsformer (Zhao et al., 2023). External strategies are discussed in detail in Section 2. Nevertheless, traditional PINNs based on MLPs still suffer from issues like lack of interpretability (Cranmer, 2023), overfitting, vanishing or exploding gradients, and scalability problems (Bachmann et al., 2024). As an alternative, Kolmogorov–Arnold Networks (KANs) (Liu et al., 2024b), inspired by the Kolmogorov–Arnold representation theorem (Kolmogorov, 1961; Braun & Griebel, 2009), have been proposed to offer greater accuracy and interpretability. KANs can be viewed as a combination of Kolmogorov networks and MLPs with learnable activation functions (Köppen, 2002; Sprecher & Draghici, 2002). Various KAN variants have emerged by replacing the B-spline functions (SS, 2024; Bozorgasl & Chen, 2024; Xu et al., 2024a). Although they still face challenges (Yu et al., 2024), KANs have shown promise in addressing issues like interpretability (Liu et al., 2024a) and catastrophic forgetting (Vaca-Rubio et al., 2024) in learning tasks (Samadi et al., 2024). Recent architectures like KINN (Wang et al., 2024b) and DeepOKAN (Abueidda et al., 2024) have applied KANs to PDE solving with promising results.

Despite the potential of KANs, the original KAN suffers from high memory consumption and long training times due to the use of B-spline functions (Shukla et al., 2024). To address these limitations, we propose the Attention-Enhanced and Chebyshev Polynomial-Based Physics-Informed Kolmogorov–Arnold Networks (AC-PKAN). Our approach replaces B-spline functions with first-kind Chebyshev polynomials, forming the Cheby1KAN layer (SS, 2024), eliminating the need for grid storage and updates. Nevertheless, networks composed solely of stacked Cheby1KAN layers exhibit pronounced rank diminution (Feng et al., 2022). By integrating Cheby1KAN with linear layers and incorporating internal attention mechanisms derived from input features, AC-PKAN addresses these limitations while efficiently modeling complex nonlinear functions and selectively emphasizing distinct aspects of the input features at each layer. Additionally, we introduce an external attention mechanism that adaptively reweights loss terms according to both gradient norms and point-wise residuals, thereby counteracting the large polynomial expansions and gradient magnitudes inherent in Cheby1KAN, mitigating residual imbalance and gradient flow stiffness, and ultimately enhancing training stability and efficiency. To our knowledge, AC-PKAN is the first PINN framework to integrate internal and external attention mechanisms into KAN layers, effectively addressing many issues of original KANs and PINNs. Our key contributions can be summarized as follows:

- **Rigorous theoretical analysis**. We provide the first formal study of *Cheby1KAN* depth, proving upper bounds on each layer's Jacobian rank and showing that stacked layers suffer an exponential rank–attenuation in depth, which establishes the theoretical limits that motivate our design.

- **Attention-enhanced internal architecture**. To overcome rank collapse and the zero-derivative pathology, we introduce *AC-PKAN*: Cheby1KAN layers are interleaved with linear projections, learnable wavelet activations, and a lightweight feature–wise attention module, together guaranteeing full-rank Jacobians and non-vanishing derivatives of any finite order.

- **Residual–Gradient Attention (RGA)**. Externally, we devise an adaptive loss–reweighting strategy that couples point-wise residual magnitudes with gradient norms. It dynamically balances competing objectives, alleviates gradient stiffness, and accelerates convergence of physics-informed neural networks.

- **Comprehensive experimental validation**. Across three categories of nine benchmark PDE problems and twelve competing models, AC-PKAN attains the best or near-best accuracy in every case, demonstrating superior generalization and robustness to PINN failure modes.

## 2 Related Works

**External Learning Strategies for PINNs.** Most advances in physics-informed neural networks improve training from the outside while keeping an MLP-like backbone. Loss rebalancing methods such as PINN-LRA, PINN-NTK and residual-based adaptation (Wang et al., 2021; 2022b; Anagnostopoulos et al., 2024) adjust PDE, boundary and initial terms with gradient or NTK statistics to correct the mismatch among losses. Sampling based approaches follow the same goal. AAS selects collocation points using adversarial optimal transport (Tang et al., 2023), RoPINN applies regional Monte Carlo sampling (Pan et al., 2024), while RAR

and PINNACLE resample high-residual areas and co-optimise all point types (Wu et al., 2023; Lau et al., 2024). Other works modify the objective or the training domain. gPINN adds gradient terms to enforce the PDE (Yu et al., 2022), vPINN uses a variational form (Kharazmi et al., 2019), LAAF and GAAF tune activations during training to accelerate convergence (Jagtap et al., 2020a;b), and FBPINN and hp-VPINN decompose the domain to make multi-scale problems tractable (Moseley et al., 2023; Kharazmi et al., 2021). These methods stabilise PINNs but they still rely on feature spaces that can degenerate when several operators or high order derivatives are imposed. Our approach keeps these mature external pipelines while adding an internal mechanism whose role is to preserve expressive bases during physics-informed optimisation.

**KAN and Chebyshev-based Variants.** Kolmogorov–Arnold Networks make the activation on edges learnable and can approximate operators with smaller models (Liu et al., 2024b). Follow up designs replace splines with faster or more structured bases, such as FastKAN with RBFs (Li, 2024), Cheby1KAN and Cheby2KAN with first and second kind polynomials for oscillatory targets (SS, 2024), rKAN and fKAN with rational or fractional Jacobi bases (Aghaei, 2024; 2025), and FourierKAN with Fourier modes (Mehrabian et al., 2024). Surveys place these models in a wider landscape of Kolmogorov-inspired approximators (Guilhoto & Perdikaris, 2024), and preliminary benchmarks still report Chebyshev-based variants as a strong speed–accuracy choice (SS, 2024). Yet most of these works optimise for approximation, interpretability or inference cost and do not discuss how to keep KANs stable when trained with collocation-based PDE losses. AC-PKAN is positioned at this intersection. It retains the approximation benefits of Chebyshev KANs, but augments them with internal feature re-injection, frequency-aware activation and rank-aware gating so that the model remains expressive under the same loss reweighting and sampling strategies used by advanced PINNs.

## 3 Motivation and Methodology

**Preliminaries:** Let $\Omega \subset \mathbb{R}^d$ be an open set with boundary $\partial\Omega$. Consider the PDE:

$$\begin{aligned}
\mathcal{D}[u(\boldsymbol{x},t)] &= f(\boldsymbol{x},t), \quad (\boldsymbol{x},t) \in \Omega, \\
\mathcal{B}[u(\boldsymbol{x},t)] &= g(\boldsymbol{x},t), \quad (\boldsymbol{x},t) \in \partial\Omega,
\end{aligned} \tag{1}$$

where $u$ is the solution, $\mathcal{D}$ is a differential operator, and $\mathcal{B}$ represents boundary/initial constraints or available data samples. Let $\hat{u}$ be a neural network approximation of $u$. PINNs minimize the loss:

$$\mathcal{L}_{\text{PINNs}} = \lambda_r \sum_{i=1}^{N_r} \|\mathcal{D}[\hat{u}(\boldsymbol{x}_i,t_i)] - f(\boldsymbol{x}_i,t_i)\|^2 + \lambda_b \sum_{i=1}^{N_b} \|\mathcal{B}[\hat{u}(\boldsymbol{x}_i,t_i)] - g(\boldsymbol{x}_i,t_i)\|^2, \tag{2}$$

where $\{(\boldsymbol{x}_i,t_i)\} \subset \Omega$ are residual points, $\{(\boldsymbol{x}_i,t_i)\} \subset \partial\Omega$ are boundary/initial constraints or available data samples, and $\lambda_r$, $\lambda_b$ balance the loss terms. The goal is to train $\hat{u}$ to minimize $\mathcal{L}_{\text{PINNs}}$ using machine learning techniques.

### 3.1 Chebyshev1-Based Kolmogorov-Arnold Network Layer

Unlike traditional Kolmogorov-Arnold Networks (KAN) that employ spline coefficients, the *First-kind Chebyshev KAN Layer* leverages the properties of mesh-free Chebyshev polynomials to enhance both computational efficiency and approximation accuracy (SS, 2024; Shukla et al., 2024).

Let $\mathbf{x} \in \mathbb{R}^{d_{\text{in}}}$ denote the input vector, where $d_{\text{in}}$ is the input dimensionality, and let $d_{\text{out}}$ be the output dimensionality. Cheby1KAN aims to approximate the mapping $\mathbf{x} \mapsto \mathbf{y} \in \mathbb{R}^{d_{\text{out}}}$ using Chebyshev polynomials up to degree $N$. For $x \in [-1,1], n = 0,1,\ldots,N$, the Chebyshev polynomials of the first kind, $T_n(x)$, are defined as:

$$T_n(x) = \cos\left(n \arccos(x)\right). \tag{3}$$

To ensure the input values fall within the domain $[-1,1]$, Cheby1KAN applies the hyperbolic tangent function for normalization:

$$\tilde{\mathbf{x}} = \tanh(\mathbf{x}). \tag{4}$$

Defining a matrix of functions $\mathbf{\Phi}(\tilde{\mathbf{x}}) \in \mathbb{R}^{d_{\text{out}} \times d_{\text{in}}}$, where each element $\Phi_{k,i}(\tilde{x}_i)$ depends solely on the $i$-th normalized input component $\tilde{x}_i$ for $k = 1, 2, \ldots, d_{\text{out}}, i = 1, 2, \ldots, d_{\text{in}}$:

$$\Phi_{k,i}(\tilde{x}_i) = \sum_{n=0}^{N} C_{k,i,n} \, T_n(\tilde{x}_i). \tag{5}$$

Here, $C_{k,i,n}$ are the learnable coefficients. The output vector $\mathbf{y} \in \mathbb{R}^{d_{\text{out}}}$ is computed by summing over all input dimensions:

$$y_k = \sum_{i=1}^{d_{\text{in}}} \Phi_{k,i}(\tilde{x}_i), \quad k = 1, 2, \ldots, d_{\text{out}}, \tag{6}$$

For a network comprising multiple Chebyshev KAN layers, the forward computation can be viewed as a recursive application of this process. Let $\mathbf{x}_l$ denote the input to the $l$-th layer, where $l = 0, 1, \ldots, L-1$. After applying hyperbolic tangent function to obtain $\tilde{\mathbf{x}}_l = \tanh(\mathbf{x}_l)$, the computation proceeds as follows:

$$\mathbf{x}_{l+1} = \underbrace{\begin{pmatrix} \Phi_{l,1,1}(\cdot) & \Phi_{l,1,2}(\cdot) & \cdots & \Phi_{l,1,n_l}(\cdot) \\ \Phi_{l,2,1}(\cdot) & \Phi_{l,2,2}(\cdot) & \cdots & \Phi_{l,2,n_l}(\cdot) \\ \vdots & \vdots & \ddots & \vdots \\ \Phi_{l,n_{l+1},1}(\cdot) & \Phi_{l,n_{l+1},2}(\cdot) & \cdots & \Phi_{l,n_{l+1},n_l}(\cdot) \end{pmatrix}}_{\mathbf{\Phi}_l} \tilde{\mathbf{x}}_l, \tag{7}$$

A general cheby1KAN network is a composition of $L$ layers: given an input vector $\mathbf{x}_0 \in \mathbb{R}^{n_0}$, the overall output of the KAN network is:

$$\text{Cheby1KAN}(\mathbf{x}) = (\mathbf{\Phi}_{L-1} \circ \mathbf{\Phi}_{L-2} \circ \cdots \circ \mathbf{\Phi}_1 \circ \mathbf{\Phi}_0)\mathbf{x}. \tag{8}$$

In order to prevent gradient vanishing induced by the use of tanh, we apply Layer-Normalization after Cheby1KAN Layer.

Compared to the original B-spline-based KANs, Chebyshev polynomials of the first kind in Equation (3) concentrate spectral energy in high frequencies with frequencies that increase linearly with the polynomial order $n$ (Xu et al., 2024b; Xiao et al., 2024), while maintaining global orthogonality over the interval $[-1, 1]$:

$$\int_{-1}^{1} \frac{T_m(x)T_n(x)}{\sqrt{1-x^2}} \, dx = \begin{cases} 0 & m \neq n, \\ \pi & m = n = 0, \\ \pi/2 & m = n \neq 0. \end{cases} \tag{9}$$

This global support and slower decay of high-frequency components outperform locally supported B-splines, which lack global orthogonality and have rapidly diminishing high-frequency capture. Furthermore, Cheby1KAN layers require only a coefficient matrix of size (*input_dim*, *output_dim*, *degree*+1), whereas B-spline-based KANs necessitate storing grids of size (*in_features*, *grid_size*+2×*spline_order*+1) and coefficient matrices of size (*out_features*, *in_features*, *grid_size*+ *spline_order*), in addition to generating polynomial bases, solving local interpolation systems, and performing recursive updates to achieve high-order interpolation within their support intervals (Liu et al., 2024b). Hence, the Cheby1KAN layer significantly reduces both computational and memory overhead compared to the original B-spline-based KANs, while more effectively capturing high-frequency features. More details can be found at Appendix B

## 3.2 Rank Diminution in Cheby1KAN Networks

While Cheby1KAN layers offer significant advantages, networks composed solely of stacked Cheby1KAN layers, as presented in equation 8, exhibit pronounced rank diminution (Feng et al., 2022). Consequently, these networks suffer a reduced capacity for feature representation, leading to severe information degradation and loss. We present a detailed derivation and proof of this phenomenon below (Roth & Liebig, 2024). The complete mathematical derivations are provided in Appendix A.

**Definitions.** Consider the $l$-th Cheby1KAN layer with input $x_l \in \mathbb{R}^{d_l}$ and output $x_{l+1} \in \mathbb{R}^{d_{l+1}}$. The layer mapping is defined by

$$x_{l+1,k} = \sum_{i=1}^{d_l} \sum_{n=0}^{N} C_{l,k,i,n} T_n(\tanh(x_{l,i})), \tag{10}$$

where $T_n$ are Chebyshev polynomials and $C_{l,k,i,n}$ are learnable coefficients. The Jacobian $J_l \in \mathbb{R}^{d_{l+1} \times d_l}$ has entries

$$J_{l,k,i} = \sum_{n=0}^{N} C_{l,k,i,n} T_n'(\tanh(x_{l,i})) \cdot (1 - \tanh^2(x_{l,i})). \tag{11}$$

For an $L$-layer network, the total Jacobian is

$$J_{\text{total}} = J_{L-1} J_{L-2} \cdots J_0. \tag{12}$$

**Theorem 1** (Single Cheb1KAN Layer Rank Constraint). *The Jacobian $J_l$ satisfies*

$$rank(J_l) \leq \min\{d_{l+1}, d_l(N+1)\}. \tag{13}$$

**Theorem 2** (Nonlinear Normalization Effect). *The normalization $\tanh(x)$ in Cheby1KAN layer reduces the numerical rank $Rank_\epsilon(J)$ of the Jacobian.*

**Theorem 3** (Exponential Decay in Infinite Depth). *When the coefficients $C_{l,k,i,n}$ are drawn from mutually independent Gaussian distributions, the numerical rank of $J_{total}$ decays exponentially to 1 as the depth $L$ of the Cheby1KAN network increases.*

In summary, Cheby1KAN networks inherently experience rank diminution due to various factors. Collectively, the bounded rank per Cheby1KAN layer (Theorem 1), the attenuation from $\tanh(\cdot)$ (Theorem 2), and the multiplicative rank bound culminate in exponential rank decay (Theorem 3), thereby demonstrating the inherent rank diminution in Cheby1KAN networks.

Therefore, there is a significant need to improve the internal structure of models based on the Cheby1KAN layer, which will be discussed in detail in Section 3.3. Additionally, to address some computational limitations associated with the use of Cheby1KAN, we propose an external attention mechanism, which will be elaborated in Section 3.4. By incorporating both internal and external attention mechanisms, our AC-PKAN model fully leverages the advantages of Chebyshev Type-I polynomials while overcoming their initial drawbacks.

### 3.3 Internal Model Architecture

To resolve the Rank Diminution issue arising from direct stacking of Cheby1KAN layers in network architectures, we propose the AC-PKAN model, featuring an attention-enhanced framework (Wang et al., 2021; 2024a) designed to mitigate feature space collapse. The architecture synergistically combines linear transformations for input-output dimensional modulation, state-of-the-art activation functions, and residual-augmented Cheby1KAN layers. These components are collectively designed to preserve hierarchical feature diversity while capturing high-order nonlinear interactions and multiscale topological dependencies inherent in complex data structures. The algorithm's details are provided in Algorithm 1.

**Linear Upscaling and Downscaling Layers** To modulate the dimensionality of the data, the model employs linear transformations at both the input and output stages. The linear layer is designed to achieve a hybridization of KAN and MLP architectures. Its role as both an initial and final projection is inspired by the Spatio-Temporal Mixer linear layer in the PINNsformer model (Zhao et al., 2023), which enhances spatiotemporal aggregation. The input features $\mathbf{x}$ are projected into a higher-dimensional space, and the final network representation $\alpha^{(L)}$ is mapped to the output space via:

$$\mathbf{h}_0 = \mathbf{W}_{\text{emb}} \mathbf{x} + \mathbf{b}_{\text{emb}}, \quad \mathbf{y} = \mathbf{W}_{\text{out}} \alpha^{(L)} + \mathbf{b}_{\text{out}}, \tag{14}$$

where $\mathbf{W}_{\text{emb}} \in \mathbb{R}^{d_{\text{model}} \times d_{\text{in}}}$, $\mathbf{b}_{\text{emb}} \in \mathbb{R}^{d_{\text{model}}}$, $\mathbf{W}_{\text{out}} \in \mathbb{R}^{d_{\text{out}} \times d_{\text{hidden}}}$, and $\mathbf{b}_{\text{out}} \in \mathbb{R}^{d_{\text{out}}}$ are learnable parameters.

**Adaptive Activation Function** We adopt the state-of-the-art *Wavelet* activation function in the field of PINNs, as detailed in (Zhao et al., 2023). Inspired by Fourier transforms, it introduces non-linearity and effectively captures periodic patterns:

$$\text{Wavelet}(x) = w_1 \sin(x) + w_2 \cos(x), \tag{15}$$

where $w_1$ and $w_2$ are learnable parameters initialized to one. This activation integrates Fourier feature embedding (Wang et al., 2023) and sine activation (Wong et al., 2022). When applied to encoders $U$ and $V$, the *Wavelet* activation preserves the gradient benefits introduced by the triangular activation function while modulating its phase and magnitude. This enhancement boosts representational capacity and facilitates adaptive Fourier embedding, thereby more effectively capturing periodic features and mitigating spectral bias.

**Attention Mechanism** An internal attention mechanism is incorporated by computing two feature representations, $\mathbf{U}$ and $\mathbf{V}$, via the *Wavelet* activation applied to linear transformations of the embedded inputs:

$$\mathbf{U} = \text{Wavelet}(\mathbf{h}_0 \boldsymbol{\Theta}_U + \mathbf{b}_U), \quad \mathbf{V} = \text{Wavelet}(\mathbf{h}_0 \boldsymbol{\Theta}_V + \mathbf{b}_V), \tag{16}$$

where $\boldsymbol{\Theta}_U, \boldsymbol{\Theta}_V \in \mathbb{R}^{d_{\text{model}} \times d_{\text{hidden}}}$ and $\mathbf{b}_U, \mathbf{b}_V \in \mathbb{R}^{d_{\text{hidden}}}$ are learnable parameters.

**Attention Integration** The attention mechanism integrates $\mathbf{U}$ and $\mathbf{V}$ iteratively across Cheby1KAN layers using the following equations:

$$\alpha_0^{(l)} = \mathbf{H}^{(l)} + \alpha^{(l-1)}, \quad \alpha^{(l)} = (1 - \alpha_0^{(l)}) \odot \mathbf{U} + \alpha_0^{(l)} \odot (\mathbf{V} + 1). \tag{17}$$

where $\alpha^{(0)} = \mathbf{U}$ and $\odot$ denotes element-wise multiplication. Here, $\mathbf{H}^{(l)} \in \mathbb{R}^{N \times d_{\text{hidden}}}$ is the output of the $l$-th Cheby1KAN layer after LayerNormalization, and $N$ is the number of nodes.

---

**Algorithm 1** Internal AC-PKAN Forward Pass

---

**Data:** Input data $\mathbf{x}$, Cheby1KAN layer parameters, Wavelet activation parameters
**Initialization:** Randomly initialize weights $\mathbf{W}_{\text{emb}}, \boldsymbol{\Theta}_U, \boldsymbol{\Theta}_V, \mathbf{W}_{\text{out}}$ and biases $\mathbf{b}_{\text{emb}}, \mathbf{b}_U, \mathbf{b}_V, \mathbf{b}_{\text{out}}$

1: **Input embedding:**
$$\mathbf{h}_0 \leftarrow \mathbf{W}_{\text{emb}} \mathbf{x} + \mathbf{b}_{\text{emb}}$$
2: **Compute representations:**
$$\mathbf{U} \leftarrow \text{Wavelet}(\mathbf{h}_0 \boldsymbol{\Theta}_U + \mathbf{b}_U), \quad \mathbf{V} \leftarrow \text{Wavelet}(\mathbf{h}_0 \boldsymbol{\Theta}_V + \mathbf{b}_V)$$
3: **Initialize attention:** $\alpha^{(0)} \leftarrow \mathbf{U}$
4: **for** $l = 1$ **to** $L$ **do**
5: $\quad \mathbf{H}^{(l)} \leftarrow \text{LayerNorm}\big(\text{Cheby1KANLayer}(\alpha^{(l-1)})\big)$
6: $\quad \alpha_0^{(l)} \leftarrow \mathbf{H}^{(l)} + \alpha^{(l-1)}$
7: $\quad \alpha^{(l)} \leftarrow (1 - \alpha_0^{(l)}) \odot \mathbf{U} + \alpha_0^{(l)} \odot (\mathbf{V} + 1)$
8: **end for**
9: **Output prediction:**
$$\mathbf{y} \leftarrow \mathbf{W}_{\text{out}} \alpha^{(L)} + \mathbf{b}_{\text{out}}$$

---

**Approximation Ability** Our AC-PKAN's inherent attention mechanism eliminates the need for an additional bias function $b(x)$ required in previous KAN models to maintain non-zero higher-order derivatives (Wang et al., 2024b). This reduces model complexity and parameter count while preserving the ability to seamlessly approximate PDEs of arbitrary finite order. By ensuring non-zero derivatives of any finite order and invoking the Kolmogorov–Arnold representation theorem, our model can approximate such PDEs.

**Proposition 1.** *Let $\mathcal{N}$ be an AC-PKAN model with $L$ layers ($L \geq 2$) and infinite width. Then, the output $y = \mathcal{N}(x)$ has non-zero derivatives of any finite-order with respect to the input $x$.*

Then we prove that the Jacobian matrix of the AC-PKAN model is full-rank, thereby rigorously precluding degenerate directions in the input space.

**Proposition 2.** *Let $\mathcal{N}$ be an AC-PKAN model with $L$ layers ($L \geq 2$) and infinite width. Then, the Jacobian matrix $J_{\mathcal{N}}(\boldsymbol{x}) = \left[ \frac{\partial \mathcal{N}_i}{\partial x_j} \right]_{m \times d}$ is full rank in the input space $\mathbb{R}^d$.*

This property effectively addresses the internal rank diminution issue of Cheby1KAN networks discussed in Section 3.2, and also ensures stable gradient backpropagation, thereby preventing rank-deficiency-induced training failures in AC-PKAN. The complete mathematical derivations are provided in Appendix A.

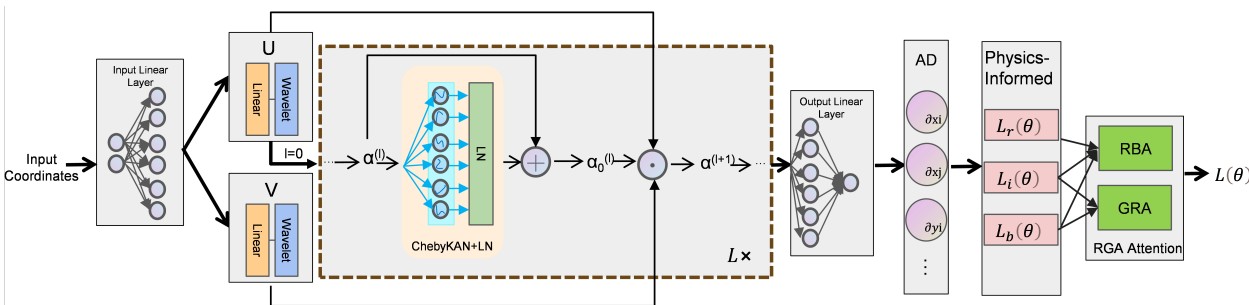

Figure 1: Architecture of the complete AC-PKAN model. It combines its internal attention architecture with an external attention strategy, yielding a weighted loss optimized to obtain the predicted solution.

### 3.4 Residual-and-Gradient Based Attention

In the canonical PINN formulation, the loss is split into an unlabeled PDE-residual term $\mathcal{L}_r$ and a labeled term $\mathcal{L}_d$ that enforces boundary/initial constraints and matches available data samples. To improve optimization efficiency and accuracy and to correct the loss imbalance introduced by Chebyshev bases, we propose Residual Gradient Attention (RGA), an adaptive mechanism that rescales each loss term according to its residual magnitude and its gradient norm. This approach ensures balanced and efficient optimization, particularly addressing challenges with boundary and initial condition losses.

**Residual-Based Attention (RBA)**   Residual-Based Attention (RBA) dynamically amplifies loss terms with the largest point-wise residuals, assigning a tensor of weights $w_{i,j}^{\mathrm{RBA}}$ to each loss component $\mathcal{L}_i$ ($i \in \{r, d\}$) at location $j$ (Anagnostopoulos et al., 2024):

$$w_{i,j}^{\mathrm{RBA}} \leftarrow (1 - \eta)\, w_{i,j}^{\mathrm{RBA}} + \eta\, \frac{|\mathcal{L}_{i,j}|}{\max_j |\mathcal{L}_{i,j}|}, \tag{18}$$

where $\eta$ is the RBA learning rate and $\max_j |\mathcal{L}_{i,r}|$ normalizes by the maximal residual. Residual-Based Attention (RBA) is a lightweight, pointwise weighting scheme that complements the Cheby1KAN layer. Cheby1KAN captures strong nonlinear structure and complex distributions but can exhibit slow or unstable convergence. RBA inserts a self-adjusting feedback loop that reweights local training signals according to residual statistics, focusing learning on poorly fitted locations and reducing the influence of noisy or saturated terms. This synergy alleviates numerical optimization difficulties and enhances global convergence efficiency.

**Gradient-Related Attention (GRA)**   Due to the Cheby1KAN layer's utilization of high-order Chebyshev polynomials, large coefficients and derivative magnitudes are introduced, resulting in an increased maximum eigenvalue of the Hessian and exacerbating gradient flow stiffness. Additionally, nonlinear operations such as $\cos(x)$ and $\arccos(x)$ create regions of vanishing and exploding gradients, respectively. The heightened nonlinearity from these high-degree polynomials further leads to imbalanced loss gradients, intensifying dynamic stiffness. Therefore, we employ Gradient-Related Attention (GRA).

GRA dynamically adjusts weights based on gradient norms of different loss components, promoting balanced training. As a **scalar** applied to one entire loss term, GRA addresses the imbalance where gradient norms of the PDE residual loss significantly exceed those of the data fitting loss (Wang et al., 2021), which can lead to pathological gradient flow issues (Wang et al., 2022b; Fang et al., 2023). Our mechanism smooths weight adjustments, preventing the network from overemphasizing residual loss terms and neglecting other essential physical constraints, thus enhancing convergence and stability.

The GRA weight $\lambda^{\mathrm{GRA}}$ is computed as:

$$\hat{\lambda}_d^{\mathrm{GRA}} = \frac{G_r^{\max}}{\epsilon + \overline{G}_d}, \tag{19}$$

where $G_r^{\max} = \max_p \left\| \frac{\partial \mathcal{L}_r}{\partial \theta_p} \right\|$ is the maximum gradient norm of the residual loss, $\overline{G}_d = \frac{1}{P} \sum_{p=1}^{P} \left\| \frac{\partial \mathcal{L}_d}{\partial \theta_p} \right\|$ is the average gradient norm for $\mathcal{L}_d$, $P$ is the number of model parameters, and $\epsilon$ prevents division by zero.

To smooth the GRA weights over iterations, we apply an exponential moving average:

$$\lambda_d^{\mathrm{GRA}} \leftarrow (1 - \beta_w)\lambda_d^{\mathrm{GRA}} + \beta_w \hat{\lambda}_d^{\mathrm{GRA}}, \tag{20}$$

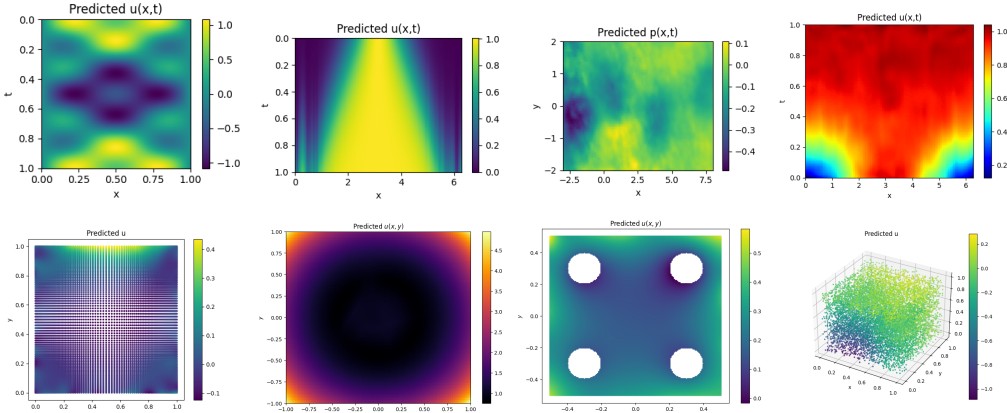

Figure 2: Visualization of AC-PKAN's predicted values for PDE experiments: (Row 1) 1D-Wave, 1D-Reaction, 2D NS Cylinder, 1D-Conv.-Diff.-Reac.; (Row 2) 2D Lid-driven Cavity, Heterogeneous Problem, Complex Geometry, and 3D Point-Cloud.

where $\beta_w$ is the learning rate for the GRA weights. We enforce a minimum value for numerical stability:

$$\lambda_d^{\text{GRA}} \leftarrow \max\left(\lambda_d^{\text{GRA}}, e + \epsilon\right). \tag{21}$$

GRA addresses the aforementioned issues by stabilizing the gradient flow, thereby ensuring more efficient and reliable training of the network. By combining our AC-PKAN internal architecture with the external RGA mechanism, we obtain the complete AC-PKAN model. Figure 1 provides a detailed illustration of our model structure.

---

**Algorithm 2** Implementation of the RGA Mechanism

---

**Data:** Model parameters $\theta$, total number of parameters $P$, learning rate $\alpha$, hyperparameters $\eta$, $\beta_w$, $\epsilon$.
**Initialization:** $w_{r,d}^{\text{RBA}} \leftarrow 0$, $\quad \lambda_d^{\text{GRA}} \leftarrow 1$.

1: **for** each training iteration **do**
2:     Compute gradients:
$$\nabla_\theta \mathcal{L}_i \leftarrow \frac{\partial \mathcal{L}_i}{\partial \theta}, \quad i \in \{r, d\}$$

3:     Update RBA weights for each data point $j$:
$$w_{i,j}^{\text{RBA}} \leftarrow (1 - \eta)w_{i,j}^{\text{RBA}} + \eta\left(\frac{|\mathcal{L}_{i,j}|}{\max_j |\mathcal{L}_{i,j}|}\right), \quad i \in \{r, d\}$$

4:     Compute gradient norms:
$$G_r^{\max} \leftarrow \max_p \|\nabla_{\theta_p} \mathcal{L}_r\|, \quad \overline{G}_i \leftarrow \frac{1}{P}\sum_{p=1}^{P} \|\nabla_{\theta_p} \mathcal{L}_i\|, \quad i \in \{d\}$$

5:     Update GRA weights:
$$\hat{\lambda}_i \leftarrow \frac{G_r^{\max}}{\epsilon + \overline{G}_i}, \quad \lambda_i^{\text{GRA}} \leftarrow (1 - \beta_w)\lambda_i^{\text{GRA}} + \beta_w \hat{\lambda}_i, \quad \lambda_i^{\text{GRA}} \leftarrow \max(\epsilon, \lambda_i^{\text{GRA}}), \quad i \in \{d\}$$

6:     Compute total loss:
$$\mathcal{L}_{\text{RGA}} \leftarrow \lambda_r w_r^{\text{RBA}} \mathcal{L}_r + \sum_{i \in \{d\}} \lambda_i w_i^{\text{RBA}} \log\left(\lambda_i^{\text{GRA}}\right) \mathcal{L}_i$$

7:     Update model parameters:
$$\theta \leftarrow \theta - \alpha \nabla_\theta \mathcal{L}_{\text{RGA}}$$

8: **end for**

---

**Combined Attention Mechanism** To equilibrate the magnitudes of GRA and RBA weights, we apply a logarithmic transformation to the GRA weights when incorporating them into the loss terms, while retaining their original form during weight updates. This preserves the direct relationship between weights and gradient information,

| Model | 1D-Wave | | 1D-Reaction | | 2D NS Cylinder | | 1D-Conv.-Diff.-Reac. | | 2D Lid-driven Cavity | |
|---|---|---|---|---|---|---|---|---|---|---|
| | rMAE | rRMSE | rMAE | rRMSE | rMAE | rRMSE | rMAE | rRMSE | rMAE | rRMSE |
| PINN | 0.3182 | 0.3200 | 0.9818 | 0.9810 | 5.8378 | 4.0529 | 0.0711 | 0.1047 | 0.6219 | 0.6182 |
| QRes | 0.3507 | 0.3485 | 0.9844 | 0.9849 | 25.8970 | 17.9767 | 0.0722 | 0.1062 | **0.5989** | **0.5674** |
| FLS | 0.3810 | 0.3796 | 0.9793 | 0.9773 | 12.4564 | 8.6473 | **0.0707** | 0.1045 | 0.6267 | 0.6267 |
| PINNsFormer | **0.2699** | **0.2825** | 0.0152 | 0.0300 | **0.3843** | **0.2801** | 0.0854 | 0.0927 | OoM | OoM |
| Cheby1KAN | 1.1240 | 1.0866 | 0.0617 | 0.1329 | 3.7107 | 2.7379 | 0.0992 | 0.1644 | 0.5689 | 0.5370 |
| Cheby2KAN | 1.1239 | 1.0865 | 1.0387 | 1.0256 | 72.1708 | 50.1039 | 1.2078 | 1.2059 | 6.1457 | 3.9769 |
| AC-PKAN (Ours) | 0.0011 | 0.0011 | 0.0375 | 0.0969 | 0.2230 | 0.2182 | 0.0114 | 0.0142 | 0.6374 | 0.5733 |
| KINN | 0.3466 | 0.3456 | 0.1314 | 0.2101 | 4.5306 | 3.1507 | 0.0721 | 0.1058 | OoM | OoM |
| rKAN | 247.7560 | 2593.0750 | 65.2014 | 54.8567 | NaN | NaN | 543.8576 | 3053.6257 | OoM | OoM |
| FastKAN | 0.5312 | 0.5229 | 0.5475 | 0.6030 | 25.8970 | 1.4085 | 0.0876 | 0.1219 | OoM | OoM |
| fKAN | 0.4884 | 0.4768 | 0.0604 | 0.1033 | 3.0766 | 2.1403 | 0.1186 | **0.0794** | 0.7639 | 0.7366 |
| FourierKAN | 1.1356 | 1.1018 | 1.4542 | 1.4217 | 9.3295 | 8.0346 | 0.91052 | 0.9708 | OoM | OoM |

Table 1: Combined experimental results across Failure PINN Modes. Results are organized from left to right in the following order: 1D-Wave, 1D-Reaction, 2D NS Cylinder, 1D-Conv.-Diff.-Reac., and 2D Lid-driven Cavity.

| Model | Heterogeneous Problem | | Complex Geometry | | 3D Point-Cloud | |
|---|---|---|---|---|---|---|
| | rMAE | rRMSE | rMAE | rRMSE | rMAE | rRMSE |
| PINN | 0.1662 | 0.1747 | 0.9010 | 0.9289 | 3.0265 | 2.4401 |
| QRes | 0.1102 | 0.1140 | 0.9024 | 0.9289 | 3.6661 | 2.8897 |
| FLS | 0.1701 | 0.1789 | 0.9021 | 0.9287 | 3.1881 | 2.5629 |
| PINNsFormer | 0.1008 | **0.1610** | **0.8851** | **0.8721** | OoM | OoM |
| Cheby1KAN | 0.1404 | 0.2083 | 0.9026 | 0.9244 | 2.4139 | 1.9646 |
| Cheby2KAN | 0.4590 | 0.5155 | 0.9170 | 1.0131 | 4.9177 | 3.5084 |
| AC-PKAN (Ours) | **0.1063** | 0.1817 | 0.5452 | 0.5896 | 0.3946 | 0.3403 |
| KINN | 0.1599 | 0.1690 | 0.9029 | 0.9261 | OoM | OoM |
| rKAN | 24.8319 | 380.5582 | 23.5426 | 215.4764 | 366.5741 | 2527.1180 |
| FastKAN | 0.1549 | 0.1624 | 0.9034 | 0.9238 | OoM | OoM |
| fKAN | 0.1179 | 0.1724 | 0.9043 | 0.9303 | 2.6279 | 2.2051 |
| FourierKAN | 0.4588 | 0.5154 | 1.4455 | 1.5341 | **0.9314** | **1.0325** |

Table 2: Combined experimental results across Complex Engineering Environments. Results are organized from left to right in the following order: Heterogeneous Problem, Complex Geometry, and 3D Point-Cloud.

ensuring sensitivity to discrepancies between residual and data gradients. The logarithmic transformation mitigates magnitude disparities, preventing imbalances among loss terms. It enables GRA weights to adjust more rapidly when discrepancies are minor and ensures stable updates when discrepancies are substantial. The coefficient $\lambda^{\mathrm{GRA}}$ not only attains excessively large values in scale but also exhibits a broad range of variation. In the training process, $\lambda^{\mathrm{GRA}}$ rapidly increases from zero to very large values, demonstrating a wide dynamic range which is shown in Figure 4 in Appendix F. The logarithmic transformation significantly constrains this range; without it, the model cannot accommodate drastic changes in $\lambda^{\mathrm{GRA}}$, and rigid manual scaling factors further exacerbate the imbalance among loss terms, ultimately causing training failure.

By integrating point-wise RBA with term-wise GRA, the total loss under the RGA mechanism is defined as:

$$\mathcal{L}_{\mathrm{RGA}} = \lambda_r w_r^{\mathrm{RBA}} \mathcal{L}_r + \lambda_d w_d^{\mathrm{RBA}} \log\left(\lambda_d^{\mathrm{GRA}}\right) \mathcal{L}_d. \tag{22}$$

where $w^{\mathrm{RBA}}$ are the RBA weights, and $\lambda_d^{\mathrm{GRA}}$ are the GRA weights for boundary/initial conditions or available data samples.

This formulation reweights the residual loss based on its magnitude and adjusts the boundary and initial condition losses according to both their magnitudes and gradient norms, promoting balanced and focused training through a dual attention mechanism.

RGA enhances PINNs by dynamically adjusting loss weights based on residual magnitudes and gradient norms. By integrating RBA and GRA, it balances loss contributions, preventing any single component from dominating the training process. This adaptive reweighting accelerates and stabilizes convergence, focusing on challenging regions with significant errors or imbalanced gradients. Consequently, RGA provides a robust framework for more accurate and efficient solutions to complex differential equations, performing well in our AC-PKAN model and potentially benefiting other PINN variants which is discussed in detail in appendix 4.4.

| Model | rMAE | rMSE | Loss |
|---|---|---|---|
| Cheby1KAN | 0.0179 | 0.0329 | **0.0068** |
| Cheby2KAN | 0.0189 | 0.0313 | 0.0079 |
| MLP | 0.0627 | 0.1250 | 0.1410 |
| AC-PKAN_s | **0.0177** | **0.0311** | 0.0081 |
| KAN | 0.0145 | 0.0278 | 0.0114 |
| rKAN | 0.0458 | 0.0783 | 0.1867 |
| fKAN | 0.0858 | 0.1427 | 0.1722 |
| FastKAN | 0.0730 | 0.1341 | 0.1399 |
| FourierKAN | 0.0211 | 0.0353 | 0.0063 |

(a) Comparison of test rMAE, rMSE, and training Loss

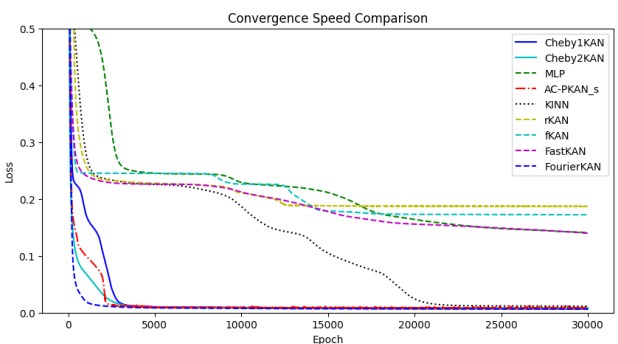

(b) Convergence Comparison of Nine Models

## 4 Experiments

**Goal.** Our empirical study highlights three principal strengths of AC-PKAN: (1) its internal architecture delivers powerful symbolic representation and function-approximation capabilities; (2) it significantly improves generalization abilities and mitigates failure modes compared to PINNs and other KAN variants; and (3) it achieves superior performance in complex real-world engineering environments. We evaluate our method on three task suites comprising nine benchmarks and compare it to 12 representative architectures, including PINN, PINNsFormer, KAN, and fKAN. Although operator learning frameworks that rely on large volumes of labeled data have recently dominated SciML (Lu et al., 2019; Li et al., 2020a;b; Tripura & Chakraborty, 2022; Calvello et al., 2024), our work remains within the established remit of PINN refinement studies, which focus on unsupervised or weakly supervised settings (Jagtap & Karniadakis, 2020; Yu et al., 2022; Zhang et al., 2025a; Hou et al., 2024; Li et al., 2023). Operator-learning methods typically require dense supervision and therefore address a different set of trade-offs compared to physics-informed approaches. Nevertheless, we include a head-to-head comparison of operator-learning and PINN-variant baselines under sparse-data conditions in Appendix C.3. The experimental setup was inspired by methodologies in (SS, 2024; Hao et al., 2023; Wang et al., 2024b; Zhao et al., 2023; Wang et al., 2023). In all experiments, the best results are highlighted in bold italics, and the second-best results in bold. For the formal definitions of the evaluation metrics and detailed descriptions of the experimental setup, please refer to appendix E.

### 4.1 Complex Function Fitting

We evaluated the AC-PKAN Simplified model, which retains only the internal architecture, on a challenging function interpolation benchmark and compared its performance to a PINN implemented as an MLP, the original KAN, and several KAN variants. Detailed experimental setups and results are provided in Appendices E and F.

As shown in Figure 3b, the AC-PKAN Simplified model converges more rapidly than MLPs, KAN, and most KAN variants, achieving lower final losses. While Cheby2KAN and FourierKAN demonstrate faster convergence, our model produces smoother fitted curves and exhibits greater robustness to noise, effectively preventing overfitting in regions with high-frequency variations. Performance metrics are presented in Table 3a.

### 4.2 Mitigating Failure Modes in PINNs

We assessed the AC-PKAN model on five complex PDEs known as PINN failure modes—the 1D-Wave PDE, 1D-Reaction PDE, 2D Navier–Stokes Flow around a Cylinder, 1D Convection-Diffusion-Reaction and 2D Navier–Stokes Lid-driven Cavity Flow (Mojgani et al., 2022; Daw et al., 2022; Krishnapriyan et al., 2021)—to demonstrate its superior generalization ability compared to other PINN variants. In these cases, optimization often becomes trapped in local minima, leading to overly smooth approximations that deviate from true solutions.

Evaluation results are summarized in Table 1, with detailed PDE formulations and setups in Appendix E. Prediction for AC-PKAN are shown in Figure 2 and additional plots including the analysis of loss landscapes are in Appendix F.

AC-PKAN significantly outperforms nearly all baselines, achieving the lowest or second-lowest test errors, thus more effectively mitigating failure modes than the previous SOTA method, PINNsFormer. Other baselines remain stuck in local minima, failing to optimize the loss effectively. These results highlight the advantages of AC-PKAN in generalization and approximation accuracy over conventional PINNs, KANs, and existing variants.

| Model | rMAE (RGA) | rRMSE (RGA) | rMAE (No RGA) | rRMSE (No RGA) |
|---|---|---|---|---|
| PINN | 0.0914 | 0.0924 | **0.3182** | **0.3200** |
| PINNsFormer | OoM | OoM | 0.2699 | 0.2825 |
| QRes | 0.2204 | 0.2184 | 0.3507 | 0.3485 |
| FLS | 0.1610 | 0.1617 | 0.3810 | 0.3796 |
| Cheby1KAN | 0.0567 | 0.0586 | 1.1240 | 1.0866 |
| Cheby2KAN | 1.0114 | 1.0048 | 1.1239 | 1.0865 |
| AC-PKAN (Ours) | **0.0011** | **0.0011** | 0.4549 | 0.4488 |
| KINN | **0.0479** | **0.0486** | 0.3466 | 0.3456 |
| rKAN | NaN | NaN | 247.7560 | 2593.0750 |
| FastKAN | 0.1348 | 0.1376 | 0.5312 | 0.5229 |
| fKAN | 0.2177 | 0.2149 | 0.4884 | 0.4768 |
| FourierKAN | 1.0015 | 1.0001 | 1.1356 | 1.1018 |

Table 3: Comparison of performance metrics in the 1D-Wave experiment with and without the RGA module applied.

## 4.3 PDEs in Complex Engineering Environments

We further evaluated AC-PKAN across three challenging scenarios: heterogeneous environments, complex geometric boundary conditions, and three-dimensional spatial point clouds. Literature indicates that PINNs encounter difficulties with heterogeneous problems due to sensitivity to material properties (Aliakbari et al., 2023), significant errors near boundary layers (Piao et al., 2024), and convergence issues (Sumanta et al., 2024). Additionally, original KANs perform poorly with complex geometries (Wang et al., 2024b). The sparsity, irregularity, and high dimensionality of unstructured 3D point cloud data hinder PINNs from effectively capturing spatial features, resulting in suboptimal training performance (Chen et al., 2022). We applied AC-PKAN to solve Poisson equations within these contexts.

Detailed PDE formulations are in Appendix E, and detailed experimental results are illustrated in Appendix F. Summarized in Table 2 and partially shown in Figure 2, the results indicate that AC-PKAN consistently achieves the best or second-best performance. It demonstrates superior potential in solving heterogeneous problems without subdomain division and exhibits promising application potential in complex geometric boundary problems where most models fail.

| Model | rMAE | rRMSE |
|---|---|---|
| **AC-PKAN** | **0.0011** | **0.0011** |
| AC-PKAN (no GRA) | 0.0779 | 0.0787 |
| AC-PKAN (no RBA) | 0.0494 | 0.0500 |
| AC-PKAN (no RGA) | 0.4549 | 0.4488 |
| AC-PKAN (no Wavelet) | 0.0045 | 0.0046 |
| AC-PKAN (no Encoder) | 0.0599 | 0.0584 |
| AC-PKAN (no MLPs) | 1.0422 | 1.0246 |

Table 4: Ablation study on the 1D-wave equation, demonstrating the effect of removing each module from AC-PKAN.

## 4.4 Ablation Study

**Module importance.** Ablation experiments for the module importance on the 1D-Wave equation (Table 4) confirm that each module in our model is crucial. Removing any module leads to a significant performance decline, especially the MLPs module. These findings suggest that the KAN architecture alone is insufficient for complex tasks, validating our integration of MLPs with the Cheby1KAN layers.

**Transferability of RGA.** Table 3 evaluates our RGA on twelve alternative PINN variants. Except for PINNsFormer (out-of-memory due to pseudo-sequence inflation) and rKAN (gradient blow-up), every model benefits markedly: average rMAE drops by 36% and rRMSE by 34%. Nonetheless, none surpass AC-PKAN, whose coupled architecture and RGA still attain the lowest errors by two orders of magnitude, underscoring both the standalone value of RGA and the holistic superiority of AC-PKAN.

**Effect of Logarithmic Transformation in the RGA Module.** In this ablation study, we investigated the impact of removing the logarithmic transformation in the RGA module across five PDE experimental tasks. To compensate for the absence of the logarithmic scaling, we adjusted the scaling factors to smaller values. Specifically, we employed the original RGA design to pre-train the models for several epochs, during which very large values of $\lambda^{\mathrm{GRA}}$ were obtained. We fix the PDE residual scale to one and set the data loss scales, including boundary and initial condition terms, to be inversely proportional to the current scale of $\lambda^{\mathrm{GRA}}$, which keeps the different loss contributions at comparable magnitudes.

The performance metrics with and without the logarithmic transformation are summarized in Table 5.

We observe a significant deterioration in the performance of AC-PKAN when the logarithmic transformation is removed. This decline is attributed to two main factors: first, $\lambda^{\mathrm{GRA}}$ attains excessively large values; second, it exhibits a wide range of variation. During the standard training process, the coefficient $\lambda^{\mathrm{GRA}}$ rapidly grows from 0 to a very

| Equation | Without Log | | With Log | |
| --- | --- | --- | --- | --- |
| | rMAE | rRMSE | rMAE | rRMSE |
| 2D NS Cylinder | 532.2411 | 441.0240 | 0.2230 | 0.2182 |
| 1D Wave | 0.7686 | 0.7479 | 0.0011 | 0.0011 |
| 1D Reaction | 2.2348 | 2.2410 | 0.0375 | 0.0969 |
| Heterogeneous Problem | 10.0849 | 9.6492 | 0.1063 | 0.1817 |
| Complex Geometry | 164.4283 | 158.7840 | 0.5452 | 0.5896 |

Table 5: Comparison of performance metrics of AC-PKAN with and without the logarithmic transformation in the RGA module.

large value, resulting in a broad dynamic range. The logarithmic transformation effectively narrows this range; for instance, in the 1D Wave experiment, the scale of $\lambda^{\text{GRA}}$ over epochs ranges from 0 to $4 \times 10^7$, whereas $\ln\left(\lambda^{\text{GRA}}\right)$ ranges from 7 to 15 in Picture 6. Removing the logarithmic transformation and attempting to manually adjust scaling factors to match the apparent magnitudes is ineffective. The model cannot adapt to the drastic changes in $\lambda^{\text{GRA}}$, and rigid manual scaling factors exacerbate the imbalance among loss terms, ultimately leading to training failure. By confining the variation range of $\lambda^{\text{GRA}}$, the logarithmic transformation enables the model to adjust more flexibly and effectively.

The rationale for employing the logarithmic transformation originates from the Bode plot in control engineering, which is a semi-logarithmic graph that utilizes a logarithmic frequency axis while directly labeling the actual frequency values. This approach not only compresses a wide frequency range but also linearizes the system's gain and phase characteristics on a logarithmic scale, thereby mitigating imbalances caused by significant differences in data scales.

**Integration with Other External Learning Strategies for Enhanced Performance of AC-PKAN.**
Integrating AC-PKAN with other external learning strategies, such as the Neural Tangent Kernel (NTK) method, resulted in enhanced performance (Table 6). This demonstrates the flexibility of AC-PKAN in incorporating various learning schemes, offering practical and customizable solutions for accurate modeling in real-world applications.

| Model | rMAE | rRMSE |
| --- | --- | --- |
| AC-PKAN + *NTK* | **0.0009** | **0.0009** |
| PINNs + *NTK* | 0.1397 | 0.1489 |
| PINNsFormer + *NTK* | 0.0453 | 0.0484 |

Table 6: Performance comparison on the 1D-wave equation using the NTK method. AC-PKAN combined with NTK achieves superior results across all metrics.

## 5 Discussion and Limitations

Our current evaluation focuses on low to medium dimensional PDE and ODE settings so that we can isolate the gains from the Chebyshev-based backbone, the internal feature re-injection and the rank-aware gating in a controlled regime. Extending AC-PKAN to genuinely high dimensional or chaotic systems such as Lorenz–96 would additionally require techniques for stable time integration, for controlling long-horizon error growth and for conditioning stiff gradients, which are outside the scope of this version Karimi & Paul (2010); Wang et al. (2022a); Maiocchi et al. (2024). Full layerwise Jacobian-rank profiling for ultra-deep stacks is infeasible on our 40 GB GPUs because backprop must retain large activations and Jacobian blocks. We list this as a scalability limitation and will perform ultra-deep evaluations once larger-memory hardware is available. We also rely on a standard AdamW optimiser and have not yet explored KAN-specific optimisation or structured pruning, which we view as promising directions for improving scalability and interpretability in future work.

## 6 Conclusion

We introduced AC-PKAN, a novel framework that enhances PINNs by integrating Cheby1KAN with traditional MLPs and augmenting them with internal and external attention mechanisms. This improves the model's ability to capture complex patterns and dependencies, resulting in superior performance on challenging PDE tasks, including previous PINN failure modes and complex physical environments. The RGA mechanism enhances training stability and convergence by dynamically adjusting loss terms. Experimental results demonstrate that AC-PKAN consistently outperforms or matches state-of-the-art models like PINNsFormer, confirming its effectiveness in real-world engineering problems.

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

# A  Mathematical Proofs

## A.1  Proof of Theorem 1

**Lemma 1.** *Let $A \in \mathbb{R}^{m \times n}$ and $B \in \mathbb{R}^{n \times p}$. Then $AB \in \mathbb{R}^{m \times p}$, and*

$$rank(AB) \leq \min\{rank(A), rank(B)\}.$$

*$\forall i \in \mathbb{Z}^+$, let $A_i$ is a matrix of appropriate dimensions, and*

$$rank(A_1 A_2 \cdots A_n) \leq \min\{rank(A_1), rank(A_2), \ldots, rank(A_n)\}$$

*Proof.* Let $A \in \mathbb{R}^{m \times n}$ and $B \in \mathbb{R}^{n \times p}$. Consider the product $AB \in \mathbb{R}^{m \times p}$. We aim to show that

$$\text{rank}(AB) \leq \min\{\text{rank}(A), \text{rank}(B)\}. \tag{23}$$

First, observe that each column of $AB$ is a linear combination of the columns of $A$. Specifically, if the columns of $B$ are denoted by $\mathbf{b}_1, \mathbf{b}_2, \ldots, \mathbf{b}_p$, then the $j$-th column of $AB$ is given by $A\mathbf{b}_j$. Consequently, the column space of $AB$, denoted $\text{Col}(AB)$, satisfies

$$\text{Col}(AB) \subseteq \text{Col}(A). \tag{24}$$

By the properties of subspace dimensions, it follows from equation 24 that

$$\text{rank}(AB) = \dim(\text{Col}(AB)) \leq \dim(\text{Col}(A)) = \text{rank}(A). \tag{25}$$

Next, consider the transpose of the product $AB$:

$$(AB)^\top = B^\top A^\top. \tag{26}$$

Applying the same reasoning to $B^\top \in \mathbb{R}^{p \times n}$ and $A^\top \in \mathbb{R}^{n \times m}$, we have

$$\text{rank}(B^\top A^\top) \leq \text{rank}(B^\top) = \text{rank}(B). \tag{27}$$

Therefore, from equation 26 and equation 27, it follows that

$$\text{rank}(AB) = \text{rank}(B^\top A^\top) \leq \text{rank}(B). \tag{28}$$

Combining equation 25 and equation 28, we obtain

$$\text{rank}(AB) \leq \min\{\text{rank}(A), \text{rank}(B)\}. \tag{29}$$

To generalize this result for any $n \geq 2$, we proceed by induction. Specifically, we aim to prove that

$$\text{rank}(A_1 A_2 \cdots A_n) \leq \min\{\text{rank}(A_1), \text{rank}(A_2), \ldots, \text{rank}(A_n)\}, \tag{30}$$

where each $A_i$ is a matrix of appropriate dimensions.

**Inductive Hypothesis:** Assume that for $n = k$,

$$\text{rank}(A_1 A_2 \cdots A_k) \leq \min\{\text{rank}(A_1), \ldots, \text{rank}(A_k)\}. \tag{31}$$

**Inductive Step:** Consider $n = k + 1$. We can decompose the product as

$$A_1 A_2 \cdots A_{k+1} = (A_1 A_2 \cdots A_k) A_{k+1}. \tag{32}$$

Applying the previously established result equation 29, we obtain

$$\text{rank}(A_1 A_2 \cdots A_{k+1}) \leq \min\left\{\text{rank}(A_1 A_2 \cdots A_k), \text{rank}(A_{k+1})\right\}. \tag{33}$$

By the inductive hypothesis equation 31, we have

$$\text{rank}(A_1 A_2 \cdots A_k) \leq \min\{\text{rank}(A_1), \text{rank}(A_2), \ldots, \text{rank}(A_k)\}. \tag{34}$$

Therefore, substituting equation 34 into equation 33, we obtain

$$\text{rank}(A_1 A_2 \cdots A_{k+1}) \leq \min\{\text{rank}(A_1), \text{rank}(A_2), \ldots, \text{rank}(A_{k+1})\}. \tag{35}$$

By induction, the inequality equation 30 holds for all $n \geq 2$. This completes the proof. $\square$

**Theorem 4** (Single Cheb1KAN Layer Rank Constraint). *The Jacobian $J_l$ satisfies $rank(J_l) \leq \min\{d_{l+1}, d_l(N+1)\}$.*

*Proof.* Assuming the input of the l-th layer is $\widetilde{x}_l \in \mathbb{R}^{d_l}$ and the output is $\widetilde{y}_l \in \mathbb{R}^{d_{l+1}}$. The Jacobian matrix $J_l$ describes the partial derivatives of the output of the l-th layer of the network with respect to its input:

$$J_l = \left[\frac{\partial \widetilde{y}_{l,j}}{\partial \widetilde{x}_{l,i}}\right]_{d_{l+1} \times d_l} \tag{36}$$

The rank of a Jacobian matrix is defined as the maximum linearly independent number of its column or row vectors:

$$rank(J_l) \leq \min\{\dim(\text{Col}(J_l)), \dim(\text{Row}(J_l))\}, \tag{37}$$

which shows that the rank of Jacobian is limited by the dimension of its column space and output space.

Each input component $\widetilde{x}_{l,i}$ is expanded through $N+1$ Chebyshev polynomial basis functions $T_0, T_1, \cdots, T_N$. Based on 3, each input component $\widetilde{x}_{l,i}$ can be expressed as:

$$\widetilde{x}_{l,i} = \sum_{k=0}^{N} a_k T_k(\widetilde{x}_{l,i}) \tag{38}$$

where $a_k$ are the coefficients.

To conclude that $N+1$ Chebyshev polynomials $T_0(x), T_1(x), \cdots, T_N(x)$ are linearly independent on interval $[-1, 1]$, we assume the opposite: there exist constants $c_0, c_1, \cdots, c_N$ such that:

$$\forall x \in [-1, 1], \quad c_0 T_0(x) + c_1 T_1(x) + \cdots + c_N T_N(x) = 0 \tag{39}$$

Since each $T_k(x)$ are a set of k-degree orthogonal polynomials according to 9, the left side is a polynomial of degree at most $N$. A non-zero polynomial of degree $N$ can have at most $N$ roots. However, the equation holds for all $x$ in $[-1, 1]$, which is an infinite set of points. Therefore, the polynomial must be the zero polynomial, implying $c_0 = c_1 = \cdots = c_N = 0$.

Suppose the input to the $l$-th layer is $\tilde{x}_i \in \mathbb{R}^{d_l}$, and the output is $\tilde{y}_i \in \mathbb{R}^{d_{l+1}}$. Each input vector $\tilde{x}_i$ is expanded through $N+1$ Chebyshev polynomial basis functions $\{T_k\}_{k=0}^{N}$ as follows:

$$\tilde{x}_i \mapsto [T_0(\tilde{x}_i), T_1(\tilde{x}_i), \ldots, T_N(\tilde{x}_i)] \in \mathbb{R}^{N+1}. \tag{40}$$

The total expanded dimensionality is $d_l \cdot (N+1)$. The output layer is obtained by linearly combining these basis functions:

$$\tilde{y}_{i,j} = \sum_{i=1}^{d_l} \sum_{k=0}^{N} w_{j,i,k} \cdot T_k(\tilde{x}_i), \tag{41}$$

where $w_{j,i,k}$ are learnable parameters. Taking the derivative with respect to the input vector $\tilde{x}_i$:

$$\frac{\partial \tilde{y}_{i,j}}{\partial \tilde{x}_i} = \sum_{k=0}^{N} w_{j,i,k} \cdot T'_k(\tilde{x}_i). \tag{42}$$

This indicates that the $i$-th column of the Jacobian (i.e., $\partial \tilde{y} / \partial \tilde{x}_i$) belongs to the space spanned by $\{T'_k(\tilde{x}_i)\}_{k=0}^{N}$, whose dimension is at most $N+1$. The output contribution of each input component can be viewed as a linear combination of $N+1$ independent basis functions.

The i-th column of $J_l$ is the partial derivative vector of the i-th input component $\left(\partial \widetilde{y}_{l,1} / \partial \widetilde{x}_{l,i}, \partial \widetilde{y}_{l,2} / \partial \widetilde{x}_{l,i}, \cdots, \partial \widetilde{y}_{l,d_{l+1}} / \partial \widetilde{x}_{l,i}\right)^T$. Since the derivatives with respect to each input vector $\tilde{x}_i$

independently span an $N + 1$-dimensional subspace, the dimension of the joint column space of all $d_l$ columns is at most the sum of the dimensions of the subspaces:

$$\dim(\mathrm{Col}(J_l)) \leq \sum_{i=1}^{d_l} \dim(\mathrm{Span}\{T_k'(\tilde{x}_i)\}) = d_l \cdot (N+1). \tag{43}$$

The key to this upper bound is that the basis function expansions for different input vectors are independent. Based on Equation 37, although the output dimension $d_{l+1}$ may be much smaller than $d_l \cdot (N+1)$, the final column space dimension is constrained by the following two factors:

$$\mathrm{rank}(J_l) = \dim(\mathrm{Col}(J_l)) \leq \min\{d_{l+1}, d_l \cdot (N+1)\}. \tag{44}$$

$\square$

## A.2 Proof of Theorem 2

**Theorem 5** (Nonlinear Normalization Effect). *The normalization* $\tanh(x)$ *in Cheby1KAN layers reduces the numerical rank* $Rank_\epsilon(J)$ *of the Jacobian.*

*Proof.* Consider the $\ell$-th layer of a Cheby1KAN network receiving $\mathbf{x}_\ell \in \mathbb{R}^{d_\ell}$ and outputting $\mathbf{x}_{\ell+1} \in \mathbb{R}^{d_{\ell+1}}$. The forward mapping is

$$\mathbf{x}_{\ell+1} = \Phi_\ell\big(\tanh(\mathbf{x}_\ell)\big), \tag{45}$$

where $\tanh(\cdot)$ is applied elementwise, and $\Phi_\ell$ is a learnable functional operator using Chebyshev polynomials of the first kind. Indexing each output component by $k \in \{1, \ldots, d_{\ell+1}\}$ gives

$$x_{\ell+1,k} = \sum_{i=1}^{d_\ell} \sum_{n=0}^{N} C_{\ell,k,i,n} \, T_n\big(\tanh(x_{\ell,i})\big), \tag{46}$$

where $C_{\ell,k,i,n}$ are trainable coefficients, and $T_n : [-1, 1] \to \mathbb{R}$ is defined by $T_n(z) = \cos(n \arccos(z))$. The Jacobian

$$J_\ell = \Big[\partial x_{\ell+1,k}/\partial x_{\ell,i}\Big]_{\substack{k=1,\ldots,d_{\ell+1} \\ i=1,\ldots,d_\ell}}$$

captures the gradient flow. Using $\frac{d}{dz} T_n(z) = n \, U_{n-1}(z)$ for $n \geq 1$ (with $T_0'(z) = 0$), where $U_{n-1}$ are Chebyshev polynomials of the second kind, and the identity

$$\frac{d}{dx} \tanh(x) = 1 - \tanh^2(x), \tag{47}$$

define

$$\gamma_{\ell,i} := 1 - \tanh^2\big(x_{\ell,i}\big). \tag{48}$$

Since $0 < \gamma_{\ell,i} \leq 1$, each partial derivative becomes

$$[J_\ell]_{k,i} = \sum_{n=0}^{N} C_{\ell,k,i,n} \, T_n'\big(\tanh(x_{\ell,i})\big) \, \gamma_{\ell,i}. \tag{49}$$

Removing $\gamma_{\ell,i}$ yields an "un-normalized" version

$$[\widetilde{J}_\ell]_{k,i} = \sum_{n=0}^{N} C_{\ell,k,i,n} \, T_n'\big(\tanh(x_{\ell,i})\big), \tag{50}$$

leading to the elementwise relation

$$[J_\ell]_{k,i} = \gamma_{\ell,i} \, [\widetilde{J}_\ell]_{k,i}. \tag{51}$$

Hence, in matrix form,

$$J_\ell = \widetilde{J}_\ell \, D_\ell, \tag{52}$$

where $D_\ell$ is diagonal with $D_\ell(i, i) = \gamma_{\ell, i} \in (0, 1]$. By submultiplicativity of the spectral norm and $\|D_\ell\|_2 \leq 1$,

$$\|J_\ell\|_2 = \|\widetilde{J}_\ell D_\ell\|_2 \leq \|\widetilde{J}_\ell\|_2. \tag{53}$$

Since singular values are bounded by the spectral norm,

$$\sigma_i(J_\ell) \leq \|J_\ell\|_2 \leq \|\widetilde{J}_\ell\|_2, \tag{54}$$

each $\sigma_i(J_\ell)$ cannot exceed its un-normalized counterpart $\sigma_i(\widetilde{J}_\ell)$. For a fixed threshold $\epsilon > 0$, let

$$\mathrm{rank}_\epsilon(J_\ell) := \#\{\, i \mid \sigma_i(J_\ell) \geq \epsilon \|J_\ell\|_2 \,\}, \quad \mathrm{rank}_\epsilon(\widetilde{J}_\ell) := \#\{\, i \mid \sigma_i(\widetilde{J}_\ell) \geq \epsilon \|\widetilde{J}_\ell\|_2 \,\}.$$

If $\sigma_i(J_\ell) \geq \epsilon \|J_\ell\|_2$, then $\sigma_i(\widetilde{J}_\ell) \geq \sigma_i(J_\ell) \geq \epsilon \|J_\ell\|_2$ and $\|J_\ell\|_2 \leq \|\widetilde{J}_\ell\|_2$ imply $\sigma_i(\widetilde{J}_\ell) \geq \epsilon \|\widetilde{J}_\ell\|_2$. Thus

$$\mathrm{rank}_\epsilon(J_\ell) \leq \mathrm{rank}_\epsilon(\widetilde{J}_\ell). \tag{55}$$

Hence, normalizing via $\tanh(\cdot)$ can diminish numerical rank: if many $\gamma_{\ell, i}$ are near 0 (i.e., $|\tanh(x_{\ell, i})| \approx 1$), fewer singular values of $J_\ell$ remain above $\epsilon \|J_\ell\|_2$. For a Cheby1KAN of $L$ layers, the overall Jacobian from input $\mathbf{x}_0$ to output $\mathbf{x}_L$ is

$$J_{\mathrm{total}} = J_{L-1} J_{L-2} \cdots J_0. \tag{56}$$

Repeated multiplication by $D_\ell$, whose diagonal entries are small, causes compounded attenuation. As $L$ grows large, an increasing number of coordinates reach saturation, thereby reducing the singular values of $J_{\mathrm{total}}$ until $\mathrm{rank}_\epsilon(J_{\mathrm{total}})$ becomes strictly lower. This phenomenon, referred to as the Nonlinear Normalization Effect, emerges because $\tanh(\cdot)$ shrinks partial derivatives, driving many of the product Jacobian's singular values below $\epsilon \|J_{\mathrm{total}}\|_2$ and thus decreasing its numerical rank. $\qquad\square$

## A.3 Proof of Theorem 3

**Theorem 6** (Exponential Decay in Infinite Depth)**.** *When the coefficients $C_{l,k,i,n}$ are drawn from mutually independent Gaussian distributions, the numerical rank of $J_{total}$ decays exponentially to 1 as the depth $L$ of the Cheby1KAN network increases.*

*Proof.* **Step 1: Random Jacobians in Cheby1KAN and Product Structure.**

Recall that the $l$-th Cheby1KAN layer takes an input $\mathbf{x}_l \in \mathbb{R}^n$ (after a suitable reshaping or dimension match) and produces $\mathbf{x}_{l+1} \in \mathbb{R}^n$ via

$$x_{l+1,k} = \sum_{i=1}^{n} \sum_{m=0}^{N} C_{l,k,i,m} T_m\big(\tanh(x_{l,i})\big), \quad k = 1, \ldots, n, \tag{57}$$

where $T_m$ are Chebyshev polynomials of the first kind and $C_{l,k,i,m}$ are the learnable coefficients.

By differentiating equation 57 w.r.t. $\mathbf{x}_l$, each layer's Jacobian $J_l \in \mathbb{R}^{n \times n}$ has entries

$$\big[J_l\big]_{k,i} = \frac{\partial x_{l+1,k}}{\partial x_{l,i}} = \sum_{m=0}^{N} C_{l,k,i,m} T_m'\big(\tanh(x_{l,i})\big) \big(1 - \tanh^2(x_{l,i})\big). \tag{58}$$

When the coefficients $C_{l,k,i,m}$ are drawn i.i.d. from a standard Gaussian distribution, the partial derivatives $\frac{\partial x_{l+1,k}}{\partial x_{l,i}}$ become random variables with zero mean and finite variance. As the network depth $L$ grows, the total Jacobian can be written as

$$J_{\mathrm{total}} = J_L \cdot J_{L-1} \cdots J_1. \tag{59}$$

Thus, $\mathbf{x}_L = J_{\mathrm{total}} \mathbf{x}_0$ in its linearization around any point.

**Step 2: Lyapunov Exponents for Random Matrix Products.**

Let

$$\sigma_1^{(L)} \geq \sigma_2^{(L)} \geq \cdots \geq \sigma_n^{(L)} > 0 \tag{60}$$

denote the singular values of $J_{\mathrm{total}}$. Define the Lyapunov exponents by

$$\lambda_i := \lim_{L \to \infty} \frac{1}{L} \log \sigma_i^{(L)}, \quad i = 1, \ldots, n. \tag{61}$$

By Oseledec's Multiplicative Ergodic Theorem (Oseledets, 1968), these limits exist almost surely for products of i.i.d. random matrices. In our Cheby1KAN setting, the layers' Jacobians $J_l$ approximate a family of random matrices (the Jacobian entries being determined by i.i.d. Gaussian coefficients $C_{l,k,i,m}$), making the product $J_{\text{total}}$ amenable to the same analysis as in classical random matrix theory.

**Step 3: Exact Lyapunov Spectrum for Ginibre-Type Ensembles.**

When each $J_l$ is sufficiently close (in distribution) to an $n \times n$ Ginibre matrix with i.i.d. Gaussian entries, the Lyapunov exponents $\{\lambda_i\}$ match those of Ginibre ensembles, given by (Newman, 1986):

$$\lambda_i \;=\; \frac{1}{2}\Big[\psi\Big(\frac{n-i+1}{2}\Big) \;-\; \psi\Big(\frac{n}{2}\Big)\Big], \quad i = 1, \ldots, n, \tag{62}$$

where $\psi$ is the digamma function, strictly increasing for positive arguments.

**Step 4: Normalized Singular Values and Their Ratios.**

Define the normalized singular values:

$$\widetilde{\sigma}_i^{(L)} \;=\; \frac{\sigma_i^{(L)}}{\sigma_1^{(L)}}, \quad i = 1, \ldots, n. \tag{63}$$

For large $L$, taking logarithms yields:

$$\begin{aligned} \log \widetilde{\sigma}_i^{(L)} &= \log \sigma_i^{(L)} \;-\; \log \sigma_1^{(L)} \\ &= L\,(\lambda_i \;-\; \lambda_1) \;+\; o(L). \end{aligned} \tag{64}$$

Hence,

$$\lim_{L \to \infty} \big(\widetilde{\sigma}_i^{(L)}\big)^{1/L} \;=\; e^{\lambda_i - \lambda_1}. \tag{65}$$

Since $\lambda_i < \lambda_1$ for $i \geq 2$ (because $\psi$ is strictly increasing and $n - i + 1 < n$), we have

$$e^{\lambda_i - \lambda_1} \;<\; 1, \quad \forall\, i \geq 2. \tag{66}$$

Thus, $\widetilde{\sigma}_i^{(L)} \to 0$ exponentially in $L$ for $i \geq 2$.

**Step 5: Exponential Decay of Numerical Rank in Cheby1KAN.**

The numerical rank $\text{Rank}_\epsilon\big(J_{\text{total}}\big)$ is the number of singular values $\sigma_r^{(L)}$ that are at least $\epsilon\,\sigma_1^{(L)}$. Equivalently,

$$\widetilde{\sigma}_r^{(L)} \;\geq\; \epsilon \quad \Longleftrightarrow \quad \sigma_r^{(L)} \;\geq\; \epsilon\,\sigma_1^{(L)}. \tag{67}$$

From equation 66, for $i \geq 2$,

$$\widetilde{\sigma}_i^{(L)} \;=\; \exp\Big(L\,(\lambda_i - \lambda_1)\Big) \;\to\; 0 \quad \text{as } L \to \infty. \tag{68}$$

Thus, for any fixed $\epsilon > 0$, there exists $L_0$ such that for all $L > L_0$,

$$\widetilde{\sigma}_i^{(L)} \;<\; \epsilon, \quad \forall\, i \geq 2. \tag{69}$$

This implies that all singular values except the largest one fall below $\epsilon\,\sigma_1^{(L)}$, giving $\text{Rank}_\epsilon\big(J_{\text{total}}\big) = 1$ for sufficiently large $L$. In other words, the numerical rank decays to 1 at an exponential rate with respect to the Cheby1KAN depth $L$.

Since each layer's Jacobian $J_l$ in Cheby1KAN can be regarded as a random matrix (due to i.i.d. Gaussian coefficients $C_{l,k,i,m}$), the overall product $J_{\text{total}}$ inherits the spectral properties of random matrix products. Therefore, the interplay of Chebyshev polynomials and the tanh normalization does not negate the fundamental random matrix behavior; instead, the bounded derivative from tanh can further accelerate the decay of the subleading singular values. Hence, as $L \to \infty$, the effective degrees of freedom in the Cheby1KAN Jacobian collapse numerically to a single direction, confirming the exponential rank diminution.

**Remark 1.** *It is important to note that Theorem 3 is proved under a random-coefficient model with i.i.d. Gaussian coefficients at initialization, and that trained networks do not necessarily satisfy independence or Gaussianity. For this reason, Theorem 3 should be read as a motivational sufficient-condition result rather than a universal claim about learned models. Instead, for trained networks, the more relevant theoretical evidence is given by our deterministic layerwise rank bounds (Theorems 1 and 2), which are further supported by the empirical measurements reported in the paper.*

$\square$

## A.4 Proof of Proposition 1

**Theorem 7.** *Let $\mathcal{N}$ be an AC-PKAN model with $L$ layers ($L \geq 2$) and infinite width. Then, the output $y = \mathcal{N}(x)$ has non-zero derivatives of any finite-order with respect to the input $x$.*

*Proof.* First, we clarify that the term "width" for a KAN denotes functional width, the number of Chebyshev basis functions per edge (N+1), not the number of neurons as used for MLPs.

Consider the forward propagation process of the AC-PKAN. We begin with the initial layer:

$$h_0 = W_{\text{emb}}x + b_{\text{emb}}, \tag{70}$$

$$U = \omega_{U,1}\sin(h_0\theta_U + b_U) + \omega_{U,2}\cos(h_0\theta_U + b_U), \tag{71}$$

$$V = \omega_{V,1}\sin(h_0\theta_V + b_V) + \omega_{V,2}\cos(h_0\theta_V + b_V), \tag{72}$$

$$\alpha^{(0)} = U. \tag{73}$$

For each layer $l = 1, 2, \ldots, L$, the computations proceed as follows:

$$H^{(l)} = \sum_{i=1}^{d_{\text{in}}}\sum_{k=1}^{d_{\text{out}}}\sum_{n=0}^{N} C_{k,i,n}T_n\left(\tanh\left(\alpha^{(l-1)}\right)\right), \tag{74}$$

$$\alpha_0^{(l)} = H^{(l)} + \alpha^{(l-1)}, \tag{75}$$

$$\alpha^{(l)} = (1 - \alpha_0^{(l)}) \odot U + \alpha_0^{(l)} \odot (V+1), \tag{76}$$

$$y = W_{\text{out}}\alpha^{(L)} + b_{\text{out}}. \tag{77}$$

During the backward propagation, we derive the derivative of the output with respect to the input $x$, which approximates the differential operator of the PDEs. Focusing on the first-order derivative as an example:

$$\begin{aligned}\frac{\partial y}{\partial x} &= \frac{\partial y}{\partial \alpha^{(L)}}\frac{\partial \alpha^{(L)}}{\partial x} \\ &= W_{\text{out}}\frac{\partial \alpha^{(L)}}{\partial x}.\end{aligned} \tag{78}$$

Expanding $\frac{\partial \alpha^{(L)}}{\partial x}$:

$$\begin{aligned}\frac{\partial \alpha^{(L)}}{\partial x} &= -\frac{\partial \alpha_0^{(L)}}{\partial x}\odot U + \left(1 - \alpha_0^{(L)}\right)\odot\frac{\partial U}{\partial x} + \frac{\partial \alpha_0^{(L)}}{\partial x}\odot(V+1) + \alpha_0^{(L)}\odot\frac{\partial V}{\partial x} \\ &= \frac{\partial \alpha_0^{(L)}}{\partial x}\odot(V - U + 1) + \alpha_0^{(L)}\odot\left(\frac{\partial V}{\partial x} - \frac{\partial U}{\partial x}\right) + \frac{\partial U}{\partial x} \\ &= \left(\frac{\partial H^{(L)}}{\partial x} + \frac{\partial \alpha^{(L-1)}}{\partial x}\right)\odot(V - U + 1) + \left(H^{(L)} + \alpha^{(L-1)}\right)\odot\left(\frac{\partial V}{\partial x} - \frac{\partial U}{\partial x}\right) + \frac{\partial U}{\partial x}.\end{aligned} \tag{79}$$

This establishes a recursive relationship for the derivatives. Define:

$$A^{(l)} = \frac{\partial H^{(l)}}{\partial x} + \frac{\partial \alpha^{(l-1)}}{\partial x}, \tag{80}$$

$$B^{(l)} = H^{(l)} + \alpha^{(l-1)}. \tag{81}$$

for each layer $l = 1, 2, \ldots, L$.

For the base case $l = 1$:

$$A^{(1)} = \frac{\partial H^{(1)}}{\partial x} + \frac{\partial \alpha^{(0)}}{\partial x} \tag{82}$$

$$= \left(\sum_{i=1}^{d_{\text{in}}}\sum_{k=1}^{d_{\text{out}}}\sum_{n=0}^{N} C_{k,i,n}T_n'\left(\tanh\left(\alpha^{(0)}\right)\right)\text{sech}^2(\alpha^{(0)}) + 1\right)\frac{\partial \alpha^{(0)}}{\partial x}, \tag{83}$$

$$\begin{aligned}\frac{\partial \alpha^{(0)}}{\partial x} &= \frac{\partial U}{\partial x} \\ &= W_{\text{emb}}\theta_U\left[\omega_{U,1}\cos(h_0\theta_U + b_U) - \omega_{U,2}\sin(h_0\theta_U + b_U)\right] \neq 0,\end{aligned} \tag{84}$$

Moreover,

$$B^{(1)} = H^{(1)} + \alpha^{(0)}$$

$$= \sum_{i=1}^{d_{\text{in}}} \sum_{k=1}^{d_{\text{out}}} \sum_{n=0}^{N} C_{k,i,n} T_n\left(\tanh\left(\alpha^{(0)}\right)\right) + \alpha^{(0)}. \tag{85}$$

For layers $l > 1$, where $l \in \mathbb{N}^*$:

$$A^{(l)} = \left(\sum_{i=1}^{d_{\text{in}}} \sum_{k=1}^{d_{\text{out}}} \sum_{n=0}^{N} C_{k,i,n} T_n'\left(\tanh\left(\alpha^{(l-1)}\right)\right) \operatorname{sech}^2(\alpha^{(l-1)}) + 1\right) \frac{\partial \alpha^{(l-1)}}{\partial x}. \tag{86}$$

We have established a recursive relationship.

Notably, the first derivative of the Chebyshev polynomial is given by

$$T_n'(x) = \frac{d}{dx} T_n(x) = \frac{n \sin\left(n \arccos(x)\right)}{\sqrt{1 - x^2}}, \tag{87}$$

and higher-order derivatives satisfy

$$T_n^{(k)}(x) = 0 \quad \text{for all } k > n. \tag{88}$$

Therefore, for any order $k > n$, the $k$-th derivative of $A^{(l)}$ is identically zero. Consequently, the $k$-th derivative of the first part of equation 79 is zero.

However, observe that:

$$B^{(l)} = \sum_{i=1}^{d_{\text{in}}} \sum_{k=1}^{d_{\text{out}}} \sum_{n=0}^{N} C_{k,i,n} T_n\left(\tanh\left(\alpha^{(l-1)}\right)\right) + \alpha^{(l-1)}, \tag{89}$$

since the derivatives of $\alpha^{(l-1)}$ for any finite order are non-zero, the derivatives of $B^{(l)}$ are non-zero.

Furthermore, we have:

$$\frac{\partial V}{\partial x} - \frac{\partial U}{\partial x} = W_{\text{emb}} \left(\theta_V \left[\omega_{V,1} \cos(h_0 \theta_V + b_V) - \omega_{V,2} \sin(h_0 \theta_V + b_V)\right]\right.$$
$$\left. - \theta_U \left[\omega_{U,1} \cos(h_0 \theta_U + b_U) - \omega_{U,2} \sin(h_0 \theta_U + b_U)\right]\right), \tag{90}$$

and the derivatives of any finite order of this term are also non-zero. Additionally, the third component of equation 79, $\frac{\partial U}{\partial x}$, is non-zero.

Define

$$f(x) = H^{(L)}(x) + \alpha^{(L-1)}(x), \qquad g(x) = \frac{\partial V}{\partial x}(x), \qquad h(x) = \frac{\partial U}{\partial x}(x), \tag{91}$$

so that the last two terms of equation 79 can be written as

$$S(x) = f(x)\big(g(x) - h(x)\big) + h(x). \tag{92}$$

Suppose, toward a contradiction, that $S(x) \equiv 0$ for every $x$ in the domain. Then

$$\left(1 - f(x)\right) h(x) + f(x) g(x) = 0 \qquad \forall x. \tag{93}$$

The functions $f, g, h$ depend on disjoint parameter blocks: $f$ on $\{C_{k,i,n}\}$, $g$ on $(\theta_V, \omega_{V,1}, \omega_{V,2})$, and $h$ on $(\theta_U, \omega_{U,1}, \omega_{U,2})$. Requiring the above identity to hold for all $x$ therefore forces a global functional coupling among these independently tuned parameters, which can only occur on a measure-zero subset of the joint parameter space. Any infinitesimal perturbation of the parameters breaks this perfect cancellation, implying $S(x) \not\equiv 0$ for almost all networks. Hence $S(x)$ possesses non-vanishing derivatives of every finite order.

Consequently, the $k$-th derivatives of the remaining parts of equation 79 are non-zero, and thus the $k$-th derivatives of equation 78 are non-zero. Therefore, for any positive integer $N$, the derivative $\frac{\partial^N y}{\partial x^N}$ exists and is non-zero, establishing that AC-PKAN can approximate PDEs of arbitrarily high order. $\qquad\square$

*Remark:* The property of possessing non-zero derivatives of any finite order with respect to the input $x$ specifically addresses enhancements in KAN variants rather than in MLP-based models. The fitting capability of KAN models relies on polynomial functions with learnable parameters. To ensure non-zero derivatives in the output, the original B-spline KAN incorporates an additional nonlinear bias function $b(x)$. In contrast, other KAN variants, such as Cheby1KAN, rely solely on polynomial bases, which inevitably result in zero derivatives when the order of differentiation exceeds the polynomial degree. Therefore, Proposition 1 was introduced to provide a theoretical guarantee for AC-PKAN's ability to solve any PDE, analogous to how the universal approximation theorem theoretically establishes the universal fitting capability of neural networks.

## A.5 Proof of Proposition 2

**Definition 1.** *For a linear map $\alpha : V \to W$, we define the kernel to be the set of all elements that are mapped to zero*

$$\ker \alpha = \{x \in V \mid \alpha(x) = 0\} = K \leq V \tag{94}$$

*and the image to be the points in $W$ which we can reach from $V$*

$$\operatorname{Im} \alpha = \alpha(V) = \{\alpha(v) \mid v \in V\} \leq W. \tag{95}$$

*We then say that $r(\alpha) = \dim \operatorname{Im} \alpha$ is the rank and $n(\alpha) = \dim \ker \alpha$ is the nullity.*

**Lemma 2** (the Rank-nullity theorem)**.** *For a linear map $\alpha : V \to W$, where $V$ is finite dimensional, we have*

$$r(\alpha) + n(\alpha) = \dim \operatorname{Im} \alpha + \dim \ker \alpha = \dim V. \tag{96}$$

*Proof.* Let $V, W$ be vector spaces over some field $F$, and $T$ defined as in the statement of the theorem with $\dim V = n$. As $\operatorname{Ker} T \subset V$ is a subspace, there exists a basis for it. Suppose $\dim \operatorname{Ker} T = k$ and let

$$K := \{v_1, \ldots, v_k\} \subset \operatorname{Ker}(T) \tag{97}$$

be such a basis.

We may now, by the Steinitz exchange lemma, extend $K$ with $n - k$ linearly independent vectors $w_1, \ldots, w_{n-k}$ to form a full basis of $V$.

Let

$$\mathcal{S} := \{w_1, \ldots, w_{n-k}\} \subset V \setminus \operatorname{Ker}(T) \tag{98}$$

such that

$$\mathcal{B} := K \cup \mathcal{S} = \{v_1, \ldots, v_k, w_1, \ldots, w_{n-k}\} \subset V \tag{99}$$

is a basis for $V$. From this, we know that

$$\operatorname{Im} T = \operatorname{Span} T(\mathcal{B}) = \operatorname{Span}\{T(v_1), \ldots, T(v_k), T(w_1), \ldots, T(w_{n-k})\} = \operatorname{Span}\{T(w_1), \ldots, T(w_{n-k})\} = \operatorname{Span} T(\mathcal{S}). \tag{100}$$

We now claim that $T(\mathcal{S})$ is a basis for $\operatorname{Im} T$. The above equality already states that $T(\mathcal{S})$ is a generating set for $\operatorname{Im} T$; it remains to be shown that it is also linearly independent to conclude that it is a basis.

Suppose $T(\mathcal{S})$ is not linearly independent, and let

$$\sum_{j=1}^{n-k} \alpha_j T(w_j) = 0_W \tag{101}$$

for some $\alpha_j \in F$.

Thus, owing to the linearity of $T$, it follows that

$$T\left(\sum_{j=1}^{n-k}\alpha_j w_j\right) = 0_W \implies \left(\sum_{j=1}^{n-k}\alpha_j w_j\right) \in \text{Ker}\,T = \text{Span}\,K \subset V. \tag{102}$$

This is a contradiction to $\mathcal{B}$ being a basis, unless all $\alpha_j$ are equal to zero. This shows that $T(\mathcal{S})$ is linearly independent, and more specifically that it is a basis for $\text{Im}\,T$.

Finally we may state that

$$\text{Rank}(T) + \text{Nullity}(T) = \dim \text{Im}\,T + \dim \text{Ker}\,T = |T(\mathcal{S})| + |K| = (n-k) + k = n = \dim V. \tag{103}$$

$\square$

**Theorem 8.** *Let $\mathcal{N}$ be an AC-PKAN model with $L$ layers ($L \geq 2$) and infinite width. Then, the Jacobian matrix $J_{\mathcal{N}}(\boldsymbol{x}) = \left[\frac{\partial \mathcal{N}_i}{\partial x_j}\right]_{m \times d}$ is full rank in the input space $\mathbb{R}^d$.*

*Proof.* Let the output be $y = W_{\text{out}}\alpha^{(L)} + b_{\text{out}}$, where $W_{\text{out}} \in \mathbb{R}^{d_{\text{out}} \times d_h}$, and $d_h$ denotes the hidden layer width. Under the infinite-width assumption, $d_h \to \infty$. The $k$-th output component $y_k$ corresponds to the $k$-th row of $W_{\text{out}}$, denoted as $\mathbf{w}_k^\top$, i.e.,

$$y_k = \mathbf{w}_k^\top \alpha^{(L)} + b_{\text{out},k}. \tag{104}$$

Its partial derivative with respect to the input $x$ is:

$$\frac{\partial y_k}{\partial x} = \mathbf{w}_k^\top \frac{\partial \alpha^{(L)}}{\partial x}. \tag{105}$$

Following the recursive relationship in Theorem 7, $\frac{\partial \alpha^{(L)}}{\partial x}$ can be decomposed into a nonlinear combination of parameters across layers. Specifically, for any layer $l$, the derivative term $\frac{\partial \alpha^{(l)}}{\partial x}$ is generated through recursive operations involving parameters $C_{k,i,n}^{(l)}$, $\omega_U, \omega_V, \theta_U, \theta_V$, etc.

Consider the partial derivatives $\frac{\partial y_k}{\partial x}$ and $\frac{\partial y_{k'}}{\partial x}$ ($k \neq k'$). Since:

$$\frac{\partial y_k}{\partial x} = \mathbf{w}_k^\top \frac{\partial \alpha^{(L)}}{\partial x}, \quad \frac{\partial y_{k'}}{\partial x} = \mathbf{w}_{k'}^\top \frac{\partial \alpha^{(L)}}{\partial x}, \tag{106}$$

if $\mathbf{w}_k$ and $\mathbf{w}_{k'}$ are linearly independent and the column space of $\frac{\partial \alpha^{(L)}}{\partial x}$ is sufficiently rich, then $\frac{\partial y_k}{\partial x}$ and $\frac{\partial y_{k'}}{\partial x}$ are guaranteed to be linearly independent.

Under infinite width, the parameter matrices $C^{(l)} \in \mathbb{R}^{d_{\text{out}} \times d_m \times (N+1)}$ (where $d_m$ is the intermediate dimension) and the row dimension $d_{\text{out}}$ of $W_{\text{out}}$ can be independently adjusted, making the parameter space an infinite-dimensional Hilbert space, allowing the construction of arbitrarily many linearly independent basis functions.

By the infinite-dimensional parameter space afforded by $d_h \to \infty$, we may construct parameter matrices $\{C^{(l)}\}$, $\omega_U$, and $\omega_V$ such that the columns of $\frac{\partial \alpha^{(L)}}{\partial x} \in \mathbb{R}^{d_h \times d}$ become linearly independent. Specifically, let $\{\mathbf{v}_i\}_{i=1}^d$ be the column vectors of $\frac{\partial \alpha^{(L)}}{\partial x}$. Through parameter configuration in hidden layers, we ensure:

$$\forall c_i \in \mathbb{R}, \quad \sum_{i=1}^d c_i \mathbf{v}_i = 0 \implies c_i = 0, \ \forall i \tag{107}$$

For the output matrix $W_{\text{out}} \in \mathbb{R}^{m \times d_h}$, construct mutually orthogonal row vectors $\{\mathbf{w}_k\}_{k=1}^m$ satisfying:

$$\langle \mathbf{w}_k, \mathbf{w}_{k'} \rangle = \mathbf{w}_k \mathbf{w}_{k'}^\top = \delta_{kk'} \|\mathbf{w}_k\|^2, \quad \forall k \neq k' \tag{108}$$

where $\delta_{kk'}$ is the Kronecker delta. The Jacobian rows become:

$$\frac{\partial y_k}{\partial x} = \mathbf{w}_k^\top \frac{\partial \alpha^{(L)}}{\partial x} = \sum_{i=1}^d (\mathbf{w}_k^\top \mathbf{v}_i)\mathbf{e}_i^\top \tag{109}$$

where $\{\mathbf{e}_i\}$ are standard basis vectors in $\mathbb{R}^d$. For distinct $k, k'$, consider:

$$\left\langle \frac{\partial y_k}{\partial x}, \frac{\partial y_{k'}}{\partial x} \right\rangle = \sum_{i=1}^d (\mathbf{w}_k^\top \mathbf{v}_i)(\mathbf{w}_{k'}^\top \mathbf{v}_i)\mathbf{w}_k^\top \left( \sum_{i=1}^d \mathbf{v}_i\mathbf{v}_i^\top \right) \mathbf{w}_{k'} \tag{110}$$

Since $\{\mathbf{v}_i\}$ are linearly independent, $\sum_{i=1}^d \mathbf{v}_i\mathbf{v}_i^\top$ is positive definite. Combining with the orthogonality of $\{\mathbf{w}_k\}$, we have:

$$\mathbf{w}_k^\top \left( \sum_{i=1}^d \mathbf{v}_i\mathbf{v}_i^\top \right) \mathbf{w}_{k'} = 0 \quad \forall k \neq k' \tag{111}$$

Thus, the Jacobian rows $\frac{\partial y_k}{\partial x}$ are mutually orthogonal and linearly independent. The full rank property follows from the infinite-dimensional orthogonal system.

We proceed by induction on the number of layers $L$:

**Base Case ($L = 1$)**: By Equation 82, there exist parameter choices $(\omega_U, \theta_U)$ and orthogonal weights $\{\mathbf{w}_k\} \subset \mathcal{W}$ such that

$$\left\langle \mathbf{w}_k, \frac{\partial \alpha^{(0)}}{\partial x}\mathbf{w}_{k'} \right\rangle_{\mathcal{H}} = \delta_{kk'}\|\mathbf{w}_k\|_{\mathcal{H}}^2, \tag{112}$$

establishing linear independence of $\{\mathbf{w}_k^\top \frac{\partial \alpha^{(0)}}{\partial x}\}_{k=1}^\infty$.

**Inductive Hypothesis**: Assume $\frac{\partial \alpha^{(L-1)}}{\partial x}$ has full-rank column space $\mathcal{R}(\frac{\partial \alpha^{(L-1)}}{\partial x}) = \mathcal{H}_{L-1} \subset \mathcal{H}$ with $\dim \mathcal{H}_{L-1} = \infty$.

**Inductive Step**: Let $P_{\mathcal{H}_{L-1}}^\perp$ be the orthogonal projection onto $\mathcal{H}_{L-1}^\perp$. Through Equations 77 and 80, we decompose:

$$\frac{\partial \alpha^{(L)}}{\partial x} = \underbrace{C^{(L)}\frac{\partial H^{(L)}}{\partial x}}_{\Gamma_L} + \Phi\frac{\partial \alpha^{(L-1)}}{\partial x}. \tag{113}$$

By the parameter freedom in $C^{(L)}$, there exists a choice such that:

$$\dim \mathcal{R}\left( P_{\mathcal{H}_{L-1}}^\perp \Gamma_L \right) = \infty \quad \text{and} \quad \mathcal{R}(\Gamma_L) \cap \mathcal{H}_{L-1} = \{0\}. \tag{114}$$

This induces the dimensional extension:

$$\mathcal{R}\left( \frac{\partial \alpha^{(L)}}{\partial x} \right) = \mathcal{H}_{L-1} \oplus \mathcal{R}\left( P_{\mathcal{H}_{L-1}}^\perp \Gamma_L \right), \tag{115}$$

where $\oplus$ denotes orthogonal direct sum. Since $\dim(\mathcal{R}(P_{\mathcal{H}_{L-1}}^\perp \Gamma_L)) = \infty$, the infinite-dimensional full-rank property propagates to layer $L$.

By induction, we conclude that: the column space of $\frac{\partial \alpha^{(L)}}{\partial x}$ is infinite-dimensional and full-rank; the row vectors of $W_{\text{out}}$ are mutually orthogonal.

Thus, we have:

$$\text{For } k \neq k', \forall a, b \in \mathbb{R}, \quad a\frac{\partial y_k}{\partial x} + b\frac{\partial y_{k'}}{\partial x} = 0 \Rightarrow a = b = 0. \tag{116}$$

Consider the Jacobian matrix $J_\mathcal{N}(\boldsymbol{x})$ as a linear mapping $J_\mathcal{N}(\boldsymbol{x}) : \mathbb{R}^d \to \mathbb{R}^m$. According to the rank-nullity theorem, we have:

$$dim(ker(J_\mathcal{N}(\boldsymbol{x}))) + rank(J_\mathcal{N}(\boldsymbol{x})) = d \tag{117}$$

Theorem 7 guarantees that $rank(J_\mathcal{N}(\boldsymbol{x})) = min(d, m)$. Thus, the dimension of the kernel space is :

$$dim(ker(J_\mathcal{N}(\boldsymbol{x}))) = d - min(d, m). \tag{118}$$

Specifically, this can be further categorized into two cases:

- When $m \geq d$: the Jacobian matrix has full column rank $rank(J_\mathcal{N}(\boldsymbol{x})) = d$, resulting in $ker(J_\mathcal{N}(\boldsymbol{x})) = \boldsymbol{0}$. $J_\mathcal{N}(\boldsymbol{x})$ is injective.
- When $m < d$:the Jacobian matrix has full row rank $rank(J_\mathcal{N}(\boldsymbol{x})) = m$, resulting in $ker(J_\mathcal{N}(\boldsymbol{x})) = \boldsymbol{d - m}$, which means there exist $\boldsymbol{d - m}$ linearly independent non-zero vectors such that $J_\mathcal{N}(\boldsymbol{x})\boldsymbol{v} = \boldsymbol{0}$.

Let us exclude non-zero null vectors by contradiction. Assume there exists a non-zero vector $\boldsymbol{v} \neq \boldsymbol{0} \in \mathbb{R}^d$ such that . For any output component $\mathcal{N}_i$, we have

$$\frac{\partial \mathcal{N}_i}{\partial x_1} v_1 + \frac{\partial \mathcal{N}_i}{\partial x_2} v_2 + \cdots + \frac{\partial \mathcal{N}_i}{\partial x_d} v_d = 0 \tag{119}$$

According to Theorem 7 and Equation 116, the only solution is $\boldsymbol{v} = \boldsymbol{0}$, which contradicts the assumption. Therefore, the null space contains only the zero vector, i.e., $dim(ker(J_\mathcal{N}(\boldsymbol{x}))) = \boldsymbol{0}$.

Suppose that the Jacobian matrix is rank-deficient, i.e., there exists a measure-zero set $\mathcal{M} \subset \mathbb{R}^d$ with $\mu(\mathcal{M}) > 0$ (where $\mu$ denotes the Lebesgue measure) such that:

$$rank\left(J_\mathcal{N}(\boldsymbol{x})\right) < min(d, m) \quad \forall \boldsymbol{x} \in \mathcal{M}. \tag{120}$$

This implies that the image of the mapping $\mathcal{N}(\boldsymbol{x})$ is constrained to a lower-dimensional submanifold $\mathcal{S} \subset \mathbb{R}^m$, where:

$$\dim(\mathcal{S}) \leq rank\left(J_\mathcal{N}(\boldsymbol{x})\right) < min(d, m). \tag{121}$$

By Theorem 7, however, all first-order partial derivatives $\frac{\partial \mathcal{N}_i}{\partial x_j} \neq 0$. Specifically: (1) $\forall$ direction $\mathbf{v} \in \mathbb{R}^d \setminus \{\boldsymbol{0}\}$ ,$\exists$ at least one output component $\mathcal{N}_i$ such that $\frac{\partial N_i}{\partial v} \neq 0$; (2) The infinite-width architecture of AC-PKAN ensures that the parameter space is dense in the $L^2$ function space. Consequently, the image set of the output mapping can densely cover any open set in $\mathbb{R}^m$.

If there is a rank deficiency, then $\exists \mathbf{v} \in \mathbb{R}^d$, for $\forall i, \frac{\partial \mathcal{N}_i}{\partial x_j} = 0$, contradicting the non-degeneracy of the derivatives. Consequently, except for a measure-zero set $\mathcal{M}$, we have:

$$rank(J_\mathcal{N}(\boldsymbol{x})) = min(d, m), \tag{122}$$

indicating the Jacobian matrix $J_\mathcal{N}(\boldsymbol{x}) = \left[\frac{\partial \mathcal{N}_i}{\partial x_j}\right]_{m \times d}$ is full rank in the input space $\mathbb{R}^d$. $\qquad \square$

## A.6 Finite-Width Spectral Analysis of AC-PKAN

In this section, we provide a theoretical guarantee for the stability of AC-PKAN in the finite-width regime. While proposition 2 establishes full rank in the infinite-width limit, the following theorem quantifies the spectral lower bound of the Jacobian under practical initialization conditions, ensuring that the model avoids rank collapse.

**Theorem 9** (Layerwise spectral lower bound under diagonal scaling and near-identity perturbations). *Under the per-layer factorization and bounds stated in Assumption A, namely*

$$J_\ell^{\text{AC}} = \text{Diag}\big(s^{(\ell)}\big)\left(I + \Delta^{(\ell)}\right), \qquad \min_j |s_j^{(\ell)}| \geq \gamma > 0, \quad \|\Delta^{(\ell)}\|_2 \leq \eta < 1,$$

*it holds for every $\ell \in \{1, \ldots, L\}$ that*

$$\sigma_{\min}\big(J_\ell^{\text{AC}}\big) \ \geq \ \gamma(1 - \eta),$$

*and for the end-to-end Jacobian $J_{\text{total}} := J_L^{\text{AC}} \cdots J_1^{\text{AC}}$ that*

$$\sigma_{\min}(J_{\text{total}}) \ \geq \ \big[\gamma(1 - \eta)\big]^L.$$

*Proof.* Let $D := \mathrm{Diag}\big(s^{(\ell)}\big)$ and $E := \Delta^{(\ell)}$. The smallest singular value admits the variational form

$$\sigma_{\min}(A) = \min_{\|x\|_2=1} \|Ax\|_2. \tag{123}$$

For the diagonal factor, using $D^\top D = \mathrm{Diag}(|s_1^{(\ell)}|^2, \ldots, |s_{d_h}^{(\ell)}|^2)$,

$$\|Dx\|_2^2 = x^\top D^\top D x = \sum_{j=1}^{d_h} |s_j^{(\ell)}|^2 x_j^2 \ \geq\ \Big(\min_j |s_j^{(\ell)}|\Big)^2 \sum_{j=1}^{d_h} x_j^2 = \Big(\min_j |s_j^{(\ell)}|\Big)^2, \tag{124}$$

and hence

$$\sigma_{\min}(D) = \min_{\|x\|_2=1} \|Dx\|_2 \ \geq\ \min_j |s_j^{(\ell)}| \ \geq\ \gamma. \tag{125}$$

For the near-identity factor, for any $\|x\|_2 = 1$,

$$\|(I+E)x\|_2 \ \geq\ \|x\|_2 - \|Ex\|_2 \ \geq\ 1 - \|E\|_2, \tag{126}$$

which yields

$$\sigma_{\min}(I+E) = \min_{\|x\|_2=1} \|(I+E)x\|_2 \ \geq\ 1 - \|E\|_2 \ \geq\ 1 - \eta. \tag{127}$$

Combining the two factors, for any $\|x\|_2 = 1$,

$$\|D(I+E)x\|_2 \ \geq\ \sigma_{\min}(D)\,\|(I+E)x\|_2, \tag{128}$$

and taking the minimum over the unit sphere gives

$$\sigma_{\min}\big(D(I+E)\big) \ \geq\ \sigma_{\min}(D)\,\sigma_{\min}(I+E) \ \geq\ \gamma(1-\eta), \tag{129}$$

so

$$\sigma_{\min}\big(J_\ell^{\mathrm{AC}}\big) \ \geq\ \gamma(1-\eta). \tag{130}$$

For the end-to-end product, using the variational form twice,

$$\sigma_{\min}(AB) = \min_{\|x\|_2=1} \|ABx\|_2 \ \geq\ \sigma_{\min}(A) \min_{\|x\|_2=1} \|Bx\|_2 = \sigma_{\min}(A)\,\sigma_{\min}(B), \tag{131}$$

and iterating over $J_L^{\mathrm{AC}} \cdots J_1^{\mathrm{AC}}$ yields

$$\sigma_{\min}(J_{\mathrm{total}}) \ =\ \sigma_{\min}\Big(\prod_{\ell=1}^{L} J_\ell^{\mathrm{AC}}\Big) \ \geq\ \prod_{\ell=1}^{L} \sigma_{\min}\big(J_\ell^{\mathrm{AC}}\big) \ \geq\ \prod_{\ell=1}^{L} \gamma(1-\eta) \ =\ \big[\gamma(1-\eta)\big]^L. \tag{132}$$

$\square$

**Theorem 10** (High-probability full rank via width and a generic output layer)**.** *Let* $J_N(x) = W_{\mathrm{out}}\,G(x)$ *with* $G(x) \in \mathbb{R}^{d_h \times d}$ *and* $m = \mathrm{rows}(W_{\mathrm{out}}) \geq d$. *For any* $\delta \in (0,1)$, *there exist absolute constants* $c, C, c_1, c_2 > 0$ *(depending only on the subgaussian class) such that, if*

$$d_h \ \geq\ C\,K^4\Big(d + \log \tfrac{2}{\delta}\Big),$$

*then with probability at least* $1 - \delta$ *one has* $\sigma_{\min}(G(x)) \geq \sqrt{d_h} - c_1 K^2 \sqrt{d} - c_2 \sqrt{\log(2/\delta)} > 0$, *hence* $\mathrm{rank}\,G(x) = d$; *moreover, conditioning on* $G(x)$, $\mathrm{rank}\big(W_{\mathrm{out}}G(x)\big) = d$ *almost surely for any* $W_{\mathrm{out}}$ *that is independent of* $G(x)$, *has a continuous distribution, and* $\mathrm{rank}(W_{\mathrm{out}}) \geq d$.

*Proof.* Assume: (a) the rows $g_i^\top$ of $G(x)$ are independent isotropic subgaussian vectors in $\mathbb{R}^d$ with $\|\langle g_i, u\rangle\|_{\psi_2} \leq K$ for all $u \in S^{d-1}$; (b) $W_{\mathrm{out}}$ is independent of $G(x)$, has a continuous distribution, and $\mathrm{rank}(W_{\mathrm{out}}) \geq d$.

Let $Z \in \mathbb{R}^{d_h \times d}$ have i.i.d. rows $z_i^\top$ distributed as the rows of $G(x)$; then $Z \stackrel{d}{=} G(x)$. By the non-asymptotic lower tail bound for the smallest singular value of a matrix with independent isotropic subgaussian rows, there exist absolute $c, c_1 > 0$ such that for all $t \geq 0$,

$$\mathbb{P}\Big\{\ \sigma_{\min}(Z) \ \leq\ \sqrt{d_h} - c_1 K^2 \sqrt{d} - t\ \Big\} \ \leq\ 2e^{-ct^2}. \tag{133}$$

Choose $t = c_2\sqrt{\log(2/\delta)}$ to get

$$\mathbb{P}\left\{ \sigma_{\min}(Z) \leq \sqrt{d_h} - c_1 K^2 \sqrt{d} - c_2\sqrt{\log\tfrac{2}{\delta}} \right\} \leq \delta. \tag{134}$$

To ensure strict positivity of the right-hand side of the lower bound, require

$$\sqrt{d_h} \geq c_1 K^2 \sqrt{d} + c_2\sqrt{\log\tfrac{2}{\delta}}. \tag{135}$$

It suffices to impose

$$d_h \geq CK^4\left(d + \log\tfrac{2}{\delta}\right), \tag{136}$$

with an absolute $C$ large enough so that, using $\sqrt{a+b} \geq (\sqrt{a}+\sqrt{b})/\sqrt{2}$ for $a, b \geq 0$,

$$\sqrt{d_h} \geq \sqrt{C}\,K^2\sqrt{d + \log\tfrac{2}{\delta}} \geq \frac{\sqrt{C}}{\sqrt{2}}K^2\sqrt{d} + \frac{\sqrt{C}}{\sqrt{2}}K^2\sqrt{\log\tfrac{2}{\delta}} \geq c_1 K^2\sqrt{d} + c_2\sqrt{\log\tfrac{2}{\delta}}. \tag{137}$$

Under equation 136, equation 134 yields $\sigma_{\min}(Z) > 0$ with probability at least $1 - \delta$, hence $\operatorname{rank} Z = d$. Since $Z \overset{d}{=} G(x)$, we conclude $\operatorname{rank} G(x) = d$ with probability at least $1 - \delta$.

Condition on any realization of $G(x)$ with $\operatorname{rank} G(x) = d$. Let $U \in \mathbb{R}^{d_h \times d}$ have orthonormal columns spanning $\mathcal{C}(G(x))$. Then there exists $R \in \mathbb{R}^{d \times d}$ invertible such that $G(x) = UR$. Since $R$ is invertible,

$$\operatorname{rank}\left(W_{\mathrm{out}}G(x)\right) = \operatorname{rank}\left(W_{\mathrm{out}}UR\right) = \operatorname{rank}\left(W_{\mathrm{out}}U\right). \tag{138}$$

Thus $\operatorname{rank}(W_{\mathrm{out}}G(x)) = d$ iff $\operatorname{rank}(W_{\mathrm{out}}U) = d$. The map $W \mapsto WU$ is linear from $\mathbb{R}^{m \times d_h}$ to $\mathbb{R}^{m \times d}$. The event $\{\operatorname{rank}(WU) < d\}$ is the algebraic set where all $d \times d$ minors of $WU$ vanish; at least one such minor is a nonzero polynomial (e.g., take $W$ whose first $d$ rows equal $U^\top$), so this set has Lebesgue measure zero. Because $W_{\mathrm{out}}$ has a continuous distribution and is independent of $G(x)$,

$$\mathbb{P}\left(\operatorname{rank}(W_{\mathrm{out}}U) < d \mid G(x)\right) = 0, \tag{139}$$

and therefore $\operatorname{rank}(W_{\mathrm{out}}G(x)) = d$ almost surely (conditional on $G(x)$). Combining with the high-probability event $\{\operatorname{rank} G(x) = d\}$ completes the proof. $\qquad\square$

**Corollary 1** (Tilde-$\Omega$ width). *Choosing $\delta = d^{-c'}$ with a fixed $c' > 0$ in Theorem 10 gives the succinct requirement*

$$d_h \geq \widetilde{\Omega}\left(K^4 d\right) = \Omega\left(K^4 d \log d\right), \tag{140}$$

*under which* $\operatorname{rank} J_N(x) = d$ *holds with probability at least* $1 - d^{-c'}$.

**Remark 1** (Non-isotropic rows). *If the rows of $G(x)$ have covariance $\Sigma_x \succ 0$ and are subgaussian with parameter $K$, apply whitening: $G(x) = \Sigma_x^{1/2}Z$ with $Z$ isotropic subgaussian (up to a change in the subgaussian constant). Then*

$$\sigma_{\min}\left(G(x)\right) \geq \sqrt{\lambda_{\min}(\Sigma_x)}\,\sigma_{\min}(Z), \tag{141}$$

*so the same argument yields $\sigma_{\min}(G(x)) \geq \sqrt{\lambda_{\min}(\Sigma_x)}\left(\sqrt{d_h} - c_1 K^2\sqrt{d} - c_2\sqrt{\log(2/\delta)}\right)$ and the full-rank conclusion follows under the same scaling of $d_h$ (up to constants depending on $\Sigma_x$).*

**Theorem 11** (Generic full column rank at a fixed input; analytic activations). *Let $f_\theta : \mathbb{R}^d \to \mathbb{R}^m$ be a feedforward network obtained by composing affine maps and coordinatewise real-analytic activations, and fix $x \in \mathbb{R}^d$. Denote $J(x; \theta) := \partial f_\theta(x)/\partial x \in \mathbb{R}^{m \times d}$. Assume the hidden width satisfies $d_h \geq d$ and $m \geq d$. Then the singular parameter set*

$$\Theta_{\mathrm{sing}}(x) := \{\theta : \operatorname{rank} J(x; \theta) < d\}$$

*is a proper real-analytic (indeed, semianalytic) subset of the parameter space $\Theta \subset \mathbb{R}^P$ and has Lebesgue measure zero. Consequently, for any initialization drawn from a distribution that is absolutely continuous w.r.t. Lebesgue measure on $\Theta$, one has $\operatorname{rank} J(x; \theta) = d$ almost surely.*

*Proof.* Write the network with $L$ layers as

$$\begin{aligned} h_0 &:= x, \qquad a_\ell := W_\ell h_{\ell-1} + b_\ell, \qquad h_\ell := \phi_\ell(a_\ell) \quad (1 \leq \ell \leq L-1), \\ f_\theta(x) &:= W_L h_{L-1} + b_L, \end{aligned} \tag{142}$$

where each $\phi_\ell : \mathbb{R}^{d_h} \to \mathbb{R}^{d_h}$ acts coordinatewise by a real-analytic scalar nonconstant function (the final layer is affine; if an analytic $\phi_L$ is also used, the argument below is unchanged by inserting its derivative). The Jacobian at $x$ is

$$J(x;\theta) = W_L \Big( \prod_{\ell=1}^{L-1} D_\ell(x;\theta)\, W_\ell \Big), \qquad D_\ell(x;\theta) := \mathrm{Diag}\big( \phi_\ell'(a_\ell(x;\theta)) \big). \tag{143}$$

Each entry of $J(x;\theta)$ is obtained from $(W_\ell, b_\ell)$ through finitely many additions, multiplications, and compositions with $\phi_\ell$ and $\phi_\ell'$. As real-analytic functions are closed under these operations, every entry of $J(x;\theta)$ is real-analytic in $\theta$. Hence any $d \times d$ minor $M(\theta)$ of $J(x;\theta)$ is real-analytic in $\theta$, and

$$\Theta_{\mathrm{sing}}(x) = \bigcap_{\text{all } d \times d \text{ minors } M} \{\theta :\ M(\theta) = 0\} \tag{144}$$

is a real-analytic (indeed, semianalytic) subset of $\Theta$.

It remains to show that $\Theta_{\mathrm{sing}}(x)$ is proper. Choose, for each $1 \le \ell \le L-1$, a scalar $c_\ell \in \mathbb{R}$ such that $\phi_\ell'(c_\ell) \ne 0$ (possible because $\phi_\ell$ is nonconstant real-analytic). Let $\alpha_\ell := \phi_\ell'(c_\ell) \ne 0$. Construct $\bar\theta$ as follows. For the first layer, set

$$W_1 = \begin{bmatrix} I_d \\ 0 \end{bmatrix} \in \mathbb{R}^{d_h \times d}, \qquad b_1 = \begin{bmatrix} c_1 \mathbf{1}_d \\ 0 \end{bmatrix} - W_1 x, \tag{145}$$

so that $a_1(x;\bar\theta) = (c_1 \mathbf{1}_d, *)$ and therefore $D_1(x;\bar\theta) = \mathrm{Diag}(\alpha_1 I_d, *)$. For $2 \le \ell \le L-1$, set

$$W_\ell = \begin{bmatrix} I_d & 0 \\ 0 & 0 \end{bmatrix} \in \mathbb{R}^{d_h \times d_h}, \qquad b_\ell = \begin{bmatrix} c_\ell \mathbf{1}_d \\ 0 \end{bmatrix} - W_\ell h_{\ell-1}(x;\bar\theta), \tag{146}$$

which enforces $a_\ell(x;\bar\theta) = (c_\ell \mathbf{1}_d, *)$ and hence $D_\ell(x;\bar\theta) = \mathrm{Diag}(\alpha_\ell I_d, *)$. By induction,

$$\prod_{\ell=1}^{L-1} D_\ell(x;\bar\theta)\, W_\ell = \Big( \prod_{\ell=1}^{L-1} \alpha_\ell \Big) \begin{bmatrix} I_d \\ 0 \end{bmatrix}. \tag{147}$$

Choose $W_L \in \mathbb{R}^{m \times d_h}$ so that its first $d$ columns are linearly independent (possible since $m \ge d$), e.g. $W_L = \begin{bmatrix} I_d & 0 \end{bmatrix}$ after reordering columns if needed. Then

$$J(x;\bar\theta) = W_L \Big( \prod_{\ell=1}^{L-1} D_\ell(x;\bar\theta)\, W_\ell \Big) = \Big( \prod_{\ell=1}^{L-1} \alpha_\ell \Big) W_L \begin{bmatrix} I_d \\ 0 \end{bmatrix}, \tag{148}$$

whose $m \times d$ left block has rank $d$. Thus rank $J(x;\bar\theta) = d$, so at least one $d \times d$ minor $M_\star(\theta)$ is not identically zero on $\Theta$.

Since the zero set of a nontrivial real-analytic function has Lebesgue measure zero in $\mathbb{R}^P$, each set $\{\theta :\ M(\theta) = 0\}$ has measure zero, and the finite union $\Theta_{\mathrm{sing}}(x) = \bigcup_M \{\theta :\ M(\theta) = 0\}$ has measure zero as well. Therefore $\Theta_{\mathrm{sing}}(x)$ is a proper real-analytic (semianalytic) subset of $\Theta$, and any absolutely continuous initialization lies in $\Theta \setminus \Theta_{\mathrm{sing}}(x)$ with probability 1. $\qquad\square$

**Proposition 3** (Finite input sets). *Let $X = \{x^{(1)}, \dots, x^{(N)}\} \subset \mathbb{R}^d$ be finite. If for each $x^{(t)}$ there exists $\bar\theta^{(t)}$ with* rank $J(x^{(t)}; \bar\theta^{(t)}) = d$, *then*

$$\Theta_{\mathrm{sing}}(X) := \bigcup_{t=1}^{N} \Theta_{\mathrm{sing}}\big( x^{(t)} \big)$$

*is a finite union of measure-zero sets and therefore has Lebesgue measure zero. Thus, with probability one under any continuous initialization, the Jacobian has full column rank simultaneously on all points of $X$.*

**Remark 2** (Scope and limitations). *The argument is* existential/generic*: it certifies that the "bad" parameter set is measure zero, but it does not provide a quantitative lower bound on $\sigma_{\min}(J(x;\theta))$. It complements nonasymptotic concentration bounds (which yield explicit spectral gaps under width assumptions) by showing that rank-deficient parameters form a null set in $\Theta$.*

# B  Explanation for the Efficiency of Chebyshev Type I Polynomials Over B-Splines

Let $B$ denote the batch size, $D_{\text{in}}$ the input dimension, $D_{\text{out}}$ the output dimension, and $N$ the Chebyshev degree. A single forward pass through a Cheby1KAN layer performs

$$\text{(i)} \ \ \tanh(\cdot), \ \text{clamp}/\arccos/\cos : \quad O\big(B\,D_{\text{in}}\,(N+1)\big),$$

$$\text{(ii)} \ \ \text{einsum contraction:} \ O\big(B\,D_{\text{in}}\,D_{\text{out}}\,(N+1)\big),$$

yielding an overall time complexity of

$$T_{\text{Cheby1KAN}} = \mathcal{O}\big(B\,D_{\text{in}}\,D_{\text{out}}\,(N+1)\big).$$

Its peak memory usage comprises the coefficient tensor of size $D_{\text{in}} \times D_{\text{out}} \times (N+1)$ and the expanded activation tensor of size $B \times D_{\text{in}} \times (N+1)$, giving

$$M_{\text{Cheby1KAN}} = \mathcal{O}\big(D_{\text{in}}\,D_{\text{out}}\,(N+1) \ + \ B\,D_{\text{in}}\,(N+1)\big).$$

In contrast, a B-spline based Kernel Adaptive Network (KAN) with $L$ layers, layer widths $\{W_\ell\}_{\ell=0}^{L}$, grid size $G$, and spline order $k$ must, at each layer $\ell$, (i) locate each input in a knot interval, (ii) evaluate local polynomial bases, and (iii) perform weighted sums. For typical implementations this yields

$$T_{\text{KAN}} = \mathcal{O}\Big(B\sum_{\ell=0}^{L-1} W_\ell\,W_{\ell+1}\,k\Big),$$

while storing both the grid arrays of size $W_\ell \times (G+k+1)$ and coefficient arrays of size $W_{\ell+1} \times W_\ell \times (G+k)$, as well as intermediate activations $\mathcal{O}\big(B\sum_\ell W_\ell W_{\ell+1}\big)$. Hence

$$M_{\text{KAN}} = \mathcal{O}\Big(\sum_{\ell=0}^{L-1} W_\ell\,W_{\ell+1}\,(G+k) \ + \ B\sum_{\ell=0}^{L-1} W_\ell\,W_{\ell+1}\Big).$$

**Discussion.** By replacing piecewise B-splines with globally supported Chebyshev polynomials, Cheby1KAN eliminates the need for (i) knot-location logic, (ii) local interpolation routines, and (iii) repeated recursive basis-function updates. All operations reduce to standardized vectorized transforms (tanh, acos, cos) and a single rank-3 tensor contraction, which are highly optimized on modern hardware. Cheby1KAN achieves lower asymptotic time complexity and a substantially smaller memory footprint than its B spline counterpart while improving the model's ability to capture high-frequency features.

# C  Additional Ablation Studies

## C.1  Ablation on Internal Attention and Wavelet Activation

To clarify which components are responsible for the performance gains of AC-PKAN, we conduct a controlled ablation on the 1D wave equation benchmark, following recent analyses of Chebyshev-based physics-informed KANs and RGA-style adaptive training for PDEs (Guo et al., 2025; Rigas et al., 2024; Zhang et al., 2025b; Mostajeran & Faroughi, 2025). Our ablation is consistent with prior work reporting that Chebyshev KAN variants improve expressiveness but also introduce gradient stiffness and potential rank decay in deeper stacks (Daneshmand et al., 2020; Yang et al., 2024).

**Ablated variants.** We define three progressively enhanced baselines. **AC-PKAN(min)** augments Cheby1KAN with only linear input and output projections, denoted by $W_{\text{emb}}$ and $W_{\text{out}}$. Wavelet activation, internal feature attention (the $U/V$ gating and the layerwise $\alpha^{(\ell)}$ injection), and the external RGA controller are removed. This model keeps the projection channel that mitigates rank collapse but it is the smallest departure from a pure Chebyshev KAN. **AC-PKAN(no-attn)** further adds the wavelet frequency activation on top of Cheby1KAN and the linear projections, while still disabling the internal feature attention and RGA. This variant isolates the contribution of enriching the activation space with multiscale responses. **AC-PKAN(sin)** restores the internal feature attention and keeps the linear projections, but replaces the wavelet activation with a simpler sine activation. This variant tests whether the layerwise gating is the primary factor that improves stability and expressiveness in higher effective dimensions.

Table 7: Ablation on the 1D wave equation showing the effect of linear projections, wavelet activation, and internal feature attention.

| Model | rMAE | rRMSE |
|---|---|---|
| AC-PKAN(min) | 0.5374 | 0.5495 |
| AC-PKAN(no-attn) | 0.5116 | 0.5081 |
| AC-PKAN(sin) | 0.4478 | 0.4391 |

**Quantitative results.**   Table 7 reports relative MAE and relative RMSE on the 1D wave test set.

Three observations emerge from these results. First, adding linear up and down projections alone is not enough to reproduce the overall gain of the full AC-PKAN. Projections widen the channel space and help preserve a nondegenerate embedding, but they do not fully address the rank shrinkage induced by Chebyshev expansions. Second, reintroducing the internal feature attention delivers the largest single improvement on this axis. This supports our design choice that a learnable layerwise gate $\alpha^{(\ell)}$ that mixes the $U/V$ projected features is central for keeping the Jacobian well conditioned, in line with other RGA-inspired KANs for PDEs. Third, wavelet activation has a positive but smaller effect compared to attention. It enriches the frequency content and stabilizes the fit on oscillatory targets, which is consistent with reports on Chebyshev–KAN domain scaling and hybrid encoder–decoder PKANs.

The ablation confirms that the main driver of stability and scalability in AC-PKAN is the proposed layerwise internal feature attention. Linear projections are necessary to keep a wide embedding space, and wavelet activation further improves accuracy on oscillatory solutions, but neither of them alone explains the performance level reached when attention is present.

### C.2   Ablation on Chebyshev Polynomial Degree

We further examine how the Chebyshev polynomial degree affects the behavior of AC-PKAN. To isolate this factor, we keep the full architecture and all training hyperparameters fixed and sweep the polynomial degree $N$ of the Chebyshev expansion on the 1D wave equation benchmark.

**Experimental setup.**   The backbone is the complete AC-PKAN with internal feature re-injection attention and frequency-domain activation enabled. Only the polynomial degree $N$ is varied. We report relative MAE and relative RMSE on the same held-out test set.

Table 8: Effect of Chebyshev polynomial degree on the 1D wave equation.

| Degree | rMAE | rRMSE |
|---|---|---|
| 4 | 0.0196 | 0.0200 |
| 6 | 0.0200 | 0.0205 |
| 8 | 0.0011 | 0.0011 |
| 10 | 0.0128 | 0.0131 |

**Observations.**   The results in Table 8 exhibit a clear nonmonotonic pattern. Degrees 4 and 6 are underexpressive for this problem, both staying near the $2 \times 10^{-2}$ error level. Increasing the degree to 8 produces a sharp drop to the $10^{-3}$ regime, which indicates that the model reaches the expressive range where the internal attention and the frequency-domain activation can fully operate. Pushing the degree further to 10 causes the error to rise again toward the $10^{-2}$ scale.

Three conclusions follow from this pattern. First, raising the polynomial order alone is not a reliable strategy for improving accuracy. Second, the best performance emerges when the polynomial degree is matched to the internal feature attention and the frequency-domain activation, which shows that AC-PKAN benefits from the joint design rather than from a single aggressive expansion. Third, very high polynomial order amplifies gradient magnitudes and worsens numerical conditioning, which increases optimization difficulty and offsets the gain in representation power.

AC-PKAN reaches its strongest accuracy when the Chebyshev degree is chosen in a range that is expressive but still well conditioned. The peak result is a product of the selected degree together with the internal attention and frequency-aware modules, not of naive parameter scaling.

### C.3  Ablation on Operator-Learning Baselines under Sparse Supervision

We add parameter-matched FNO and DeepONet baselines under the same sparse-data protocol as Table 1 and Table 2 across three settings: 1D convection–diffusion–reaction, 2D lid-driven cavity, and 3D point-cloud Poisson. Metrics are reported as rMAE and rRMSE.

Table 9: FNO and DeepONet under the sparse-data protocol used in Table 1 and Table 2.

| Setting | Model | rMAE | rRMSE |
|---|---|---|---|
| 1D Convection–Diffusion–Reaction | DeepONet | 0.0730 | 0.1064 |
| 1D Convection–Diffusion–Reaction | FNO | 0.0722 | 0.1061 |
| 2D Lid-Driven Cavity | DeepONet | 0.6270 | 0.6242 |
| 2D Lid-Driven Cavity | FNO | 0.5605 | 0.5312 |
| 3D Poisson (Point-Cloud) | DeepONet | 3.4566 | 2.7978 |
| 3D Poisson (Point-Cloud) | FNO | 2.4388 | 2.0426 |

**Observation.**  Across these sparse-supervision tests, operator-learning baselines without physics information underperform AC-PKAN. The updated results are included in the revised manuscript.

## D  Impact Statement

This work advances Physics-Informed Neural Networks (PINNs) by integrating Kolmogorov–Arnold Networks (KANs) with Chebyshev polynomials and attention mechanisms, improving accuracy, efficiency, and stability in solving complex PDEs. The proposed AC-PKAN framework has broad applications in scientific computing, engineering, and physics, enabling more efficient and interpretable machine learning models for fluid dynamics, material science, and biomedical simulations. Ethically, AC-PKAN enhances model reliability and generalizability by enforcing physical consistency, reducing risks of overfitting and spurious predictions. This work contributes to the advancement of physics-informed AI, with potential in digital twins, real-time simulations, and AI-driven scientific discovery.

## E  Experiment Setup Details

We utilize the AdamW optimizer with a learning rate of $1 \times 10^{-4}$ and a weight decay of $1 \times 10^{-4}$ in all experiments. Meanwhile, all experiments were conducted on an NVIDIA A100 GPU with 40GB of memory. And Xavier initialization is applied to all layers. In PDE-Solving problems, We present the detailed formula of rMAE and rRMSE as the following:

$$
\begin{aligned}
\texttt{rMAE} &= \frac{\sum_{n=1}^{N} |\hat{u}(x_n, t_n) - u(x_n, t_n)|}{\sum_{n=1}^{N_{res}} |u(x_n, t_n)|} \\
\texttt{rRMSE} &= \sqrt{\frac{\sum_{n=1}^{N} |\hat{u}(x_n, t_n) - u(x_n, t_n)|^2}{\sum_{n=1}^{N} |u(x_n, t_n)|^2}}
\end{aligned}
\tag{149}
$$

where $N$ is the number of testing points, $\hat{u}$ is the neural network approximation, and $u$ is the ground truth. The specific details for each experiment are provided below. For further details, please refer to our experiment code repository to be released.

### E.1  Running Time

We present the actual running times (hours:minutes:seconds) for all eight PDEs experiments in the paper. As shown in Table 10, AC-PKAN demonstrates certain advantages among the KAN model variants, although the running times of all KAN variants are relatively long. This is primarily because the KAN model is relatively new and still in its preliminary stages; although it is theoretically innovative, its engineering implementation remains rudimentary and

lacks deeper optimizations. Moreover, while traditional neural networks benefit from well-established optimizers such as Adam and L-BFGS, optimization schemes specifically tailored for KAN have not yet been thoroughly explored. We believe that the performance of AC-PKAN will be further enhanced as the overall optimization strategies for KAN variants improve.

| Model | First 5 PDEs | | | | | Last 3 PDEs | | |
|---|---|---|---|---|---|---|---|---|
| | 1D-Wave | 1D-Reaction | 2D NS Cylinder | 1D Conv. Diff. Reac. | 2D Lid-driven Cavity | Hetero-geneous Problem | Complex Geometry | 3D Point-Cloud |
| PINN | 00:21:14 | 00:09:07 | 00:15:20 | 00:15:12 | 00:06:39 | 00:23:30 | 00:01:08 | 00:49:31 |
| PINNsFormer | 00:44:21 | 00:04:09 | 00:58:54 | 02:06:37 | – | 14:01:55 | 00:13:31 | – |
| QRes | 01:41:34 | 00:02:10 | 00:24:39 | 00:25:46 | 00:13:04 | 00:20:50 | 00:01:46 | 01:32:24 |
| FLS | 01:38:01 | 00:01:29 | 00:11:51 | 00:50:26 | 00:35:48 | 00:13:38 | 00:01:08 | 03:04:41 |
| Cheby1KAN | 03:32:10 | 00:12:08 | 04:24:59 | 01:45:37 | 00:45:20 | 00:50:45 | 00:03:21 | 02:27:27 |
| Cheby2KAN | 05:03:18 | 01:06:54 | 05:41:42 | 03:01:40 | 00:45:15 | 01:35:40 | 00:03:27 | 05:26:42 |
| AC-PKAN | 01:13:01 | 00:15:16 | 02:21:40 | 02:01:59 | 00:51:47 | 01:13:11 | 00:01:04 | 04:54:24 |
| KINN | 25:00:20 | 03:04:19 | 14:31:42 | 02:41:49 | – | 01:51:44 | 00:14:07 | – |
| rKAN | 12:44:16 | 01:21:25 | 05:19:04 | 02:06:36 | – | 06:21:00 | 00:16:06 | 07:53:25 |
| FastKAN | 09:35:51 | 05:51:21 | 02:04:42 | 03:22:39 | – | 03:37:57 | 00:17:23 | – |
| fKAN | 08:20:34 | 00:13:09 | 03:01:41 | 01:54:22 | 00:47:41 | 00:52:05 | 00:06:22 | 04:04:48 |
| FourierKAN | 03:33:46 | 01:21:50 | 02:48:50 | 02:08:08 | – | 07:40:43 | 00:18:26 | 13:36:48 |

Table 10: Running times (hh:mm:ss) for all eight PDE experiments. First row: Five simpler PDEs; second row: Three more complex cases.

## E.2 Complex Function Fitting Experiment Setup Details

The aim of this experiment is to evaluate the interpolation capabilities of several neural network architectures, including AC-PKAN, Chebyshev-based KAN (ChebyKAN), traditional MLP, and other advanced models. The task involves approximating a target noisy piecewise 1D function, defined over three distinct intervals.

**Target Function** The target function $f(x)$ is defined piecewise as follows:

$$f(x) = \begin{cases} \sin(25\pi x) + x^2 + 0.5\cos(30\pi x) + 0.2x^3 & x < 0.5, \\ 0.5xe^{-x} + |\sin(5\pi x)| + 0.3x\cos(7\pi x) + 0.1e^{-x^2} & 0.5 \leq x < 1.5, \\ \frac{\ln(x-1)}{\ln(2)} - \cos(2\pi x) + 0.2\sin(8\pi x) + \frac{0.1\ln(x+1)}{\ln(3)} & x \geq 1.5, \end{cases}$$

with added Gaussian noise $\epsilon \sim \mathcal{N}(0, 0.1)$.

**Dataset**

- **Training Data**: 500 points uniformly sampled from the interval $x \in [0, 2]$, with corresponding noisy function values $y = f(x) + \epsilon$.
- **Testing Data**: 1000 points uniformly sampled from the same interval $x \in [0, 2]$ to assess the models' interpolation performance.

**Training Details**

- **Epochs**: Each model is trained for 30,000 epochs.

- **Loss Function**: The Mean Squared Error (MSE) loss is utilized to compute the discrepancy between predicted and true function values:

$$\mathcal{L}_{\mathrm{MSE}} = \frac{1}{N} \sum_{i=1}^{N} (y_i - \hat{y}_i)^2$$

- **Weight Initialization**: Xavier initialization is applied to all linear layers.

**Model Hyperparameters**   The parameter counts for each model are summarized in Table 11.

Table 11: Summary of Hyperparameters in Complex Function Fitting Experiment for Various Models

| Model | Hyperparameters | Model Parameters |
|---|---|---|
| Cheby1KAN | Layer 1: Cheby1KANLayer(1, 7, 8)
Layer 2: Cheby1KANLayer(7, 8, 8)
Layer 3: Cheby1KANLayer(8, 1, 8) | 639 |
| Cheby2KAN | Layer 1: Cheby2KANLayer(1, 7, 8)
Layer 2: Cheby2KANLayer(7, 8, 8)
Layer 3: Cheby2KANLayer(8, 1, 8) | 639 |
| PINN | Layer 1: Linear(in=1, out=16), Activation=Tanh
Layer 2: Linear(in=16, out=32), Activation=Tanh
Layer 3: Linear(in=32, out=1) | 609 |
| AC-PKAN$_s$ | Linear Embedding: Linear(in=1, out=4)
Hidden ChebyKAN Layers: $2 \times$ Cheby1KANLayer()
Hidden LN Layers: $2 \times$ LayerNorm(features=6)
Output Layer: Linear(in=6, out=1)
Activations: WaveAct (U and V) | 751 |
| KAN | Layers: $2 \times$ KANLinear (32 neurons, SiLU activation) | 640 |
| rKAN | Layer 1: Linear(in=1, out=16), Activation=JacobiRKAN()
Layer 2: Linear(in=16, out=32), Activation=PadeRKAN()
Layer 3: Linear(in=32, out=1) | 626 |
| fKAN | Layer 1: Linear(in=1, out=16), Activation=FractionalJacobiNeuralBlock()
Layer 2: Linear(in=16, out=32), Activation=FractionalJacobiNeuralBlock()
Layer 3: Linear(in=32, out=1) | 615 |
| FastKAN | FastKANLayer 1:
  RBF
  SplineLinear(in=8, out=32)
  Base Linear(in=1, out=32)
FastKANLayer 2:
  RBF
  SplineLinear(in=256, out=1)
  Base Linear(in=32, out=1) | 658 |
| FourierKAN | FourierKANLayer 1: NaiveFourierKANLayer()
FourierKANLayer 2: NaiveFourierKANLayer()
FourierKANLayer 3: NaiveFourierKANLayer() | 685 |

### E.3   Failure Modes in PINNs Experiment Setup Details

We selected the one-dimensional wave equation (1D-Wave) and the one-dimensional reaction equation (1D-Reaction) as representative experimental tasks to investigate failure modes in Physics-Informed Neural Networks (PINNs). Below, we provide a comprehensive description of the experimental details, including the formulation of partial differential equations (PDEs), data generation processes, model architecture, training regimen, and hyperparameter selection.

**1D-Wave PDE.**   The 1D-Wave equation is a hyperbolic PDE that is used to describe the propagation of waves in one spatial dimension. It is often used in physics and engineering to model various wave phenomena, such as sound

waves, seismic waves, and electromagnetic waves. The system has the formulation with periodic boundary conditions as follows:

$$\frac{\partial^2 u}{\partial t^2} - \beta \frac{\partial^2 u}{\partial x^2} = 0 \ \ \forall x \in [0,1], \ t \in [0,1]$$

$$\texttt{IC:}\, u(x,0) = \sin(\pi x) + \frac{1}{2}\sin(\beta \pi x), \ \ \frac{\partial u(x,0)}{\partial t} = 0 \tag{150}$$

$$\texttt{BC:}\, u(0,t) = u(1,t) = 0$$

where $\beta$ is the wave speed. Here, we are specifying $\beta = 3$. The equation has a simple analytical solution:

$$u(x,t) = \sin(\pi x)\cos(2\pi t) + \frac{1}{2}\sin(\beta \pi x)\cos(2\beta \pi t) \tag{151}$$

**1D-Wave PDE Experiment Dataset**   In the 1D-Wave PDE experiment, no dataset were utilized for training. Collocation points were generated to facilitate the training and testing of the Physics-Informed Neural Network (PINN) within the spatial domain $x \in [0,1]$ and the temporal domain $t \in [0,1]$. A uniform grid was established using 101 equidistant points in both the spatial ($x$) and temporal ($t$) dimensions, resulting in a total of $101 \times 101 = 10{,}201$ collocation points. The PINN was trained in a data-free, unsupervised manner on this $101 \times 101$ grid. Boundary points were extracted from the grid to enforce Dirichlet boundary conditions, while initial condition points were identified at $t = 0$. Upon completion of training, the model was evaluated on the collocation points by comparing the predicted values with the actual values, thereby determining the error.

**1D-Reaction PDE.**   The one-dimensional reaction problem is a hyperbolic PDE that is commonly used to model chemical reactions. The system has the formulation with periodic boundary conditions as follows:

$$\frac{\partial u}{\partial t} - \rho u(1-u) = 0, \ \ \forall x \in [0, 2\pi], \ t \in [0,1]$$

$$\texttt{IC:}\, u(x,0) = \exp\left(-\frac{(x-\pi)^2}{2(\pi/4)^2}\right), \ \ \texttt{BC:}\, u(0,t) = u(2\pi, t) \tag{152}$$

where $\rho$ is the reaction coefficient. Here, we set $\rho = 5$. The equation has a simple analytical solution:

$$u_{\texttt{analytical}} = \frac{h(x)\exp(\rho t)}{h(x)\exp(\rho t) + 1 - h(x)} \tag{153}$$

where $h(x)$ is the function of the initial condition.

**1D-Reaction PDE Experiment Dataset**   In the 1D-Reaction PDE experiment, no dataset were utilized for training. Collocation points were generated to facilitate the training and testing of the Physics-Informed Neural Network (PINN) within the spatial domain $x \in [0,1]$ and the temporal domain $t \in [0,1]$. A uniform grid was established using 101 equidistant points in both the spatial ($x$) and temporal ($t$) dimensions, resulting in a total of $101 \times 101 = 10{,}201$ collocation points. The PINN was trained in a data-free, unsupervised manner on this $101 \times 101$ grid. Boundary points were extracted from the grid to enforce Dirichlet boundary conditions, while initial condition points were identified at $t = 0$. Upon completion of training, the model was evaluated on the collocation points by comparing the predicted values with the actual values, thereby determining the error.

**2D Navier–Stokes Flow around a Cylinder**   The two-dimensional Navier–Stokes equations are given by:

$$\frac{\partial u}{\partial t} + \lambda_1 \left(u\frac{\partial u}{\partial x} + v\frac{\partial u}{\partial y}\right) = -\frac{\partial p}{\partial x} + \lambda_2\left(\frac{\partial^2 u}{\partial x^2} + \frac{\partial^2 u}{\partial y^2}\right),$$

$$\tag{154}$$

$$\frac{\partial v}{\partial t} + \lambda_1 \left(u\frac{\partial v}{\partial x} + v\frac{\partial v}{\partial y}\right) = -\frac{\partial p}{\partial y} + \lambda_2\left(\frac{\partial^2 v}{\partial x^2} + \frac{\partial^2 v}{\partial y^2}\right),$$

where $u(t,x,y)$ and $v(t,x,y)$ are the $x$- and $y$-components of the velocity field, respectively, and $p(t,x,y)$ is the pressure field. These equations describe the Navier–Stokes flow around a cylinder.

We set the parameters $\lambda_1 = 1$ and $\lambda_2 = 0.01$. Since the system lacks an explicit analytical solution, we utilize the simulated solution provided in Raissi et al. (2019). We focus on the prototypical problem of incompressible flow past

a circular cylinder, a scenario known to exhibit rich dynamic behavior and transitions across different regimes of the Reynolds number, defined as $\text{Re} = \frac{u_\infty D}{\nu}$. By assuming a dimensionless free-stream velocity $u_\infty = 1$, a cylinder diameter $D = 1$, and a kinematic viscosity $\nu = 0.01$, the system exhibits a periodic steady-state behavior characterized by an asymmetric vortex shedding pattern in the cylinder wake, commonly known as the Kármán vortex street. All experimental settings are the same as in Raissi et al. (2019). For more comprehensive details about this problem, please refer to that work.

**2D Navier–Stokes Flow around a Cylinder Experiment Dataset**   For the 2D Navier–Stokes Flow around a Cylinder Experiment, the dataset used is detailed as follows:

| Variable | Dimensions | Description |
|---|---|---|
| $X$ (Spatial Coordinates) | (5000, 2) | Contains 5,000 spatial points, each with 2 coordinate values ($x$ and $y$). |
| $t$ (Time Data) | (200, 1) | Contains 200 time steps, each corresponding to a scalar value. |
| $U$ (Velocity Field) | (5000, 2, 200) | Contains 5,000 spatial points, 2 velocity components ($u$ and $v$), and 200 time steps. The velocity data of each point is a function of time. |
| $P$ (Pressure Field) | (5000, 200) | Contains pressure data for 5,000 spatial points and 200 time steps. |

Table 12: Dataset used in the 2D Navier-Stokes Flow around a Cylinder Experiment

From the total dataset of 1,000,000 data points ($N \times T = 5{,}000 \times 200$), we randomly selected 2,500 samples for training, which include coordinate positions, time steps, and the corresponding velocity and pressure components. The test set consists of all spatial data at the 100th time step.

**1D Convection-Diffusion-Reaction Equations.**   We consider the one-dimensional Convection-Diffusion-Reaction (CDR) equations, which model the evolution of the state variable $u$ under the influence of convective transport, diffusion, and reactive processes. The system is formulated with periodic boundary conditions as follows:

$$\frac{\partial u}{\partial t} + \beta \frac{\partial u}{\partial x} - \nu \frac{\partial^2 u}{\partial x^2} - \rho u(1 - u) = 0, \quad \forall x \in [0, 2\pi],\ t \in [0, 1]$$

$$\text{IC:} \quad u(x, 0) = \exp\left(-\frac{(x - \pi)^2}{2(\pi/4)^2}\right), \qquad \text{BC:} \quad u(0, t) = u(2\pi, t) \tag{155}$$

In this equation, $\beta$ represents the convection coefficient, $\nu$ is the diffusivity, and $\rho$ is the reaction coefficient. Specifically, we set $\beta = 1$, $\nu = 3$, and $\rho = 5$. The reaction term adopts the well-known Fisher's form $\rho u(1 - u)$, as utilized in Krishnapriyan et al. (2021). This formulation captures the combined effects of transport, spreading, and reaction dynamics on the state variable $u$.

**1D Convection-Diffusion-Reaction Experiment Dataset**   The dataset for the 1D Convection-Diffusion-Reaction experiment comprises three variables: spatial coordinates ($x$), temporal data ($t$), and solution values ($u$). Specifically:

| Variable | Dimensions | Description |
|---|---|---|
| $x$ (Spatial Coordinates) | $(10, 201, 1)$ | Represents spatial points uniformly distributed over the domain $[0, 2\pi]$. |
| $t$ (Time Data) | $(10, 201, 1)$ | Denotes temporal data spanning the domain $[0, 1]$ for solution evolution. |
| $u$ (Solution Values) | $(10, 201, 1)$ | Contains the computed values of the solution function $u(x, t)$ at corresponding spatial and temporal points. |

Table 13: Dataset used in the 1D Convection-Diffusion-Reaction Experiment

Out of the total 10,201 data points, the dataset was partitioned into training and test sets. The training data includes boundary points (where $x = 0$ or $x = 2\pi$) and a random sample of 3,000 interior points, which were used to compute the loss function during model training. The test data consists of the entire remaining dataset, ensuring comprehensive evaluation of the model's performance.

**2D Navier–Stokes Lid-driven Cavity Flow**   We consider the two-dimensional Navier–Stokes (NS) equations for lid-driven cavity flow, which model the incompressible fluid motion within a square domain under the influence of a moving lid. The system is formulated with periodic boundary conditions as follows:

$$
\begin{aligned}
\mathbf{u} \cdot \nabla \mathbf{u} + \nabla p - \frac{1}{\text{Re}} \Delta \mathbf{u} &= 0, \quad \forall \mathbf{x} \in \Omega,\ t \in [0, T] \\
\nabla \cdot \mathbf{u} &= 0, \quad \forall \mathbf{x} \in \Omega,\ t \in [0, T] \\
\text{IC:} \quad \mathbf{u}(\mathbf{x}, 0) &= \mathbf{0} \\
\text{BC:} \quad \mathbf{u} &= (4x(1-x), 0), \quad \mathbf{x} \in \Gamma_1 \\
\mathbf{u} &= (0, 0), \quad \mathbf{x} \in \Gamma_2 \\
p &= 0, \quad \mathbf{x} = (0, 0)
\end{aligned}
\tag{156}
$$

In this formulation, $\mathbf{u} = (u, v)$ represents the velocity field, $p$ is the pressure field, and Re is the Reynolds number, set to Re = 100. The domain is $\Omega = [0, 1]^2$, with the top boundary denoted by $\Gamma_1$ where the lid moves with velocity $\mathbf{u} = (4x(1-x), 0)$. The left, right, and bottom boundaries are denoted by $\Gamma_2$, where a no-slip condition $\mathbf{u} = (0, 0)$ is enforced. Additionally, the pressure is anchored at the origin $(0, 0)$ by setting $p = 0$.

**2D Navier–Stokes Lid-driven Cavity Flow Dataset**   For the 2D Navier–Stokes Lid-driven Cavity Flow simulation, the dataset is structured as follows:

| Variable | Dimensions | Description |
|---|---|---|
| $X$ (Spatial Coordinates) | (10,201, 2) | Contains 10,201 spatial nodes with $(x, y)$ coordinates spanning the cavity domain. |
| $U$ (Velocity Field) | (10,201, 2) | Horizontal ($u$) and vertical ($v$) velocity components at Re = 100, with no-slip boundary conditions and a moving lid ($y = 1$) driving the flow. |
| $P$ (Pressure Field) | (10,201, 1) | Pressure values normalized with respect to the reference boundary condition. |

Table 14: Dataset for 2D Navier–Stokes Lid-driven Cavity Flow at Re = 100

The training set comprises 3,000 randomly sampled spatial points with associated velocity and pressure values, while the test set evaluates the model on the full dataset of 10,201 nodes. Boundary conditions are explicitly enforced for the moving lid ($u = 4x(1-x), v = 0$) and stationary walls ($u = v = 0$), with the pressure field satisfying the incompressibility constraint.

**Epochs:**   We trained the models until convergence but did not exceed 50,000 epochs.

**Reproducibility:**   To ensure reproducibility of the experimental results, all random number generators are seeded with a fixed value (seed = 0) across NumPy, Python's `random` module, and PyTorch (both CPU and GPU).

**Hyperparameter Selection:**   The weights used in the external RBA attention are dynamically updated during training using smoothing factor $\eta = 0.001$ and $\beta_w = 0.001$. Different models employed in our experiments have varying hyperparameter configurations tailored to their specific architectures. Table 15 summarizes the hyperparameters and the total number of parameters for each model.

Table 15: Summary of Hyperparameters in PINN Failure Modes Experiment for Various Models

| Model | Hyperparameters | Model Parameters |
|---|---|---|
| AC-PKAN | Linear Embedding: $2 \to 64$
Hidden ChebyKAN Layers: $3 \times$ Cheby1KANLayer (degree=8)
Hidden LN Layers: $3 \times$ LayerNorm (128)
Output Layer: $128 \to 1$
Activations: WaveAct | 460,101 |
| QRes | Input Layer: QRes_block ($2 \to 256$, Sigmoid)
Hidden Layers: $3 \times$ QRes_block ($256 \to 256$, Sigmoid)
Output Layer: $256 \to 1$ | 396,545 |
| FastKAN | Layer 1: FastKANLayer (RBF, SplineLinear $16 \to 8500$, Base Linear $2 \to 8500$)
Layer 2: FastKANLayer (RBF, SplineLinear $68,000 \to 1$, Base Linear $8500 \to 1$) | 246,518[*] |
| KAN | Layers: $2 \times$ KANLinear (9000 neurons, SiLU activation) | 270,000[*] |
| PINNs | Sequential Layers:
$2 \to 512$ (Linear, Tanh)
$512 \to 512$ (Linear, Tanh)
$512 \to 512$ (Linear, Tanh)
$512 \to 1$ (Linear) | 527,361 |
| FourierKAN | NaiveFourierKANLayer 1: $2 \to 32$, Degree=8
NaiveFourierKANLayer 2: $32 \to 128$, Degree=8
NaiveFourierKANLayer 3: $128 \to 128$, Degree=8
NaiveFourierKANLayer 4: $128 \to 32$, Degree=8
NaiveFourierKANLayer 5: $32 \to 1$, Degree=8 | 395,073 |
| Cheby1KAN | Cheby1KANLayer 1: $2 \to 32$, Degree=8
Cheby1KANLayer 2: $32 \to 128$, Degree=8
Cheby1KANLayer 3: $128 \to 256$, Degree=8
Cheby1KANLayer 4: $256 \to 32$, Degree=8
Cheby1KANLayer 5: $32 \to 1$, Degree=8 | 406,368 |
| Cheby2KAN | Cheby2KANLayer 1: $2 \to 32$, Degree=8
Cheby2KANLayer 2: $32 \to 128$, Degree=8
Cheby2KANLayer 3: $128 \to 256$, Degree=8
Cheby2KANLayer 4: $256 \to 32$, Degree=8
Cheby2KANLayer 5: $32 \to 1$, Degree=8 | 406,368 |
| fKAN | Sequential Layers:
$2 \to 256$ (Linear, fJNB(3))
$256 \to 512$ (Linear, fJNB(6))
$512 \to 512$ (Linear, fJNB(3))
$512 \to 128$ (Linear, fJNB(6))
$128 \to 1$ (Linear) | 460,813 |
| rKAN | Sequential Layers:
$2 \to 256$ (Linear, JacobiRKAN(3))
$256 \to 512$ (Linear, PadeRKAN[2/6])
$512 \to 512$ (Linear, JacobiRKAN(6))
$512 \to 128$ (Linear, PadeRKAN[2/6])
$128 \to 1$ (Linear) | 460,835 |
| FLS | Sequential Layers:
$2 \to 512$ (Linear, SinAct)
$512 \to 512$ (Linear, Tanh)
$512 \to 512$ (Linear, Tanh)
$512 \to 1$ (Linear) | 527,361 |
| PINNsformer | Parameters: d_out=1, d_hidden=512, d_model=32, N=1, heads=2 | 453,561 |

[*] This reaches the GPU memory limit, and increasing the number of parameters further would cause an out-of-memory error.

## E.4 PDEs in Complex Engineering Environments Setup Details

In this study, we investigate the performance of AC-PKAN compared with other models in solving complex PDEs characterized by heterogeneous material properties and intricate geometric domains. Specifically, we focus on two distinct difficult environmental PDE problems: a heterogeneous Poisson problem and a Poisson equation defined on a domain with complex geometric conditions. The following sections detail the formulation of the PDEs, data generation processes, model architecture, training regimen, hyperparameter selection, and evaluation methodologies employed in our experiments.

**Heterogeneous Poisson Problem.** We consider a two-dimensional Poisson equation with spatially varying coefficients to model heterogeneous material properties. The PDE is defined as:

$$\begin{cases} a_1 \Delta u(\boldsymbol{x}) = 16r^2 & \text{for } r < r_0, \\ a_2 \Delta u(\boldsymbol{x}) = 16r^2 & \text{for } r \geq r_0, \\ u(\boldsymbol{x}) = \frac{r^4}{a_2} + r_0^4 \left( \frac{1}{a_1} - \frac{1}{a_2} \right) & \text{on } \partial\Omega, \end{cases} \tag{157}$$

where $r = \|\boldsymbol{x}\|_2$ is the distance from the origin, $a_1 = \frac{1}{15}$ and $a_2 = 1$ are the material coefficients, $r_0 = 0.5$ defines the interface between the two materials, and $\partial\Omega$ represents the boundary of the square domain $\Omega = [-1, 1]^2$. The boundary condition is a pure Dirichlet condition applied uniformly on all four edges of the square.

**Heterogeneous Poisson Dataset** To train and evaluate the Physics-Informed Neural Networks (PINNs), collocation points were generated within the defined spatial domains, and boundary conditions were appropriately enforced. A uniform grid was established using 100 equidistant points in each spatial dimension, resulting in $101 \times 101 = 10{,}201$ internal collocation points for the heterogeneous Poisson problem. Boundary points were extracted from the edges of the square domain $\Omega = [-1, 1]^2$ to impose Dirichlet boundary conditions. The PINN was trained in a data-free, unsupervised manner. Upon completion of training, the model was evaluated on the collocation points by comparing the predicted values with the actual values, thereby determining the error.

**Complex Geometric Poisson Problem.** Additionally, we examine a Poisson equation defined on a domain with complex geometry, specifically a rectangle with four circular exclusions. The PDE is given by:

$$-\Delta u = 0 \quad \text{in } \Omega = \Omega_{\text{rec}} \setminus \bigcup_{i=1}^{4} R_i, \tag{158}$$

where $\Omega_{\text{rec}} = [-0.5, 0.5]^2$ is the rectangular domain and $R_i$ for $i = 1, 2, 3, 4$ are circular regions defined as:

$$\begin{aligned} R_1 &= \left\{ (x, y) : (x - 0.3)^2 + (y - 0.3)^2 \leq 0.1^2 \right\}, \\ R_2 &= \left\{ (x, y) : (x + 0.3)^2 + (y - 0.3)^2 \leq 0.1^2 \right\}, \\ R_3 &= \left\{ (x, y) : (x - 0.3)^2 + (y + 0.3)^2 \leq 0.1^2 \right\}, \\ R_4 &= \left\{ (x, y) : (x + 0.3)^2 + (y + 0.3)^2 \leq 0.1^2 \right\}. \end{aligned}$$

The boundary conditions are specified as:

$$u = 0 \quad \text{on } \partial R_i, \quad \forall i = 1, 2, 3, 4, \tag{159}$$

$$u = 1 \quad \text{on } \partial\Omega_{\text{rec}}. \tag{160}$$

**Complex Geometric Poisson Dataset** To train and evaluate the Physics-Informed Neural Networks (PINNs), collocation points were generated within the defined spatial domains, and boundary conditions were appropriately enforced. A uniform grid was established using 100 equidistant points in each spatial dimension, resulting in $101 \times 101 = 10{,}201$ internal collocation points for the Complex Geometric Poisson problem. Boundary points are sampled from both the outer boundary $\partial\Omega_{\text{rec}}$ and the boundaries of the excluded circular regions $\partial R_i$ for $i = 1, 2, 3, 4$. The PINN was trained in a data-free, unsupervised manner. Upon completion of training, the model was evaluated on the collocation points by comparing the predicted values with the actual values, thereby determining the error.

**3D Point-Cloud Poisson Problem** We investigate a three-dimensional Poisson equation defined on a unit cubic domain, $\Omega = [0, 1]^3$, where the data distribution is represented as a point cloud, capturing the complex geometry introduced by excluding four spherical regions. The governing equation is a non-homogeneous, layered Helmholtz-type partial differential equation given by

$$-\mu(z)\Delta u(\mathbf{x}) + k(z)^2 u(\mathbf{x}) = f(\mathbf{x}) \quad \text{in } \Omega = [0, 1]^3 \setminus \bigcup_{i=1}^{4} \mathcal{C}_i, \tag{161}$$

where the spherical exclusion regions $\mathcal{C}_i$ for $i = 1, 2, 3, 4$ are defined as

$$\mathcal{C}_1 = \left\{ (x, y, z) : (x - 0.4)^2 + (y - 0.3)^2 + (z - 0.6)^2 \leq 0.2^2 \right\}, \tag{162}$$

$$\mathcal{C}_2 = \left\{ (x, y, z) : (x - 0.6)^2 + (y - 0.7)^2 + (z - 0.6)^2 \le 0.2^2 \right\}, \tag{163}$$

$$\mathcal{C}_3 = \left\{ (x, y, z) : (x - 0.2)^2 + (y - 0.8)^2 + (z - 0.7)^2 \le 0.1^2 \right\}, \tag{164}$$

$$\mathcal{C}_4 = \left\{ (x, y, z) : (x - 0.6)^2 + (y - 0.2)^2 + (z - 0.3)^2 \le 0.1^2 \right\}. \tag{165}$$

The material properties exhibit a layered structure at $z = 0.5$, with

$$\mu(z) = \begin{cases} \mu_1 = 1, & z < 0.5, \\ \mu_2 = 1, & z \ge 0.5, \end{cases} \quad k(z) = \begin{cases} k_1 = 8, & z < 0.5, \\ k_2 = 10, & z \ge 0.5. \end{cases} \tag{166}$$

The source term $f(\mathbf{x})$ incorporates strong nonlinearities, defined as

$$f(\mathbf{x}) = A_1 e^{\sin(m_1 \pi x) + \sin(m_2 \pi y) + \sin(m_3 \pi z)} \frac{x^2 + y^2 + z^2 - 1}{x^2 + y^2 + z^2 + 1} + A_2 \left[ \sin(m_1 \pi x) + \sin(m_2 \pi y) + \sin(m_3 \pi z) \right], \tag{167}$$

where the parameters are set to $A_1 = 20$, $A_2 = 100$, $m_1 = 1$, $m_2 = 10$, and $m_3 = 5$. Homogeneous Neumann boundary conditions are imposed on the boundary of the cubic domain, ensuring that

$$\frac{\partial u}{\partial n} = 0 \quad \text{on } \partial\Omega, \tag{168}$$

where $\partial\Omega$ consists of the six faces of the unit cube.

**3D Point-Cloud Poisson Dataset**   The 3D Point-Cloud Poisson Problem dataset is derived from an extensive collection of 65,202 points, each defined by three spatial coordinates $(x, y, z)$ and an associated scalar solution value $u$, collectively representing the solution to a Poisson equation within a three-dimensional domain. To achieve computational feasibility, a randomized subset of 10,000 points is selected from the original dataset for model training and evaluation. This reduced dataset maintains the structural integrity of the original data, with spatial coordinates organized in a $(10{,}000 \times 3)$ matrix and the solution field in a $(10{,}000 \times 1)$ vector. From this subset, a further random selection of 1,000 points constitutes the supervised training set, which includes exact solution values essential for calculating data loss, while the remaining 9,000 points are utilized to enforce physics-informed loss during the training process. This approach ensures computational efficiency while preserving a representative sample of the three-dimensional domain. Subsequently, testing and validation are conducted on the entire reduced dataset to assess the model's predictive accuracy across the domain.

**Tensor Conversion**   : All collocation and boundary points are converted into PyTorch tensors with floating-point precision and are set to require gradients to facilitate automatic differentiation. The data resides on an NVIDIA A100 GPU with 40GB of memory to expedite computational processes.

**Training Regimen:**   All PDE problems are trained for a total of 50,000 epochs to allow sufficient learning iterations. And the RBA attention mechanism for AC-PKAN is configured with smoothing factors $\eta = 0.001$ and $\beta_w = 0.001$.

**Reproducibility:**   To ensure the reproducibility of our experimental results, all random number generators are seeded with a fixed value (seed = 0) across NumPy, Python's `random` module, and PyTorch (both CPU and GPU). This deterministic setup guarantees consistent initialization and training trajectories across multiple runs.

**Hyperparameter Selection:**   For the 3D Point-Cloud Poisson Problem, Table 15 provides a detailed summary of the hyperparameters and the total number of parameters for each model. Similarly, for the other two problems, Table 16 summarizes the hyperparameters and the total number of parameters for each model.

Table 16: Summary of Hyperparameters in Complex Engineering Environmental PDEs for Various Models

| Model | Hyperparameters | Model Parameters |
|---|---|---|
| AC-PKAN | Linear Embedding: in=2, out=32
ChebyKAN Layers: 4 layers, degree=8
LN Layers: 4 layers, features=64
Output Layer: in=64, out=1
Activation: WaveAct | 152,357 |
| QRes | Input Layer: in=2, out=128
Hidden Layers: 5 QRes blocks, units=128
Output Layer: in=128, out=1
Activation: Sigmoid | 166,017 |
| PINN | Layer 1: $2 \to 256$, Activation=Tanh
Layer 2: $256 \to 512$, Activation=Tanh
Layer 3: $512 \to 128$, Activation=Tanh
Layer 4: $128 \to 1$ | 198,145 |
| PINNsformer | d_out=1
d_hidden=128
d_model=8
N=1
heads=2 | 158,721 |
| FLS | Layer 1: $2 \to 256$, Activation=SinAct
Layer 2: $256 \to 256$, Activation=Tanh
Layer 3: $256 \to 256$, Activation=Tanh
Layer 4: $256 \to 1$ | 132,609 |
| Cheby1KAN | Layer 1: $2 \to 32$, Degree=8
Layer 2: $32 \to 128$, Degree=8
Layer 3: $128 \to 64$, Degree=8
Layer 4: $64 \to 32$, Degree=8
Layer 5: $32 \to 1$, Degree=8 | 129,888 |
| Cheby2KAN | Layer 1: $2 \to 32$, Degree=8
Layer 2: $32 \to 128$, Degree=8
Layer 3: $128 \to 64$, Degree=8
Layer 4: $64 \to 32$, Degree=8
Layer 5: $32 \to 1$, Degree=8 | 129,888 |
| KAN[*] | Layers: $2 \times$ KANLinear
Neurons: 9000
Activation: SiLU | 60,000[*] |
| rKAN | Layer 1: $2 \to 256$, Activation=JacobiRKAN(3)
Layer 2: $256 \to 256$, Activation=PadeRKAN[2/6]
Layer 3: $256 \to 256$, Activation=JacobiRKAN(6)
Layer 4: $256 \to 128$, Activation=PadeRKAN[2/6]
Layer 5: $128 \to 1$ | 165,411 |
| FastKAN[*] | FastKANLayer 1: RBF, SplineLinear $16 \to 2600$, Base Linear $2 \to 2600$
FastKANLayer 2: RBF, SplineLinear $20800 \to 1$, Base Linear $2600 \to 1$ | 75,418[*] |
| fKAN | Layer 1: $2 \to 256$, Activation=fJNB(3)
Layer 2: $256 \to 512$, Activation=fJNB(6)
Layer 3: $512 \to 512$, Activation=fJNB(3)
Layer 4: $512 \to 128$, Activation=fJNB(6)
Layer 5: $128 \to 1$ | 132,618 |
| FourierKAN | Layer 1: $2 \to 32$
Layer 2: $32 \to 64$
Layer 3: $64 \to 64$
Layer 4: $64 \to 64$
Layer 5: $64 \to 1$
Degree=8 | 166,113 |

[*] This reaches the GPU memory limit, and increasing the number of parameters further would cause an out-of-memory error.

## F   Results Details and Visualizations.

Firstly, in the context of the 1D-Wave experiment, we present the logarithm of the GRA weights, $\log\left(\lambda_{IC,BC}^{\mathrm{GRA}}\right)$, across epochs in Figure 6. Additionally, the progression of $\lambda_{IC,BC}^{\mathrm{GRA}}$ over epochs is illustrated in Figure 4 (see below).

In Figure 4, we see that the mean RBA weights for all loss terms eventually converge, indicating mitigation of residual imbalance. In contrast, the GRA weights continue to increase, suggesting persistent gradient imbalance. The steadily growing GRA weights effectively alleviate the gradient stiffness problem, consistent with findings in (Wang et al.,

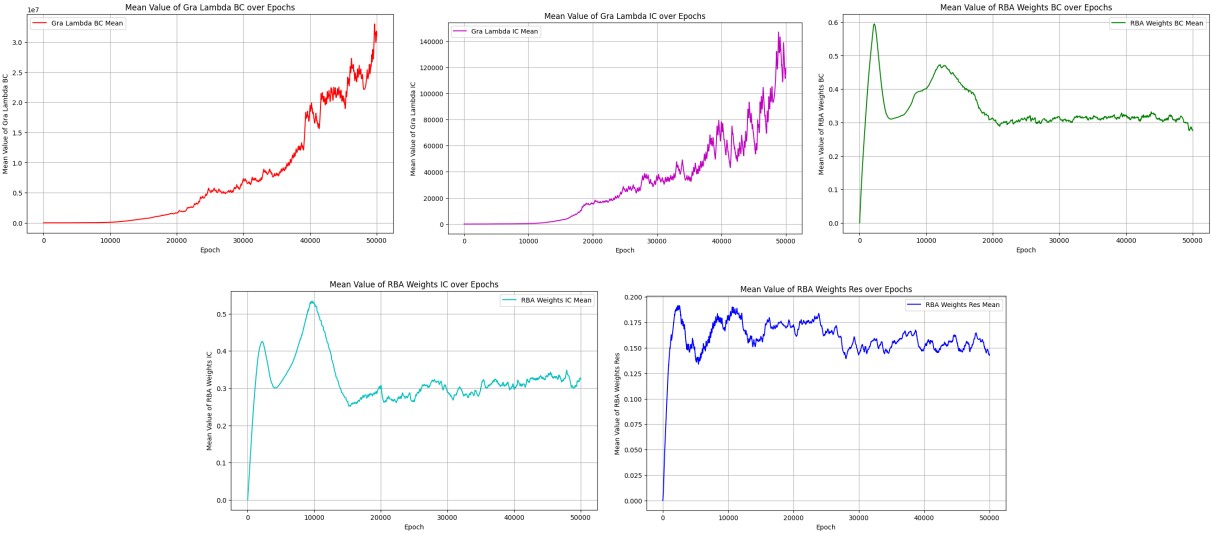

Figure 4: Mean values of GRA and RBA weights over epochs for the 1D-Wave experiment. From left to right in the first row: GRA $\lambda_{BC}$, GRA $\lambda_{IC}$, and RBA weights (BC). Second row: RBA weights (IC) and RBA weights (Residual).

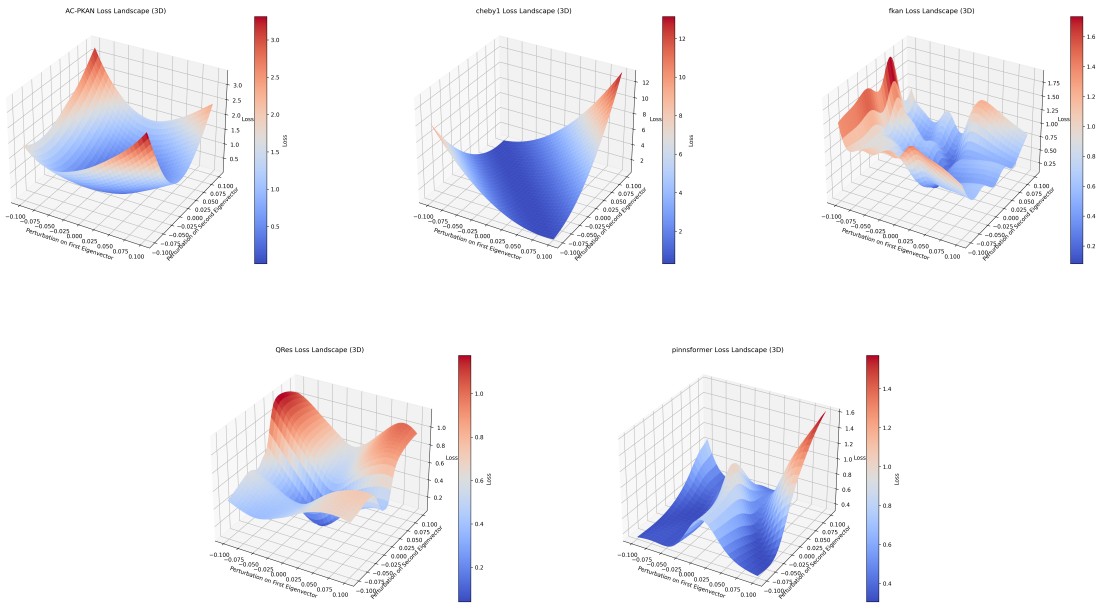

Figure 5: Loss landscapes of various models in the 1D-Wave experiment. From left to right in the first row: AC-PKAN, Cheby1KAN and fKAN. Second row: QRes and Pinnsformer.

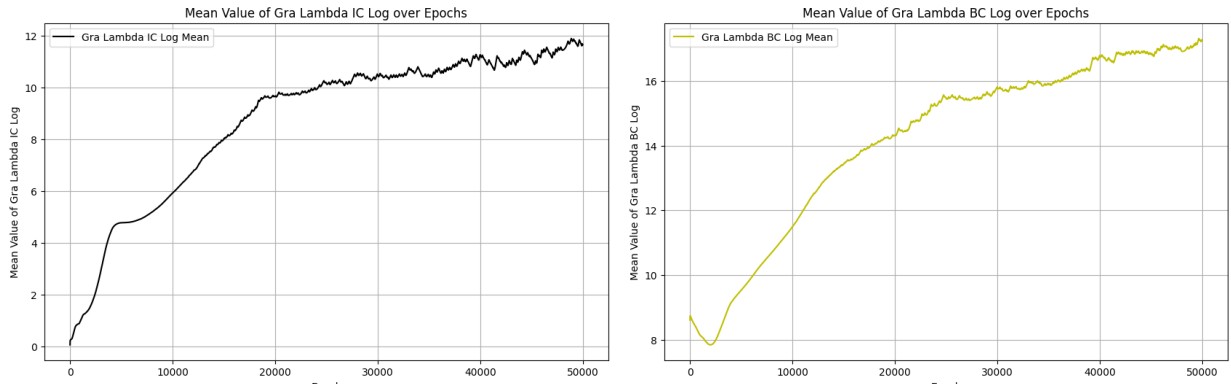

Figure 6: Mean values of GRA weights after logarithmic transformation over epochs for the 1D-Wave experiment.

2021). The significant magnitude discrepancy between GRA and RBA data justifies using a logarithmic function for GRA weights in loss weighting (Figure 6).

Moreover, Figure 5 illustrates the loss landscapes of AC-PKAN, Cheby1KAN, fKAN, QRes, and PINNsFormer. Although Cheby1KAN appears to have a simpler loss landscape, its steep gradients hinder optimization. PINNsFormer, fKAN, and QRes exhibit more complex, multi-modal surfaces, leading to convergence challenges near the optimal point. In contrast, AC-PKAN shows a relatively smoother trajectory, facilitating training stability and efficiency.

Then we illustrate the fitting results of nine models for complex functions in Figure 7. Additionally, we present the plots of ground truth solutions, neural network predictions, and absolute errors for all evaluations conducted in the five PDE-solving experiments. The results for the 1D-Reaction, 1D-Wave, 2D Navier-Stokes, Heterogeneous Poisson Problem, and Complex Geometric Poisson Problem are displayed in Figures 10, 8, 9, and 13, respectively.

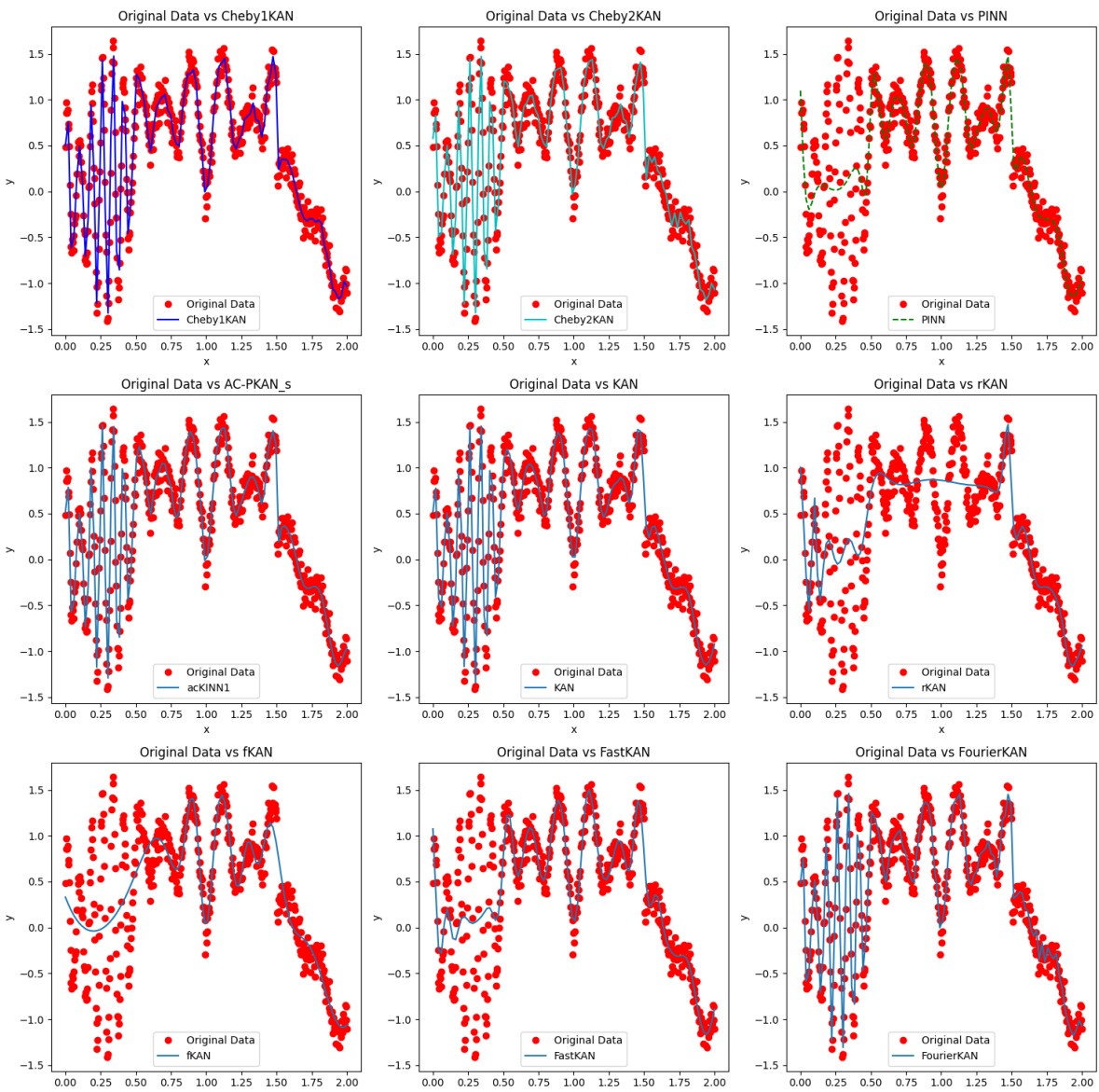

Figure 7: Illustration of 9 Various Models for Complex Function Fitting

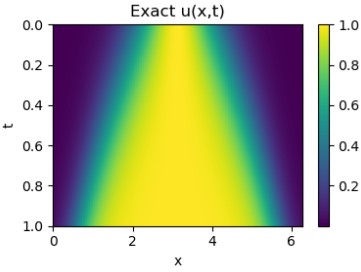

(a) Ground Truth Solution for the 1D-Reaction Equation

(b) From left to right, the first, third, and fifth rows display the predictions of the AC-PKAN, Cheby1KAN, Cheby2KAN, and FastKAN models; the PINNs, QRes, rKAN, and fKAN models; and the PINNsformer, FLS, FourierKAN, and KINN models, respectively. The second, fourth, and sixth rows present their corresponding absolute errors.

Figure 8: Comparison of the ground truth solution for the 1D-Reaction equation with predictions and error maps from various models.

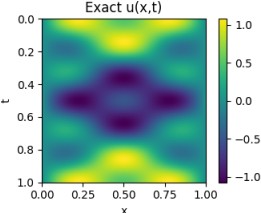

(a) Ground Truth Solution for the 1D-Wave Equation

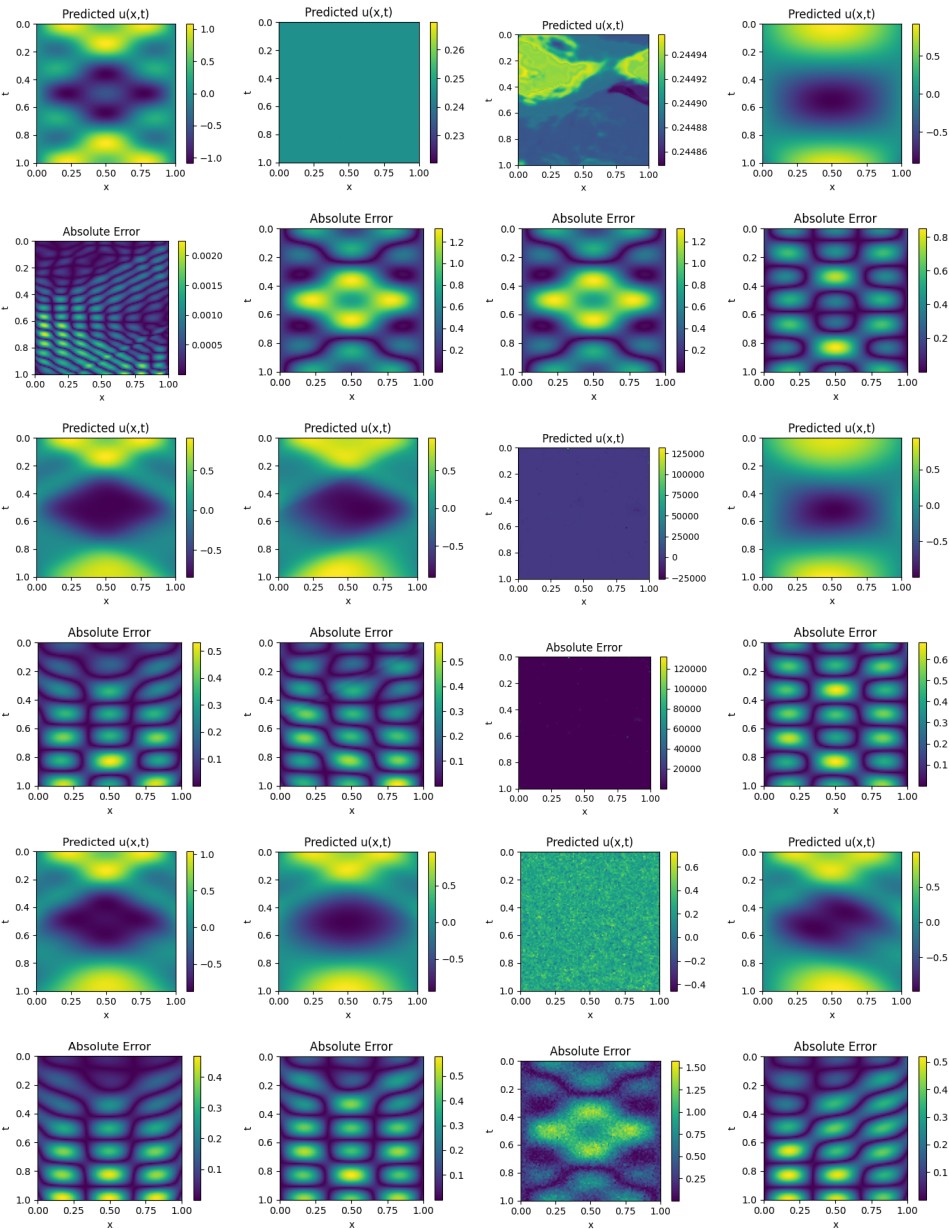

(b) From left to right, the first, third, and fifth rows display the predictions of the AC-PKAN, Cheby1KAN, Cheby2KAN, and FastKAN models; the PINNs, QRes, rKAN, and fKAN models; and the PINNsformer, FLS, FourierKAN, and KINN models, respectively. The second, fourth, and sixth rows present their corresponding absolute errors.

Figure 9: Comparison of the ground truth solution for the 1D-Wave equation with predictions and error maps from various models.

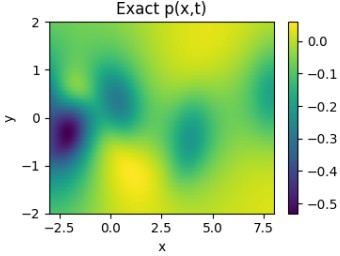

(a) Ground Truth Solution for the 2D Navier–Stokes Cylinder Flow

(b) From left to right, the first, third, and fifth rows display the predictions of the AC-PKAN, Cheby1KAN, Cheby2KAN, and FastKAN models; the PINNs, QRes, and fKAN models; and the PINNsformer, FLS, FourierKAN, and KINN models, respectively. The second, fourth, and sixth rows present their corresponding absolute errors.

Figure 10: Comparison of the ground truth pressure field $P$ of the 2D Navier–Stokes cylinder flow with predictions and corresponding error maps generated by various models.

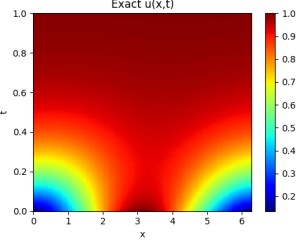

(a) Ground Truth Solution for the 1D-Conv.-Diff.-Reac. Equation

(b) From left to right, the first, third, and fifth rows display the predictions of the AC-PKAN, Cheby1KAN, Cheby2KAN, and FastKAN models; the PINNs, QRes, rKAN, and fKAN models; and the PINNsformer, FLS, FourierKAN, and KINN models, respectively. The second, fourth, and sixth rows present their corresponding absolute errors.

Figure 11: Comparison of the ground truth solution for the 1D-Conv.-Diff.-Reac. Equation with predictions and error maps from various models.

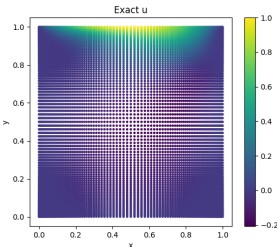

(a) Ground Truth Solution for the 2D Lid-driven cavity flow

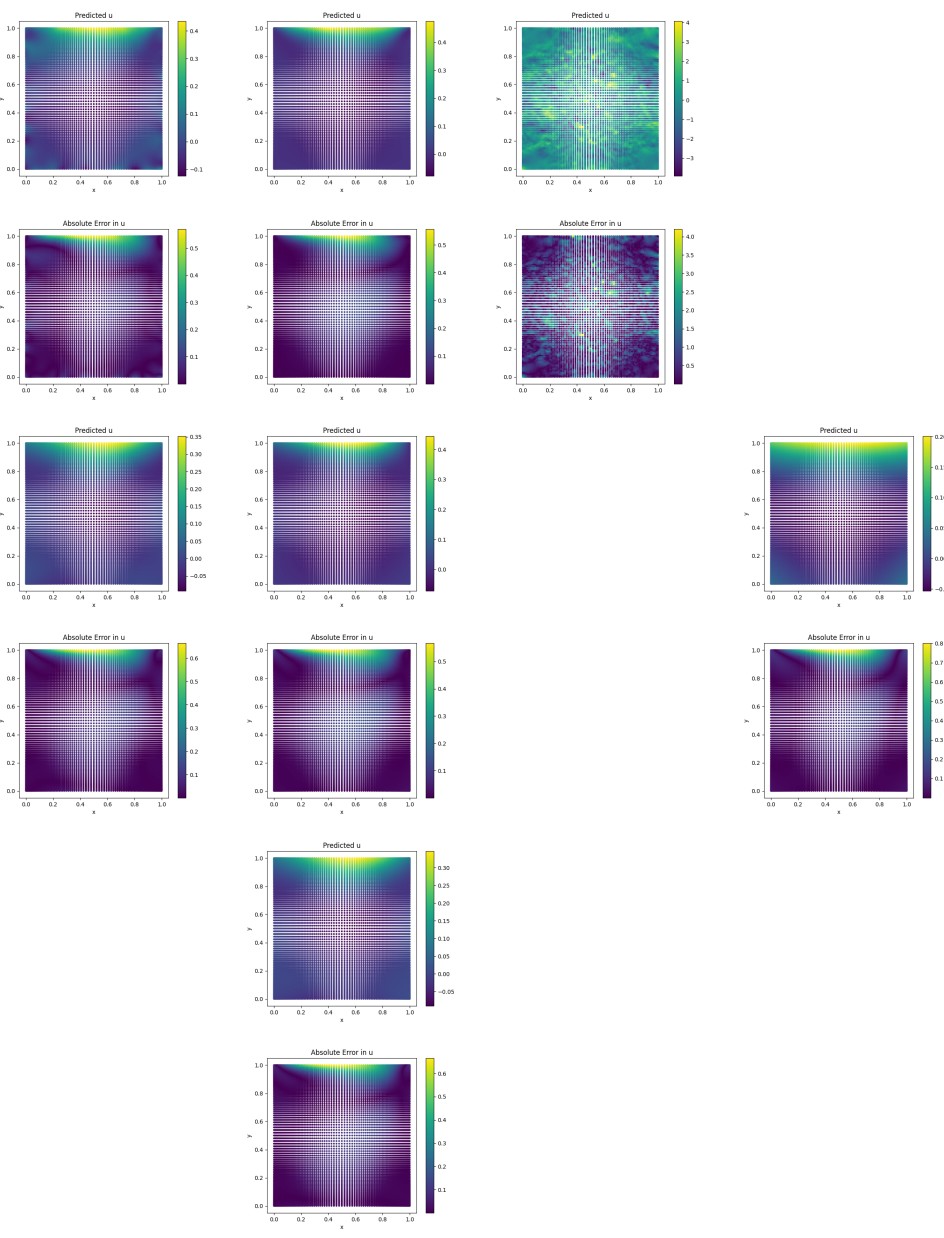

(b) From left to right, the first, third, and fifth rows display the predictions of the AC-PKAN, Cheby1KAN and Cheby2KAN; the PINNs, QRes and fKAN models; and the FLS models, respectively. The second, fourth, and sixth rows present their corresponding absolute errors.

Figure 12: Comparison of the ground truth solution for the 2D Lid-driven cavity flow with predictions and error maps from various models.



(a) Ground Truth Solution for the Heterogeneous Possion equation

(b) From left to right, the first, third, and fifth rows display the predictions of the AC-PKAN, Cheby1KAN, Cheby2KAN, and FastKAN models; the PINNs, QRes, rKAN, and fKAN models; and the PINNsformer, FLS, FourierKAN, and KINN models, respectively. The second, fourth, and sixth rows present their corresponding absolute errors.

Figure 13: Comparison of the ground truth solution for the Heterogeneous Possion equation problem with predictions and error maps from various models.

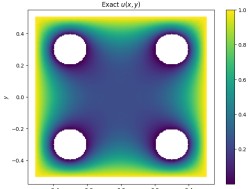

(a) Ground Truth Solution for the Complex Geometry Possion equation

(b) From left to right, the first, third, and fifth rows display the predictions of the AC-PKAN, Cheby1KAN, Cheby2KAN, and FastKAN models; the PINNs, QRes, rKAN, and fKAN models; and the PINNsformer, FLS, FourierKAN, and KINN models, respectively. The second, fourth, and sixth rows present their corresponding absolute errors.

Figure 14: Comparison of the ground truth solution for the Complex Geometry Possion equation problem with predictions and error maps from various models.

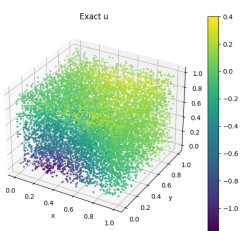

(a) Ground Truth Solution for the 3D Point-Cloud Problem

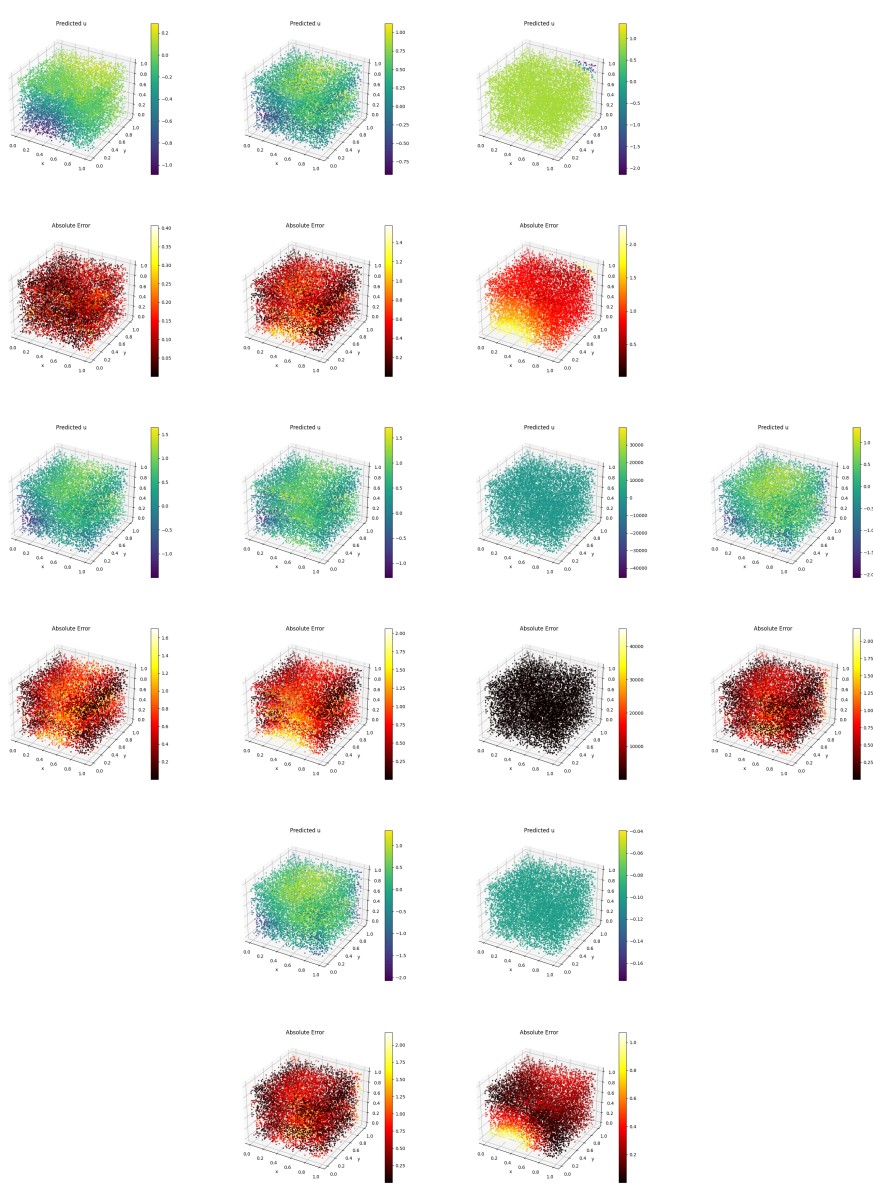

(b) From left to right, the first, third, and fifth rows display the predictions of the AC-PKAN, Cheby1KAN and Cheby2KAN models; the PINNs, QRes, rKAN, and fKAN models; and the FLS and FourierKAN models, respectively. The second, fourth, and sixth rows present their corresponding absolute errors.

Figure 15: Comparison of the ground truth solution for the 3D Point-Cloud Problem with predictions and error maps from various models.

