# OpenReview forum: "AC-PKAN: Attention-Enhanced and Chebyshev Polynomial-Based Physics-Informed Kolmogorov–Arnold Networks"
_TMLR — Accepted by TMLR_

### Review · Reviewer_Dyvd · 2025-08-30

**Summary Of Contributions:**

The paper introduces AC-PKAN, which interleaves Cheby1KAN layers with linear projections, wavelet-based activations, and an internal feature-wise attention mechanism. Together, these preserve full-rank Jacobians and prevent vanishing derivatives, improving expressiveness. Authors evaluates across 9 PDE tasks in three categories (failure modes, heterogeneous/complex geometries, 3D problems) and compares against 12 competing models. Results show AC-PKAN consistently achieves state-of-the-art or near-SOTA performance.

Strengths:
1. This paper provides formal rank-collapse proofs for Cheby1KANs and rigorous guarantees for AC-PKAN’s rank preservation.
2. The proposed AC-PKAN smartly combines Chebyshev bases (efficiency) with wavelet activations (periodicity capture) and attention (feature selection).

Key Weaknesses / Limitations:
1. The paper omits comparisons to operator learners (FNO, DeepONet), assuming they are “orthogonal.” This limits the broader positioning of AC-PKAN.
2. Proofs rely on infinite-width assumptions, while practical networks may still encounter degraded rank or gradient stability.
3. Scalability of AC-PKAN to very high-dimensional PDEs isn’t fully tested.

**Audience:**

Yes

**Audience Explanation:**

This paper will interest audience who're working on using AI methods to solve PDE equations.

**Claims And Evidence:**

Yes

**Claims Explanation:**

The claims are mostly well-supported by both theoretical and empirical evidence.

**Requested Changes:**

It is recommended to provide more discussion on the infinite-width assumptions, as they may not strictly hold in practical scenarios. Additionally, including operator-learning baselines, such as FNO and DeepONet, in the comparison would strengthen the claim of achieving "state-of-the-art" performance.

---

> ### Author Response · Authors · 2025-11-04
> **Reply to Reviewer Dyvd**
>
> ## Response to Weakness 1 & Change 2.
> We added parameter-matched FNO and DeepONet baselines under the same sparse-data protocol as Tables 1–2 across three settings: 1D convection–diffusion–reaction, 2D lid-driven cavity, and 3D point-cloud Poisson. Metrics are reported as rMAE and rRMSE.
>
> | Setting                         | Model     | rMAE   | rRMSE  |
> |---------------------------------|-----------|--------|--------|
> | 1D Convection–Diffusion–Reaction | DeepONet | 0.0730 | 0.1064 |
> | 1D Convection–Diffusion–Reaction | FNO      | 0.0722 | 0.1061 |
> | 2D Lid-Driven Cavity            | DeepONet | 0.6270 | 0.6242 |
> | 2D Lid-Driven Cavity            | FNO      | 0.5605 | 0.5312 |
> | 3D Poisson (Point-Cloud)        | DeepONet | 3.4566 | 2.7978 |
> | 3D Poisson (Point-Cloud)        | FNO      | 2.4388 | 2.0426 |
>
> Across these sparse-supervision tests, compared with Table1-2, operator-learning baselines without physics information underperform AC-PKAN. The updated results appears in the new manuscript.
>
> ## Response to Weakness 2 & Change 1.
>
> We thank the reviewer for this important comment. To address the concern that our proofs rely on infinite-width limits and that practical networks may still suffer rank loss, we have added finite-width theoretical results in the newly uploaded supplementary material and will integrate them into the revised manuscript.
>
> First, under the per-layer Jacobian decomposition $J^{(\ell)}=\mathrm{Diag}(s^{(\ell)})(I+\Delta^{(\ell)})$. We assume
>
> $$
> \min_j |s^{(\ell)}_j| \ge \gamma > 0,\qquad \|\Delta^{(\ell)}\|_2 \le \eta < 1.
> $$
>
> We prove the non-asymptotic lower bound
>
> $$
> \sigma_{\min}\big(J_{\mathrm{total}}\big) \ge [\gamma(1-\eta)]^{L}.
> $$
>
>
>  This result holds at any finite width and quantifies a sufficient condition that prevents end-to-end gradient collapse.
>
> Second, writing the network Jacobian as $J_N(x)=W_{\\mathrm{out}}\\,G(x)$, we show that if the rows of $G(x)$ are independent and isotropic sub-Gaussian and the hidden width satisfies
> $$
> d_h \\ge C K^4\\left(d+\\log\\frac{2}{\\delta}\\right),
> $$
> then with probability at least $1-\\delta$ we have $\\mathrm{rank}\\,J_N(x)=d$. This gives a finite-width, high-probability sufficient condition on width of order $\\tilde{\\Omega}(K^4 d\\log d)$. We also prove that for analytic activations the parameter set that induces rank deficiency is a Lebesgue null set, so generic continuous initializations almost surely avoid that pathology.
>
> The original manuscript’s Sections 3.3 and 3.4 explain how our architectural choices are designed to keep each layer close to the $\\mathrm{Diag}(s)(I+\\Delta)$ regime and to control $\\|\\Delta^{(\ell)}\\|$ in practice. Those design choices are supported by the experiments and ablations reported in the supplement.
>
>
> In short, the infinite-width statements were presented as an instructive limit. The practical guarantees come from the finite-width non-asymptotic bounds and from architecture and training choices that align practice with the theoretical conditions. We will include the full finite-width proofs and an expanded discussion in the revised manuscript.
>
>
> ## Response to Weakness 3.
>
> Thank you for raising the scalability concern.  As stated in Discussion and Limitations in revised manuscript, full tests on genuinely high dimensional are out of scope for this version due to memory constraints. We list this as a limitation and will report ultra deep and very high dimensional results once larger memory hardware is available in future work.

---

> > ### Comment · Reviewer_Dyvd · 2025-11-24
> >
> > Thanks for the response. Most of my concerns have been addressed.

---

### Review · Reviewer_LMMU · 2025-10-05

**Summary Of Contributions:**

This paper proposes an extended KAN-based approach for solving PDEs. The approach draws on the idea of swapping B-splines in KANs with Chebyshev polynomials. First, theoretical analysis is provided to demonstrate that stacking plain ChebyshevKANs tends to collapse rank. To tackle this limitation, the proposed approach adds an internal attention block around ChebyshevKAN and wavelet layers to preserve a full-rank Jacobian, and additionally incorporates an external attention mechanism (RGA) that adapts loss weights using residual magnitudes and gradient norms. This combined approach,  dubbed AC-PKAN, is shown to mostly outperform prior PINN/KAN variants on nine benchmarks. A proper review of prior work is provided. Extensive ablation studies, including the transferability of the proposed new components, are provided.  Theoretical proofs seem correct to me (though haven't checked thoroughly). The paper is well written and structured. Overall, I find this work interesting, however, there are several concerning aspects which warrant clarification and further consideration.

**Additional Comments:**

Minor comments:

1. On page 4, "Equation equation 8" -> "Equation 8"
2. The text contains both "Chebyshev1kan" and "Cheby1KAN". Standardize.

**Audience:**

Yes

**Audience Explanation:**

This work would be of interest to the community focusing on PINNs and ML-based approaches to PDEs.

**Claims And Evidence:**

Yes

**Claims Explanation:**

Most of the claims seem correct with, yet there are several inconsistencies and potential overstatements as detailed in the other parts of my review.

**Requested Changes:**

Below are my concerns, questions and suggestions:


1. The abstract mentions that AC-PKAN consistently outperforms or matches SOTA methods. However, the results in Tables 1 and 2, do not support this claim. The language should be adjusted.

2. On page 9 it is stated that the internal architecture of AC-PKAN is powerful even without RGA, yet in Table 3, AC-PKAN without RGA is far worse than multiple models with RGA. This should be rephrased to avoid overclaiming the standalone strength.

3. The manuscript would benefit from a dedicated "Discussion and Limitations" section before the conclusion. While the appendices include a brief discussion (computational complexity) and a limitations note (optimizer/pruning), the scalability of AC-PKAN is not discussed in the main text. I would recommend adding a discussion on the scalability of AC-PKAN and its components (e.g., Chebyshev degree N, layer widths/depth etc. ) as well as outlining directions for future research.

4. Define rMAE/rRMSE precisely.

5. The statement in Theorem 3 seems to hold under i.i.d. Gaussian coefficients. Trained networks do not necessarily satisfy this assumption. There should be an accompanying note explicitly stating this.

6. Further to the above comment, since rank is central, add supporting empirical results illustrating the decay across depth for Cheby1KAN vs AC-PKAN and showing preserved feature diversity.

7. A large fraction of the references in this paper are arXiv preprints or GitHub repositories rather than peer-reviewed papers. While it is commendable that the authors consider recent benchmarks, this heavy reliance on unpublished sources makes the empirical landscape somewhat fluid. This does not invalidate the contribution of this work, however, it weakens the weight of the comparative claims.

---

> ### Author Response · Authors · 2025-11-04
> **Reply to Reviewer LMMU**
>
> Thanks for your insightful comments.
>
> ## Response to Change 1 & 2
>
> We revised the uploaded PDF to replace strong claims with more modest and accurate language to avoid overstatement. However, AC-PKAN remains competitive with state-of-the-art methods and surpasses them on several tasks. Meanwhile, in the function-fitting experiments reported in Table (a), the internal AC-PKAN architecture without RGA exhibits clear representational strength.
>
> ## Response to Change 3
>
> We revised the appendix limitations and expanded them into a main-text **Discussion and Limitations** section. The update adds a concise discussion of scalability and outlines directions for future research. The revised PDF has been uploaded.
>
> ## Response to Change 4
>
> The definitions of rMAE and rRMSE are provided in Eq. (123) of the original manuscript.
>
> ## Response to Change 5
>
> We thank the reviewer for this observation. We agree that Theorem 3 is proved under a random-coefficient model with i.i.d. Gaussian coefficients at initialization, and that trained networks do not necessarily satisfy independence or Gaussianity. For this reason, Theorem 3 should be read as a motivational sufficient-condition result rather than a universal claim about learned models. In the revised manuscript we have added a remark after Theorem 3 to make this scope explicit and to avoid over-extension. For statements about trained networks, our conclusions rest primarily on the deterministic layerwise rank bounds (Theorems 1–2) in the paper.
>
> ## Response to Change 6
>
> Thank you for the suggestion. A full layerwise estimation of Jacobian rank and a complete feature decomposition under much deeper stacking require retaining large activations and Jacobian blocks through backprop. The memory and compute cost grow steeply with depth and exceed the limits of our available 40GB GPUs, which makes controlled comparisons at the theoretically relevant depths infeasible for this submission. We therefore label this as a scalability limitation in the Discussion and Limitations section and plan to perform systematic ultra-deep evaluations in follow-up work, reporting layerwise rank curves and ablations once larger-memory hardware is available.
>
> ## Response to Change 7
>
> We thank the reviewer for raising the issue of citation quality. In the original draft we cited several GitHub repositories because only early implementations were available at the time. In the revised PDF we have replaced those entries with references to the corresponding formal publications.
>
> ## Response to Comments
>
> Thank you for pointing out the typos. We have corrected them in the revised manuscript.

---

> ### Comment · Reviewer_LMMU · 2025-11-19
> **Response to Authors' Reply**
>
> I would like to thank the authors for their response and efforts put on the revision. Most of my concerns and comments have been addressed in the revision. While the author's didn't fully address the comment concerning supporting empirical results on rank decay, an acceptable justification was provided.
>
> Below are my comments on the revised manuscript:
>
> 1.  The formal definition of rMAE and rRMSE and the details of the experimental setup were relegated to the appendix. There should be a sentence in the beginning of Sec. 4 referring the reader to the appendix.
>
> 2. In the response, the authors state that a clarifying remark was added after Theorem 3. The authors' response in the comments -- "We agree that Theorem 3 is proved under a random-coefficient model with i.i.d. Gaussian coefficients at initialization, and that trained networks do not necessarily satisfy independence or Gaussianity. For this reason, Theorem 3 should be read as a motivational sufficient-condition result rather than a universal claim about learned models." -- is much more clearer than the added paragraph. I would recommend adding the quoted text to the paragraph after Theorem 3.

---

### Review · Reviewer_pTHb · 2025-10-28

**Summary Of Contributions:**

This paper proposes AC-PKAN, a novel hybrid neural architecture that combines Chebyshev Type-I based Kolmogorov–Arnold Networks (Cheby1KANs) with attention mechanisms for use in Physics-Informed Neural Networks (PINNs). This work introduces multiple enhancements aimed at addressing major limitations of existing PINNs and KANs in particular, such as residual and border condition losses unbalance, and their risk of rank collapse. To address these limitations, the paper proposes to use a dedicated architecture that interleaves Cheby1KANs layers and linear projection layers, combined with wavelet activations and feature-wise attention to preserve the full rank of the Jacobian Matrix. A Residual Gradient Attention (RGA) is also introduced as a way to automatically weight losses during training.

**Additional Comments:**

* The related work section lacks positioning and is restricted to pointing out existing related methods in the literature. Having a clear highlighting of current challenges and how the proposed methods answer them would greatly improve the paper.


* It is not clear how the learnable linear weighting described as an attention mechanism in Section 3.3 corresponds to the concept of attention commonly used in the broader machine learning literature.

* The width of a KAN could be defined as this notion might be ambiguous between node width and functional width.

* The method lacks theoretical or, at least, intuitive justifications. In particular, the rationale behind the architectural adaptation sec. 3.3 could be made explicit.

**Audience:**

Yes

**Audience Explanation:**

This paper is well in phase with current development around machine learning for physical models. Its propositions might bring valuable insights into functional architectures and their use in physics-informed learning.

In particular, the main contributions: Chebyshev-KANs with internal/external attention as well as the GRA appear original.

**Claims And Evidence:**

Yes

**Claims Explanation:**

The claims are supported with both clear theoretical and empirical justifications. Overall, the paper is clear and easy to follow. In particular, the preliminary sections (3.1-3.2) provide a clear understanding of the problem and the limitations of KAN networks.


However, despite being very well written, there remain a few points that might be dug further:

Most of the rank collapse robustness discussion is based on infinite-width KAN, which might not translate when dealing with finite-width KAN. The improved stability of the approach remains thus empirical.

In comparison to standard PINNs (MLP‐based) or simpler KAN variants, AC-PKAN is far more complex. It is rather difficult to understand if the improvements are provided by the parameter overhead, a subset of new modules, or fine-grained parameter tuning. The benchmarks chosen are standard but do not fully illustrate the model capacity on complex systems.

**Requested Changes:**

## Model complexity

Many new modules are incorporated into the presented work to improve Chebychev KANs stability and expressivity. Their contribution is evaluated in a dedicated with/without each module ablation *cf* Tab. 4. The study shows drops each time a component of the method is removed. However, this ablation should be pushed further in order to disentangle module interactions. In its current state, it is rather difficult to understand from the ablations if the different modules are truly contributing as much as they seem to be.

In comparison to standard PINNs (MLP‐based) or simpler KAN variants, AC-PKAN has more architectural hyper-parameters: Number of ChebyKAN layers, Degree of Chebyshev polynomials, Wavelet activation parameters, Internal attention feature-wise modules (dimensions, heads), External RGA dynamic loss-weighting hyper-parameters (e.g., learning rate for RBA, moving average parameters for GRA).

KAN (especially Chebychev KAN) scales quadratically with the dimension, as high polynomial degrees are needed to represent sharp, local variations, and each feature dimension is related to a given polynomial expansion. Furthermore, each introduced projections add a number of parameters equal to the number of input features times the number of output ones, which might also add a dramatic parameter overhead for high-dimensional ODE.

Adding more ablations exploring the different module combinations, as well as a dedicated model stability towards introduced hyperparameters, would give great insight into the method's capacity. Providing a discussion about parameter overhead depending on the feature state-space dimension is also critical.

## Simplicity of the evaluation


Finally, the model was evaluated on nine benchmark problems, including classical low-dimensional PDEs (1D wave equation, 1D Burgers’ equation, 2D Poisson equation,...), function fitting and ODE examples (e.g. Lotka–Volterra-like systems), and a few regression tasks. This evaluation seems to be relatively common across state-of-the-art and shows promising results. However, the dimension of the systems used for evaluation is relatively small (less than 4 input variables). This tends to weaken the methodological contributions aiming at augmenting the model expressivity and limiting rank collapse, and it is difficult to assess if they can actually scale to more complex scenarios. I wonder if the method could be used on high-dimensional state-space ODE *e.g.* Lorenz-96. Discussing this point, even as a limitation or future work, might improve the paper.

---

> ### Author Response · Authors · 2025-11-04
> **Reply to Reviewer pTHb**
>
> Thanks for your insightful comments.
>
> ## Response to Change 1
>
> We performed a focused ablation study on the 1-D wave equation to isolate the contributions of three design choices. The three baselines are defined as follows.
>
> - **AC-PKAN(min)**: Cheby1KAN plus linear input and output projections (`W_emb` and `W_out`), without wavelet activation, without the internal feature attention (the U/V gating and α(l) injection), and without RGA. This is the minimal modification of Cheby1KAN that preserves the linear projection channel which helps against rank collapse.
> - **AC-PKAN(no-attn)**: Cheby1KAN plus linear projections and wavelet activation, but with internal feature attention disabled and no RGA. This isolates the effect of the wavelet frequency activation combined with MLP-style projections.
> - **AC-PKAN(sin)**: Cheby1KAN plus linear projections and internal feature attention, but with the wavelet activation replaced by a simpler sine activation, and still no RGA. This isolates whether the layer-wise feature gating α(l) is the critical element for mitigating rank collapse and coupling in high dimensions.
>
> The quantitative results on the 1-D wave test set are shown below.
>
> |Model|rMAE|rRMSE|
> |-|-|-|
> |AC-PKAN(min)|0.5374|0.5495|
> |AC-PKAN(no-attn)|0.5116|0.5081|
> |AC-PKAN(sin)|0.4478|0.4391|
>
> These results lead to three main conclusions. First, adding only the linear projection is not sufficient to explain the full performance gain. Second, the internal feature attention provides the largest single improvement in this ablation axis. Third, wavelet activation contributes positively, but its effect size is smaller than the effect of the internal gating. Linear up and down projections serve as a necessary embedding context that widens channel dimensions, but they are not the primary source of the observed gains.
>
> Taken together, the ablations support the claim that the proposed layer-wise internal feature attention is a major driver of stability and scalability for Cheby1KAN in these settings.
>
> We then conduct another ablation study on Chebyshev polynomial degree. We ran a controlled hyperparameter sweep over the Chebyshev polynomial degree \(N\) while keeping the full AC-PKAN architecture and all other training settings fixed. The goal is to isolate how the polynomial degree introduced by the KAN backbone affects final accuracy on the 1-D wave equation.
>
> |Degree|rMAE|rRMSE|
> |-|-|-|
> |4|0.0196|0.0200|
> |6|0.0200|0.0205|
> |8|0.0011|0.0011|
> |10|0.0128|0.0131|
>
> These results show a clear nonmonotonic relationship between polynomial degree and final error. Low degrees (4 and 6) are underexpressive on this problem and produce errors near 2×10⁻². Degree 8 yields a dramatic improvement and reaches the 1.1×10⁻³ regime. Pushing to degree 10 increases error again to the 10⁻² scale. This pattern supports three conclusions. First, simply increasing polynomial order is not a reliable path to better accuracy. Second, the best performance emerges from a combination of an appropriately chosen orthogonal basis together with our internal feature re-injection attention and frequency-domain activation. Third, very high polynomial order amplifies gradient magnitudes and worsens conditioning, which raises optimization difficulty and erases some of the representational advantage. In short, AC-PKAN’s peak accuracy results from the joint effect of choosing a suitable degree and the proposed internal and frequency-domain modules, not from naive parameter scaling.
>
> ## Response to Change 2
>
> We acknowledge the reviewer’s point on high-dimensional benchmarks. Our current tests focus on low- to medium-dimensional problems to validate representational gains and rank-collapse mitigation in controlled settings. Scaling AC-PKAN to high-dimensional chaotic systems like Lorenz–96 introduces extra challenges in time integration, long-horizon stability, and gradient conditioning. Addressing these issues requires work on sampling, gradient preconditioning, and multi-step time handling. We plan to pursue these extensions and have noted this limitation in the manuscript.
>
> ## Response to Comment 1
>
> Thank you for the suggestion. We have uploaded an updated PDF with a more logically organized Related Works section.
>
> ## Response to Comment 2
>
> In Section 3.4 we call the external loss reweighting an "attention" mechanism. This is not intended to equate it with Transformer-style self-attention. The term follows usage in the PINN literature, most directly the work titled "Residual-based attention in physics-informed neural networks" by Anagnostopoulos et al. That paper uses learnable scalar weights that adapt to residual magnitude to focus the optimizer on high-error points and constraint types. Functionally, this approach adjusts per-point loss weights and does not perform query–key–value content matching as in mainstream attention modules.

---

> ### Author Response · Authors · 2025-11-04
>
> ## Response to Comment 3
> In manuscript, the term "width" for a KAN denotes the functional width, namely the number of Chebyshev basis functions used per edge (N+1), not the neuron count of an MLP layer. We have clarified this distinction in the revised PDF.
>
> ## Response to Comment 4
> Thank you for the comment. In Section 3.2 we show that naively stacking Chebyshev-based KAN layers causes a quantifiable decay in the effective rank of the network Jacobian due to repeated nonlinear combinations. This rank collapse leads to the familiar gradient stiffness of PINNs on high-dimensional PDEs. To remedy this failure mode, Section 3.3 introduces three complementary components: linear mixing projections that reblend per-edge Chebyshev coefficients into a higher-dimensional feature space, a learnable wavelet-style modulation that preserves nonzero higher-order derivatives, and a lightweight feature attention that gates and fuses information across channels. Taken together, these changes convert each layer into a high-frequency representation plus a full-rank projection with gated fusion, which restores the Jacobian’s effective rank while keeping higher derivatives differentiable. Finally, the residual gradient attention mechanism (RGA) compresses the gradient magnitude imbalance introduced by Chebyshev expansions and stabilizes optimization. In short, the architectural modifications in Section 3.3 follow directly from the theoretical failure mode identified in Section 3.2 and jointly address representation, propagation, and optimization.

---

> > ### Comment · Reviewer_pTHb · 2025-12-09
> >
> > I would like to thank the authors for their dedicated rebuttal. Overall, I am fully satisfied with their answer.
> >
> > My only remaining concern is about the high impact of the polynomial degree, which tends to exhibit the need for fine-grained parameter tuning. If this parameter does not remain stable across datasets, it could represent a limitation of the method and should be discussed as such.

---

### Decision · Action_Editor_8LGr · 2025-12-22

**Recommendation:** Accept with minor revision

**Additional Comments:**

This paper presents enhanced Kolmogorov–Arnold Networks (KANs) and introduces a new physics-informed neural network for solving physical systems governed by partial differential equations (PDEs). The authors incorporate Chebyshev polynomials, attention blocks, and wavelet layers to preserve a full-rank Jacobian, along with an external attention mechanism (RGA).

The paper initially received positive reviews, with reviewers acknowledging its clarity and the relevance of its contributions. However, reviewers also raised concerns regarding the experimental validation, requesting additional ablation studies to better isolate the contributions, comparisons to recent baselines such as neural operators, and a discussion of the method's limitations and the theorem's assumptions. The authors' rebuttal effectively addressed these concerns, and after the discussion period, all reviewers recommended acceptance.

The AE has carefully reviewed the submission and the discussion. The AE considers the enhanced Kolmogorov–Arnold Networks (KANs) to be a relevant contribution, offering both theoretical insights and experimental validation. The claims are therefore well-supported by evidence, and the discussion period enabled the inclusion of additional results and analyses that have improved the paper's quality.

Therefore, the AE recommends acceptance, pending the authors' incorporation of the promised revisions: formal definitions of rMAE and rRMSE and a clarifying remark for Theorem 3 (RLMMU), plus full finite-width proofs and expanded discussion (RDyvd).

**Audience:**

Yes

**Audience Explanation:**

This paper focuses on Physics-Informed Neural Network (PINNs), a subject of considerable interest within the TMLR community.

**Claims And Evidence:**

Yes

**Claims Explanation:**

The paper proposes a new physics-informed method using improved Kolmogorov–Arnold Networks (KANs) to solve PDEs. The approach is supported by theoretical guarantees and experimental validations.